# CSPP1 stabilizes growing microtubule ends and damaged lattices from the luminal side

Cyntha M. van den Berg[1], Vladimir A. Volkov[1,2], Sebastian Schnorrenberg[3], Ziqiang Huang[3], Kelly E. Stecker[4,5], Ilya Grigoriev[1], Sania Gilani[6,7], Kari-Anne M. Frikstad[6], Sebastian Patzke[6], Timo Zimmermann[3], Marileen Dogterom[2], and Anna Akhmanova[1]

**Microtubules are dynamic cytoskeletal polymers, and their organization and stability are tightly regulated by numerous cellular factors. While regulatory proteins controlling the formation of interphase microtubule arrays and mitotic spindles have been extensively studied, the biochemical mechanisms responsible for generating stable microtubule cores of centrioles and cilia are poorly understood. Here, we used in vitro reconstitution assays to investigate microtubule-stabilizing properties of CSPP1, a centrosome and cilia-associated protein mutated in the neurodevelopmental ciliopathy Joubert syndrome. We found that CSPP1 preferentially binds to polymerizing microtubule ends that grow slowly or undergo growth perturbations and, in this way, resembles microtubule-stabilizing compounds such as taxanes. Fluorescence microscopy and cryo-electron tomography showed that CSPP1 is deposited in the microtubule lumen and inhibits microtubule growth and shortening through two separate domains. CSPP1 also specifically recognizes and stabilizes damaged microtubule lattices. These data help to explain how CSPP1 regulates the elongation and stability of ciliary axonemes and other microtubule-based structures.**

## Introduction

Microtubules (MTs) are dynamic cytoskeletal polymers that serve as tracks for intracellular transport and drive chromosome separation during cell division. The majority of cellular MTs turn over rapidly because MTs frequently switch between phases of growth and shortening (Desai and Mitchison, 1997). Proteins controlling MT dynamics in interphase and mitosis have been studied in detail by a combination of genetic, cell-biological, biochemical, and biophysical experiments (Akhmanova and Steinmetz, 2015; Gudimchuk and McIntosh, 2021). In particular, in vitro reconstitution studies with purified components have been instrumental in understanding the mechanisms underlying the activity of these proteins (Bieling et al., 2007; Gell et al., 2010). However, cells also form stable MT-based structures, such as centrioles and cilia. Multiple molecular players responsible for their biogenesis have been identified, but their biochemical properties and their effects on MT dynamics are very poorly understood because most of them have never been investigated using purified proteins. Furthermore, most studies of MT dynamics have been focused on proteins binding to the outer MT surface and not to the MT lumen. However, stable MTs, such as those in cilia and neurons, often contain intraluminal particles (ILPs; Cuveillier et al., 2020; Garvalov et al., 2006; Nicastro et al., 2006; Sui and Downing, 2006). Recent advances in structural analysis generated high-resolution density maps, which enabled the identification of MT inner proteins (MIPs), organized in a repetitive pattern in motile cilia and flagella (Gui et al., 2021; Ma et al., 2019). Intraluminal proteins were also sporadically found in primary cilia, but their identity and mechanisms of action are unknown (Kiesel et al., 2020). Apart from proteins, also small molecules like taxanes can bind to the luminal side of MTs and stabilize them (reviewed in Steinmetz and Prota, 2018), but it is currently unknown whether any features of MT stabilization from the luminal side are shared by proteins and drugs.

Here, we focused on the centrosome/spindle pole associated protein 1 (CSPP1; Patzke et al., 2005; Patzke et al., 2006), the orthologues of which are found across a broad variety of ciliated eukaryotes, including all vertebrates, monoflagellates, and choanoflagellates, though not worms or flies (EggNOG v5.0 database; Huerta-Cepas et al., 2019). Previous work established

[1]Cell Biology, Neurobiology and Biophysics, Department of Biology, Faculty of Science, Utrecht University, Utrecht, The Netherlands;   [2]Department of Bionanoscience, Kavli Institute of Nanoscience, Delft University of Technology, Delft, The Netherlands;   [3]EMBL Imaging Centre, EMBL-Heidelberg, Heidelberg, Germany;   [4]Biomolecular Mass Spectrometry and Proteomics, Bijvoet Center for Biomolecular Research and Utrecht Institute for Pharmaceutical Sciences, Utrecht University, Utrecht, The Netherlands;   [5]Netherlands Proteomics Center, Utrecht, The Netherlands;   [6]Department of Radiation Biology, Institute of Cancer Research, Norwegian Radium Hospital, Oslo University Hospital, Oslo, Norway;   [7]Department of Molecular Cell Biology, Institute of Cancer Research, Norwegian Radium Hospital, Oslo University Hospital, Oslo, Norway.

Correspondence to Anna Akhmanova: a.akhmanova@uu.nl

V.A. Volkov's current affiliation is School of Biological and Behavioural Sciences, Queen Mary University of London, London, UK.



that CSPP1 binds to spindle poles and the central spindle during mitosis and to ciliary axonemes, centrosomes, and centriolar satellites in interphase (Asiedu et al., 2009; Frikstad et al., 2019; Patzke et al., 2005; Patzke et al., 2010; Patzke et al., 2006). CSPP1 accumulates at ciliary tips, interacts with several other ciliary tip proteins, contributes to ciliogenesis, and controls the axoneme length. Loss of CSPP1 leads to the formation of shortened cilia and impaired Hedgehog signaling, which depends on ciliary function (Frikstad et al., 2019; Latour et al., 2020; Patzke et al., 2010). Mutations in genes encoding CSPP1 and its ciliary binding partners lead to defects in ciliogenesis and result in a range of ciliopathies, such as the neurodevelopmental disorder known as Joubert syndrome, or the more severe Meckel-Gruber syndrome with multiple developmental abnormalities (Akizu et al., 2014; Latour et al., 2020; Shaheen et al., 2014; Tuz et al., 2014).

While the tissue and cellular phenotypes associated with CSPP1 defects have been analyzed in some detail, very little is known about its mechanism of action. To close this knowledge gap, we have performed in vitro reconstitution experiments and found that CSPP1 specifically associates with growing MT ends when their polymerization is slowed down or perturbed. CSPP1 stabilizes such ends and induces MT pausing followed by growth. This effect of CSPP1 on MT behavior strikingly resembles that of MT-stabilizing compounds, taxanes and epothilones (Rai et al., 2020). Using cryo-electron tomography (cryo-ET) and MINFLUX microscopy, we found that CSPP1 localizes to the MT lumen, similar to taxanes. We also observed that CSPP1 efficiently binds to and stabilizes damaged MTs. Altogether, our findings reveal how MT dynamics can be controlled from the luminal side. These data have important implications for understanding how highly stable MT populations, such as those in ciliary axonemes, are generated and maintained.

## Results

### CSPP1 suppresses catastrophes by binding to polymerizing ends where it induces pausing

CSPP1 contains several predicted helical domains interspersed with regions of unknown structure and is represented by two isoforms, the long isoform CSPP-L and a shorter isoform, termed here CSPP-S, which lacks 294 amino acids at the N-terminus and contains an internal deletion of 52 amino acids (Fig. 1 A; Frikstad et al., 2019; Patzke et al., 2006). The middle region of CSPP1 was previously identified as a MT organization domain and the C-terminal region as the centrosome-targeting domain (Patzke et al., 2006). To gain insight into the autonomous effects of CSPP1 on MT dynamics, we have purified N-terminally tagged CSPP-L from HEK293 cells (Fig. S1 A). Mass spectrometry-based analysis (Fig. S1 B) demonstrated that CSPP-L preparations contained no other known regulators of MT dynamics but did contain a small amount of the known CSPP1 interactor PCM1 (Frikstad et al., 2019; Shearer et al., 2018), as well as some contamination with the heat shock protein Hsp70, which we often observe in our protein preparations and which, to our knowledge, has no effect in MT dynamics. We used purified

GFP-CSPP-L to perform in vitro assays where MTs grown from GMPCPP-stabilized seeds were observed by total internal reflection fluorescence (TIRF) microscopy (Fig. S1 C; Aher et al., 2018; Bieling et al., 2007). The use of single MTs to study CSPP1 behavior is appropriate because the tips of primary cilia, where CSPP1 acts, are composed of MT singlets (Kiesel et al., 2020).

In the presence of tubulin alone, MTs regularly switched from growth to shortening that proceeded all the way back to the seed. However, the addition of 10 nM CSPP-L suppressed shrinkage and led to frequent pausing of MT plus ends, while their growth rate was slightly reduced, though not to values characteristic for the minus ends (Doodhi et al., 2016), and the two MT ends could therefore still be reliably distinguished based on their dynamics (Fig. 1, B, C, E, and F; Fig. S1 D; and Video 1). Pausing and suppression of shrinkage were also observed when we included in the assay mCherry-EB3, a marker of growing MT ends, which by itself increases MT growth rate and promotes catastrophes (Fig. 1, D–F; Komarova et al., 2009). In our in vitro assays, CSPP-L also bound to growing MT minus ends and strongly accumulated along the lattice formed by minus-end polymerization (Fig. 1, B–D). However, in cells, this protein normally acts at the distal tip of the cilium, which contains MT plus ends; therefore, we have not investigated the effects of CSPP-L on MT minus-end dynamics. CSPP-L binding was always initiated close to the growing MT end, and after binding, CSPP-L showed very little lateral diffusion along MTs so that CSPP-L binding zones remained well-confined (Fig. 1, C and D). The low lateral mobility of CSPP1 was confirmed by spiking experiments where 0.5 nM GFP-CSPP-L was combined with 9.5 nM mCherry-CSPP-L (Fig. S1 E). When CSPP-L concentration was increased, the zones of CSPP-L accumulation coincided with longer and more frequent MT pausing events (Fig. 1, D–F). CSPP-L-induced pausing was almost always (in ~95% of the cases) followed by MT growth and not by shrinkage, and at CSPP-L concentrations exceeding 5 nM, very little MT depolymerization was observed (Fig. 1, B–F and Fig. S1 E). At a low, 0.5-nM concentration of CSPP-L, long MT depolymerization episodes were still present, but zones of CSPP-L accumulation triggered MT rescues (Fig. 1, D and F).

To investigate whether these in vitro assays recapitulate some aspects of the cellular behavior of CSPP1, we next analyzed its localization and dynamics in cells. In interphase RPE-1 cells stably expressing very low levels of CSPP-L N-terminally tagged with the monomeric NeonGreen protein (mNG-CSPP-L), the protein was localized to centrosomes and PCM1-positive centriolar satellites but not to cytoplasmic MTs, as described previously (Frikstad et al., 2019; Fig. S1 F). PCM1 is a scaffolding protein essential for the formation of centriolar satellites (Dammermann and Merdes, 2002; Odabasi et al., 2020), and in cells depleted of PCM1, mNG-CSPP-L relocalized to cytoplasmic MTs (Fig. S1, F and G). This indicates that CSPP1 can bind to MTs throughout the cell if it is not sequestered in centriolar satellites. In ciliated RPE-1 cells, mNG-CSPP-L accumulated along the primary cilia as well as to centrosomes and centriolar satellites (Fig. S1 H). Based on fluorescence recovery after photobleaching (FRAP) experiments, mNG-CSPP-L was dynamically associated with basal bodies but showed very little turnover in cilia (Fig. S1, H and I). The latter observation

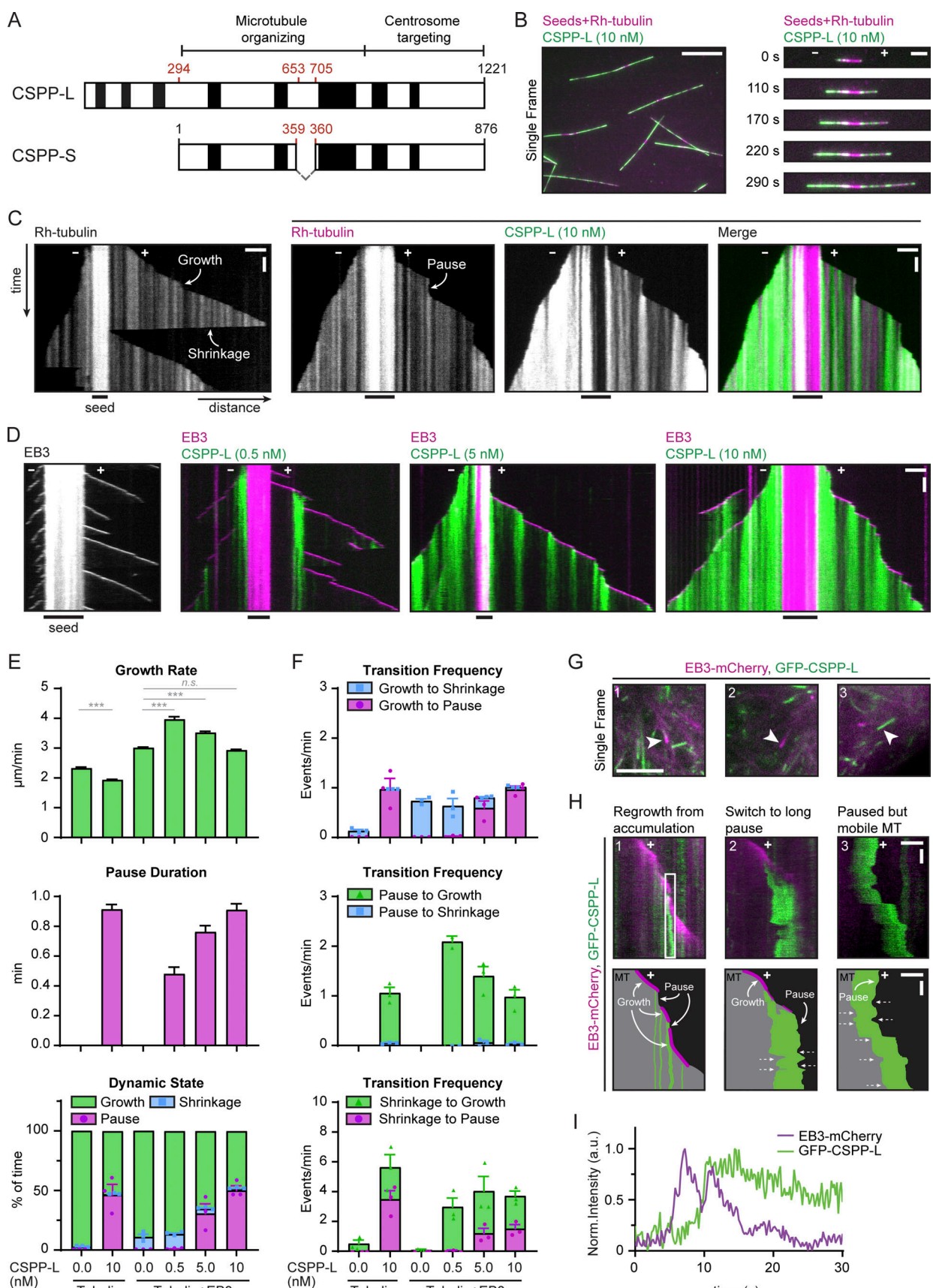

Figure 1. **CSPP1 suppresses catastrophes by binding to polymerizing ends where it induces pausing. (A)** Schematic representation of the two isoforms expressed by the CSPP1 gene in mammals. Black boxes represent α-helical domains larger than 20 amino acids predicted by AlphaFold. **(B)** Field of view (left,

scale bar 10 µm) and time-lapse images (right, scale bar 3 µm) illustrating MT growth from GMPCPP-stabilized MT seeds in the presence of 15 µM tubulin supplemented with 3% rhodamine-labeled tubulin and 10 nM GFP-CSPP-L. MT polarity is indicated. **(C and D)** Kymographs illustrating MT growth either with rhodamine-tubulin (C) or mCherry-EB3 (D), supplemented, where indicated, with the indicated concentrations of GFP-CSPP-L. Time and distance axes are indicated with black arrows; growth, shrinkage, and pause events are indicated with bent, white arrows. Scale bars, 2 µm (horizontal) and 60 s (vertical). **(E and F)** Parameters of MT plus end dynamics in the presence of rhodamine-tubulin alone or together with 20 nM mCherry-EB3 in combination with the indicated GFP-CSPP-L concentrations (from kymographs as shown in C and D). Events were classified as pauses when the pause duration was longer than 20 s. Total number of growth events, pauses, and MTs analyzed (E); tubulin alone, $n$ = 394, 0, 110; tubulin with 10 nM CSPP-L, $n$ = 596, 481, 78; EB3 alone, $n$ = 514, 0, 53; EB3 with 0.5 nM CSPP-L, $n$ = 476, 10, 44; EB3 with 5 nM CSPP-L, $n$ = 564, 241, 47; EB3 with 10 nM CSPP-L, $n$ = 731, 518, 89. Total number of transition events analyzed (F): tubulin alone, $n$ = 194, 0, 0, 0, 15, 0; tubulin with 10 nM CSPP-L, $n$ = 0, 443, 410, 25, 7, 17; EB3 alone, $n$ = 461, 0, 0, 0, 4, 0; EB3 with 0.5 nM CSPP-L, $n$ = 309, 8, 10, 0, 216, 2; EB3 with 5 nM CSPP-L, $n$ = 75, 209, 224, 9, 57, 27; EB3 with 10 nM CSPP-L, $n$ = 24, 465, 455, 22, 25, 21. Bars for growth rate and pause duration represent pooled data from three independent experiments. For dynamic state and transition frequencies, bars represent the average of the means (symbols) of three independent experiments. Error bars represent SEM ***, P < 0.001; n.s., not significant; Kruskal–Wallis test followed by Dunn's post-test. **(G)** Single frame images of a COS-7 cell overexpressing GFP-CSPP-L and EB3-mCherry, imaged by TIRF microscopy. Scale bars, 5 µm. **(H)** Kymographs (top) and schematic representation of these kymographs (bottom) of the events indicated with white arrowheads in G. In the schemes, unlabeled MT is visualized in grey, mCherry-EB3 in magenta, and GFP-CSPP-L in green. Bent white arrows indicate growth and pause events; dashed white arrows indicate the direction of movement of the whole MT at that time point. Scale bars, 2 µm (horizontal) and 4 s (vertical). **(I)** Normalized intensity graphs of EB3-mCherry and GFP-CSPP-L within the white box in H. See also Fig. S1 and Videos 1 and 2.

could be consistent with the low mobility of GFP-CSPP-L on MTs in vitro. Also, in non-ciliated cells, such as COS-7 cells, endogenous CSPP1 was localized to centrosomes and PCM1-positive centriolar satellites but not to cytoplasmic MTs, similar to its localization in interphase RPE-1 cells (Fig. S1 J).

We followed up on these observations by mildly overexpressing GFP-CSPP-L in COS-7 cells and found that it formed accumulations along MTs, similar to what we observed in vitro (Fig. 1 G, Fig. S1 K, and Video 2). Elevated levels of CSPP-L led to an increase in MT acetylation (Fig. S1, K and L), a hallmark of MT stabilization (Magiera et al., 2018). Moreover, the number of MT plus ends labeled with EB1, a marker of growing MT ends (Mimori-Kiyosue et al., 2000), was strongly reduced (Fig. S1, K and M), indicating that MT dynamics were suppressed. Live cell imaging in COS-7 cells co-expressing GFP-CSPP-L and EB3-mCherry showed that CSPP-L bound to growing, EB3-positive MT ends concomitantly with EB3 signal reduction, and CSPP-L accumulation led to MT pausing (Fig. 1, G–I). MTs could regrow from CSPP-L accumulations (Fig. 1, G and H, image and kymograph 1) or stay paused for longer periods of time (Fig. 1, G and H, image and kymograph 2). Many pausing MT ends strongly labeled with CSPP-L, which were undergoing short-range back-and-forth displacements, were observed throughout the cell (Fig. 1, G and H, image and kymograph 3; and Video 2). We conclude that CSPP-L binds to growing MT ends, prevents their shrinkage, and induces pausing both in vitro and in cells.

### CSPP1 binds to precatastrophe MT ends, resembling taxane behavior

The formation of confined accumulation zones that initiate at growing MT ends and prevent MT shrinkage makes the dynamic behavior of CSPP-L strikingly similar to that we have recently described for taxanes (Rai et al., 2020). To determine if CSPP-L and taxanes recognize the same features of MTs, we have tested whether fluorescently labeled taxane Fchitax-3 colocalized with CSPP-L, and we found that this was indeed the case (Fig. 2, A and B). Over time, the intensity of both Fchitax-3 and CSPP-L first increased and then decreased in a similar way (Fig. 2, A and C). Measurements of fluorescence intensity of 10 nM CSPP-L within

accumulation zones, performed as described previously (Rai et al., 2020), indicated that on average, one CSPP-L molecule was bound per 8 nm of MT length (corresponding to the length of one layer of α/β-tubulin dimers; Fig. 2 D), indicating that the binding sites are likely not saturated in these conditions.

Since our previous work has demonstrated that binding of Fchitax-3 is triggered by perturbed MT growth and occurs when MTs enter a precatastrophe state manifested by the loss of GTP cap and reduced EB3 binding (Rai et al., 2020), we tested whether the same is true for CSPP-L. Indeed, periods of strong CSPP-L accumulation always initiated shortly before or concomitantly with the reduction of the EB3 signal (Fig. 2, E–G), very similar to the CSPP-L accumulation pattern observed in cells (Fig. 1, G–I). To support our interpretation that perturbed MT growth triggers CSPP-L binding, we supplemented the assay with 100 nM vinblastine, which promotes frequent catastrophes at low concentrations in the presence of EB3 (Mohan et al., 2013). Catastrophes indeed became much more frequent, and this resulted in the increased number of CSPP-L accumulation zones, leading to a higher overall binding of the protein along MTs (Fig. 2, H–K). Similar to the conditions without vinblastine, 0.5 nM CSPP-L did not block depolymerization completely but induced the formation of rescue sites, whereas 5 nM CSPP-L induced more frequent pausing episodes followed by regrowth (Fig. 2, H–K). To further prove that an increase in catastrophe frequency promotes CSPP-L binding, we combined CSPP-L with EB3 and the kinesin-13 MCAK, an MT depolymerase that triggers frequent catastrophes in the presence of EB3 (Montenegro Gouveia et al., 2010). In these conditions, we indeed observed enhancement of CSPP-L accumulation along MTs (Fig. 2 L), again very similar to our previous observations with Fchitax-3 (Rai et al., 2020). We conclude that similar to taxanes, CSPP-L strongly accumulates at MT ends that undergo a growth perturbation, inhibits both their growth and shortening, and gradually dissociates when MT growth resumes.

### Separate CSPP1 domains control the balance between MT polymerization and depolymerization

Next, we examined which CSPP1 domains are responsible for their effects on MT dynamics. Structure predictions made by a recently developed neural network AlphaFold (Jumper et al.,

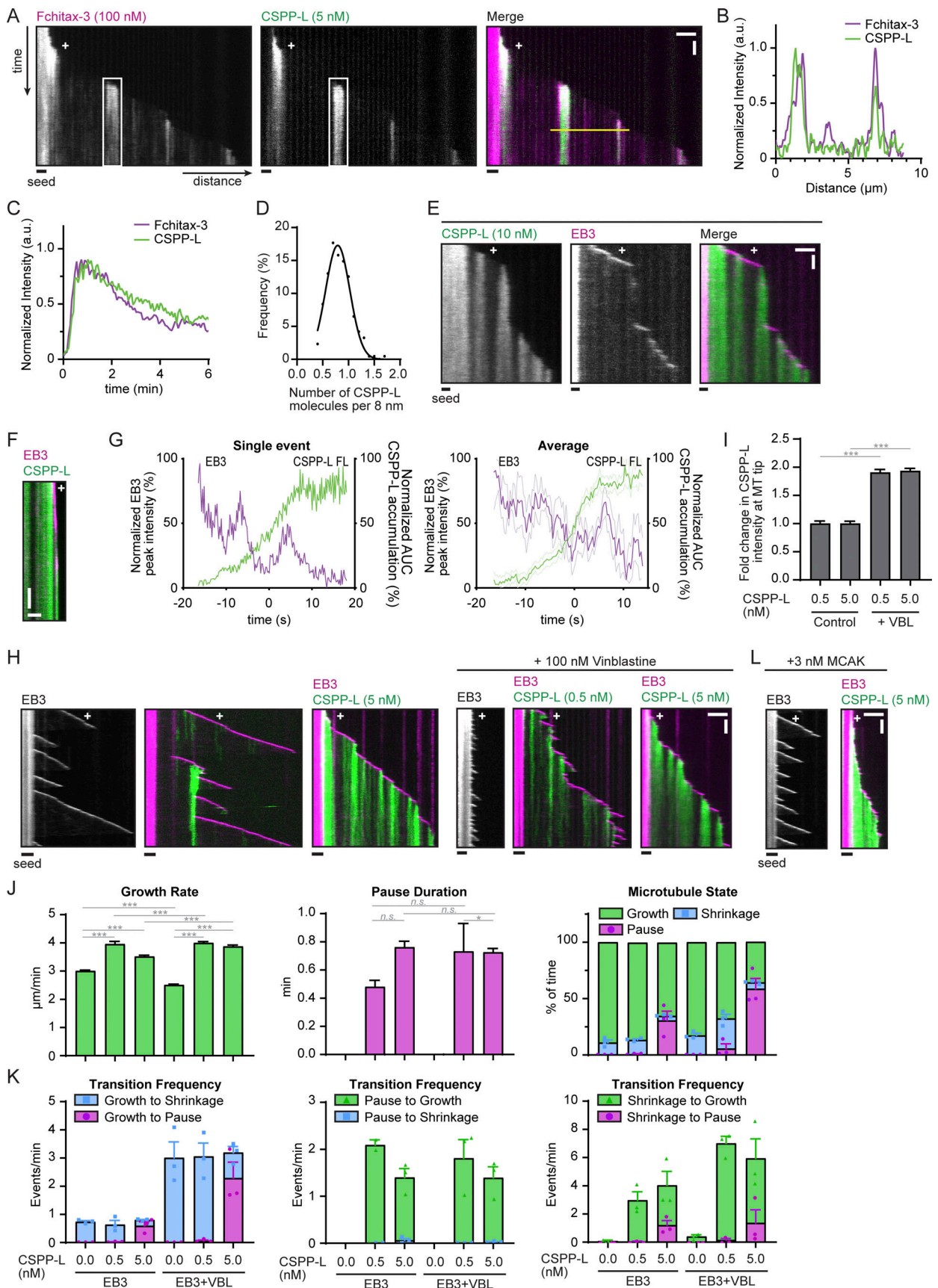

Figure 2. **CSPP1 binds to precatastrophe MT ends, resembling taxane behavior. (A)** Kymographs of MT growth with 100 nM Fchitax-3 together with 5 nM mCherry-CSPP-L in the presence of 20 nM dark EB3. Scale bars, 2 μm (horizontal) and 60 s (vertical). **(B)** Normalized intensity graph of Fchitax-3 and mCherry-

CSPP-L along the yellow line in A. **(C)** Normalized intensity graph of Fchitax-3 and mCherry-CSPP-L within the white box in A. **(D)** Quantification of the number of GFP-CSPP-L molecules per 8 nm MTs. The integrated intensity of one GFP-CSPP-L accumulation in an in vitro assay was divided by the average intensity of single GFP monomers in a separate chamber on the same coverslip and subsequently normalized to 8 nm accumulation length. The number of GFP-CSPP-L accumulations, $n$ = 215 from two independent experiments. **(E and F)** Kymographs illustrating MT growth in the presence of 20 nM mCherry-EB3 together with 10 nM GFP-CSPP-L. Scale bars, 2 μm (horizontal) and 2 min (vertical, E) or s (vertical, F). **(G)** Time plot of the normalized maximum intensity profile of a single mCherry-EB3 comet and the normalized area under the curve (AUC) of a single GFP-CSPP-L accumulation (left) and averaged EB3 and GFP-CSPP-L profiles, normalized and aligned using half-maximum effective intensity values from Hill equation fits as reference points (right; from kymographs as shown in F). Light, thin lines represent SEM. Number of events analyzed, $n$ = 12 from two independent experiments. **(H)** Kymographs illustrating MT growth in the presence of 20 nM mCherry-EB3 alone or together with the indicated concentrations of GFP-CSPP-L in the presence or absence of 100 nM vinblastine (VBL). Scale bars, 2 μm (horizontal) and 60 s (vertical). **(I)** Quantification of the mean GFP-CSPP-L intensity at the MT tip per growth event. The average mean intensity of GFP-CSPP-L in the presence of 100 nM vinblastine was normalized to the average mean intensity in absence of vinblastine. Total number of growth events analyzed; 0.5 nM GFP-CSPP-L control, $n$ = 474; 5 nM GFP-CSPP-L control, $n$ = 598; 0.5 nM GFP-CSPP-L with vinblastine, $n$ = 1,363; 5 nM GFP-CSPP-L with vinblastine, $n$ = 897. Bars represent pooled data from three independent experiments. **(J and K)** Parameters of MT plus end dynamics in the presence of 20 nM mCherry-EB3 together with the indicated GFP-CSPP-L concentrations (from kymographs as shown in G). Events were classified as pauses when the pause duration was longer than 20 s. Total number of growth events, pauses and MTs analyzed (J); EB3 alone, $n$ = 514, 0, 53; EB3 with 0.5 nM CSPP-L, EB3 with 0.5 nM CSPP-L, $n$ = 476, 10, 44; EB3 with 5 nM CSPP-L, $n$ = 564, 241, 47; EB3 with vinblastine, $n$ = 915, 0, 54; EB3 with 0.5 nM CSPP-L and vinblastine, $n$ = 1,204, 33, 40; EB3 with 5 nM CSPP-L and vinblastine, $n$ = 632, 408, 47. Total number of transition events analyzed (K): EB3 alone, $n$ = 461, 0, 0, 4, 0; EB3 with 0.5 nM CSPP-L, $n$ = 309, 8, 10, 0, 216, 2; EB3 with 5 nM CSPP-L, $n$ = 75, 209, 224, 9, 57, 27; EB3 with vinblastine, $n$ = 162, 0, 0, 0, 33, 0; EB3 with 0.5 nM CSPP-L and vinblastine, $n$ = 1,079, 19, 31, 0, 1,002, 14; EB3 with 5 nM CSPP-L and vinblastine, $n$ = 147, 372, 386, 7, 127, 27. Bars for growth rate and pause duration represent pooled data from three independent experiments. For dynamic state and transition frequencies, bars represent the average of the means (symbols) of three independent experiments. Data for conditions without vinblastine is the same as in Fig. 1, E and F. **(L)** Kymographs illustrating MT growth in the presence of 20 nM mCherry-EB3 alone or together with 5 nM GFP-CSPP-L in presence of 3 nM MCAK. Scale bars, 2 μm (horizontal) and 60 s (vertical). For all plots. Error bars represent SEM. ***, P < 0.001; *, P < 0.1; n.s., Kruskal–Wallis test followed by Dunn's post-test.

2021; Varadi et al., 2022) indicated that CSPP-L contains several putative α-helical domains (H1-8) interspersed with regions of unknown structure (L1-L7; Fig. 3 A). Compared with the previously published analyses, this prediction suggested the presence of two additional α-helical regions, H4 and H8, in the middle and C-terminal part of CSPP1. Based on the predicted domains, we generated various fragments of CSPP1 N-terminally tagged with GFP (Fig. 3 A) and tested them in the in vitro assays. The short isoform of CSPP1, CSPP-S, behaved similarly to CSPP-L, though at 10 nM it was less efficient at preventing MT depolymerization and could also occasionally block MT outgrowth from the seed, whereas we have never observed this effect with CSPP-L (Fig. 3 A and Fig. S2 A). Next, we focused on the middle part of CSPP1, previously identified as the MT-organizing region (Frikstad et al., 2019; Patzke et al., 2006; MTORG, Fig. 3 A; and Fig. S2, B and C). The MTORG region derived from the CSPP-L isoform displayed local accumulations along MTs and prevented catastrophes at 10 nM but did not cause long pauses, even at 40 nM concentration (Fig. S2 B). The MTORG version with the internal deletion present in CSPP-S showed little MT binding at 10 nM, but the binding became visible at 40 nM and was accompanied by frequent pauses, followed by either growth or shrinkage (Fig. 3 A and Fig. S2 C). Further, deletion mapping at the C-terminus of the MTORG domain (the construct H4+L4+H5) showed that the helical domain H6 with the preceding linker L5 was not essential for MT binding or rescue activity but was needed to trigger pausing (Fig. 3 A and Fig. S2 D). An even shorter truncation mutant, which also lacked helical domain H5 (H4+L4), displayed only a very weak binding to MTs (Fig. 3 A and Fig. S2 E). However, the affinity of this fragment for MTs was increased by linking it to the leucine zipper dimerization domain of GCN4 (H4+L4+LZ; Fig. 3 A and Fig. S2 F).

Restoration of MT affinity by introducing an artificial dimerization domain suggested a potential oligomerization function for H5. To investigate this further, we first set out to determine the oligomeric state of CSPP-L. We compared the intensity of CSPP-L molecules upon initial binding to MTs to the intensity of single GFP molecules on the same coverslip in another chamber and found that CSPP-L intensity was close to single GFP intensity (Fig. 3 B), indicating that CSPP-L binds to MTs as a monomer. The measured intensity value was somewhat lower than the intensity of a single GFP (∼0.8) because the measurement was performed by TIRF microscopy and was therefore very sensitive to the position of the fluorophore along the z-axis. The CSPP-L molecules on MTs are further away from the coverslip compared with GFP molecules absorbed on glass, and therefore their signal is somewhat lower, as we have described previously for an MT-bound kinesin (van Riel et al., 2017). Surprisingly, the CSPP-L signal increased to an average of two molecules within 25 s after initial binding, suggesting that CSPP-L binds cooperatively and potentially oligomerizes on MTs (Fig. 3 C). If H5 would facilitate di- or oligomerization of CSPP1, similar results would be expected for H4+L4+H5, while no increase in the signal would be expected for H4+L4. Unfortunately, H4+L4 diffused a lot on the MT lattice, precluding precise quantification of the signal over time (Fig. 3 D). Diffusive behavior was less pronounced for H4+L4+H5, indicating that H5 decreases lateral mobility (Fig. 3 D), making the construct more similar to the full-length CSPP-L, which shows hardly any lateral mobility. Thus, the helical domain H5 inhibits lateral mobility, possibly by facilitating di- or oligomerization of CSPP-L on MTs.

As the addition of the leucine zipper restored MT affinity, we linked the newly identified short α-helical domain H4 directly to the leucine zipper through a short flexible linker and this yielded a construct that weakly bound to MTs but did not induce rescues, even at concentrations up to 300 nM (H4+LZ; Fig. 3, A and E–G and Fig. S2 G). Extension of H4 with a part of linker L4 (amino acids 375–453, a protein fragment we termed MTB for "MT-binding"), fused to the leucine zipper, resulted in a construct that was sufficient for MT binding and rescue induction

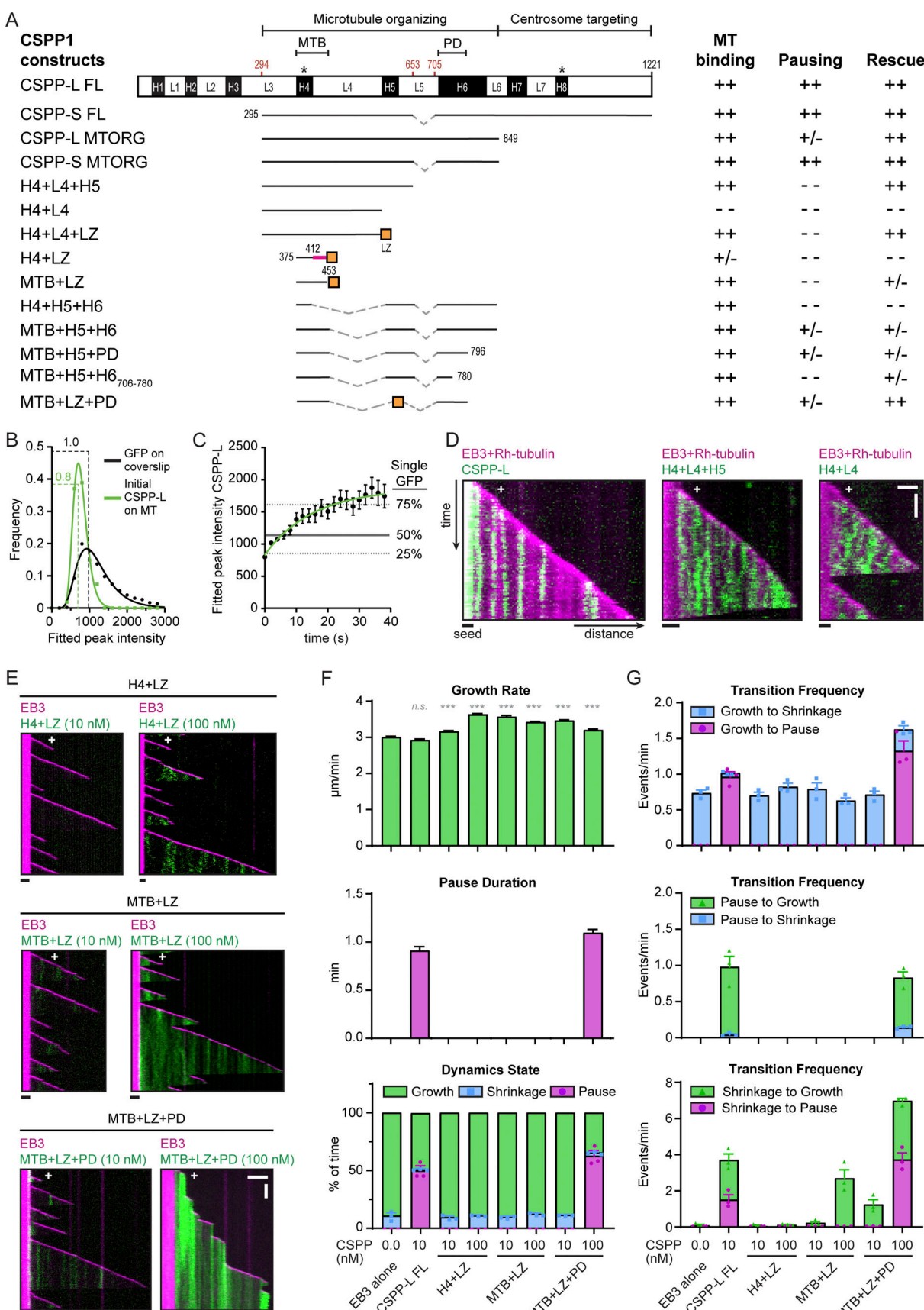

Figure 3. **Separate CSPP1 domains control the balance between MT polymerization and depolymerization. (A)** Schematic representation of the different CSPP1 constructs used and a summary of their MT binding ability and their effects on MT dynamics. Black boxes represent α-helical domains larger than

20 amino acids predicted by AlphaFold; asterisks indicate previously unidentified helices. ++: frequently observed at protein below 40 nM; +/−: occasionally observed at protein concentrations below 40 nM and/or frequently observed at protein concentrations up to 100 nM; - -: observed infrequently or not observed at all even at protein concentrations higher than 100 nM. MTB, MT binding domain; PD, pausing domain. **(B)** Histograms of fluorescence intensities of single GFP molecules immobilized on coverslips and GFP-CSPP-L molecules upon initial binding to the MT lattice in a separate chamber on the same coverslip (symbols) and the corresponding fits with lognormal distributions (lines). Number of molecules in analysis: single GFP, $n$ = 16,795; GFP-CSPP-L, $n$ = 54 (for the latter, initial binding events were manually selected for analysis) from one representative experiment. Dashed lines show corresponding modal values. **(C)** Quantification of the fluorescence intensity of GFP-CSPP-L molecules over time after initial binding. Green line shows one-phase association fit. Grey horizontal lines correspond to the quartile values of fluorescence intensities of single GFP in the histogram shown in B. Number of molecules in analysis; GFP-CSPP-L, $n$ = 54 from one representative experiment. Error bars represent SEM. **(D)** Kymographs illustrating the behavior of the indicated GFP-CSPP fragments on the MT lattice in presence of 20 nM mCherry-EB3 and 15 µM rhodamine-tubulin. Scale bars, 2 µm (horizontal) and 30 s (vertical). **(E)** Kymographs illustrating MT growth with 20 nM mCherry-EB3 alone or together with 10 or 100 nM of the indicated GFP-CSPP1 constructs. Scale bars, 2 µm (horizontal) and 60 s (vertical). **(F and G)** Parameters of MT plus end dynamics in the presence of 20 nM mCherry-EB3 together with 10 or 100 nM of the indicated GFP-CSPP1 constructs (from kymographs as shown in B). Events were classified as pauses when the pause duration was longer than 20 s. Total number of growth events, pauses, and MTs analyzed (C); EB3 alone, $n$ = 514, 0, 53; EB3 with 10 nM CSPP-L, $n$ = 731, 518, 89; EB3 with 10 nM H4+LZ, $n$ = 855, 0, 87; EB3 with 100 nM H4+LZ, $n$ = 987, 0, 103; EB3 with 10 nM MTB+LZ, $n$ = 1,006, 0, 109; EB3 with 100 nM MTB+LZ, $n$ = 1,206, 0, 139; EB3 with 10 nM MTB+LZ+PD, $n$ = 934, 0, 104; EB3 with 100 nM MTB+LZ+PD, $n$ = 776, 707, 123. Total number of transition events analyzed (D): EB3 alone, $n$ = 461, 0, 0, 0, 4, 0; EB3 with 10 nM CSPP-L, $n$ = 24, 465, 455, 22, 25, 21; EB3 with 10 nM H4+LZ, $n$ = 751, 0, 0, 0, 3, 0; EB3 with 100 nM H4+LZ, $n$ = 889, 0, 0, 0, 15, 0; EB3 with 10 nM MTB+LZ, $n$ = 902, 0, 0, 0, 26, 0; EB3 with 100 nM MTB+LZ, $n$ = 1,035, 0, 0, 0, 582, 0; EB3 with 10 nM MTB+LZ+PD, $n$ = 797, 0, 0, 0, 191, 0; EB3 with 100 nM MTB+LZ+PD, $n$ = 126, 545, 520, 105, 107, 121. Bars for growth rate and pause duration represent pooled data from three independent experiments. For dynamic state and transition frequencies, bars represent the average of the means (symbols) of three independent experiments. Error bars represent SEM. ***, $P < 0.001$; n.s., not significant; Kruskal–Wallis test followed by Dunn's post-test. Data for EB3 alone and EB3 with 10 nM CSPP-L is the same as in Fig. 1, E and F. See also Fig. S2.

(MTB+LZ; Fig. 3, A and E–G and Fig. S2 H). MT binding of CSPP1 thus depends on a short region, which is predicted to be α-helical and is augmented by additional regions distributed throughout the CSPP1 molecule, including the region missing in the CSPP-S isoform.

Importantly, all CSPP1 fragments lacking the domain H6 did not cause MT pausing, suggesting that H6 could be responsible for pause induction. To test this idea, we first directly fused the H5 and H6 domains to H4 (H4+H5+H6). Already at 40 nM concentration, this construct strongly inhibited MT outgrowth from the seed and induced catastrophes (Fig. 3 A and Fig. S2 I). Attaching the H5 and H6 domains to MTB (MTB+H5+H6) resulted in a construct that could induce MT pausing and inhibit depolymerization at 100 nM, whereas at lower concentrations (40 nM), it showed occasional rescues but no pauses (Fig. 3 A and Fig. S2 J). To determine which part of H6 is responsible for inhibiting MT growth, we truncated it at the C-terminus and found that MTB+H5+H6$_{706-796}$, but not a shorter version, MTB+H5+H6$_{706-780}$, still triggered pausing and inhibited MT shrinkage when fused to MTB and H5 (Fig. 3 A and Fig. S2, K and L). We, therefore, termed H6$_{706-796}$ the pausing domain (PD). Swapping H5 within this construct for the leucine zipper (MTB+LZ+PD) yielded a construct with similar properties (Fig. 3, A, E–G and Fig. S2 M), supporting the idea that H5 acts as a di- or oligomerization domain. Thus, the MTB+LZ+PD construct recapitulates the major effects of full-length CSPP-L on MT dynamics. This construct lacks the C-terminal domain of CSPP1 responsible for localization to centrosomes and centriolar satellites (Patzke et al., 2006), indicating that the minor CSPP1 contamination with PCM1 is not responsible for the observed in vitro activities of the protein. However, the concentration required to achieve these effects is 10-fold higher than for the full-length construct, suggesting that the domains missing in this construct increase MT affinity and/or provide the geometry needed for the stabilizing and growth-inhibiting activities.

Next, we compared the impact of truncated CSPP1 constructs with that of GFP-CSPP-L on MTs in COS-7 cells. GFP-MTB+LZ+PD, GFP-MTB+LZ, and GFP-H4+LZ localized to MTs in interphase cells. However, compared with CSPP-L, the shorter constructs were less potent in inducing MT acetylation and reducing the number of EB1 comets, indicating that they are less efficient in stabilizing MTs (Fig. S2, N–Q; control in Fig. S1, K–M). Altogether, we conclude that CSPP1 has multiple regions contributing to MT binding, but the minimal construct that reproduces the major effects of CSPP-L on MT dynamics is MTB+LZ+PD. These effects appear to depend on the interplay between two separate activities, residing in two predicted helical regions: MT binding and stabilization by the MTB and the growth-inhibiting activity of the truncated α-helical domain H6, the PD.

## CSPP1 binds to the MT lumen

As described above, the behavior and effect of CSPP1 on dynamic MTs resembles that of taxanes. Taxanes are known to bind to the MT lumen (reviewed in Steinmetz and Prota, 2018), and therefore, we set up cryo-ET experiments to investigate whether CSPP1 is an intraluminal protein. Using a previously established experimental design (Ogunmolu et al., 2021 Preprint), we polymerized dynamic MTs from GMPCPP-stabilized seeds in the presence or absence of 10 nM CSPP-L, with or without 250 nM vinblastine, and vitrified them on EM grids. To increase the signal-to-noise ratio in the reconstructed tomograms, we used the cryoCARE denoising method (Buchholz et al., 2019). MTs polymerized in presence of CSPP-L frequently contained luminal densities, which were absent in CSPP-L-free samples (Fig. 4, A and B and Fig. S3 A). The presence of vinblastine resulted in a higher percentage of MTs containing ILPs: 68 ± 22% (mean ± SD) compared with 35 ± 11% in absence of vinblastine (P < 10⁻⁴, Fig. 4, A and B). We did not observe CSPP-L densities outside of the MT lumen.

We further used automated segmentation of denoised tomograms (Chen et al., 2017) to get a better understanding of the ILPs. CSPP-L particles appeared quite disordered and could either block the MT lumen completely or only partially (Fig. 4 C

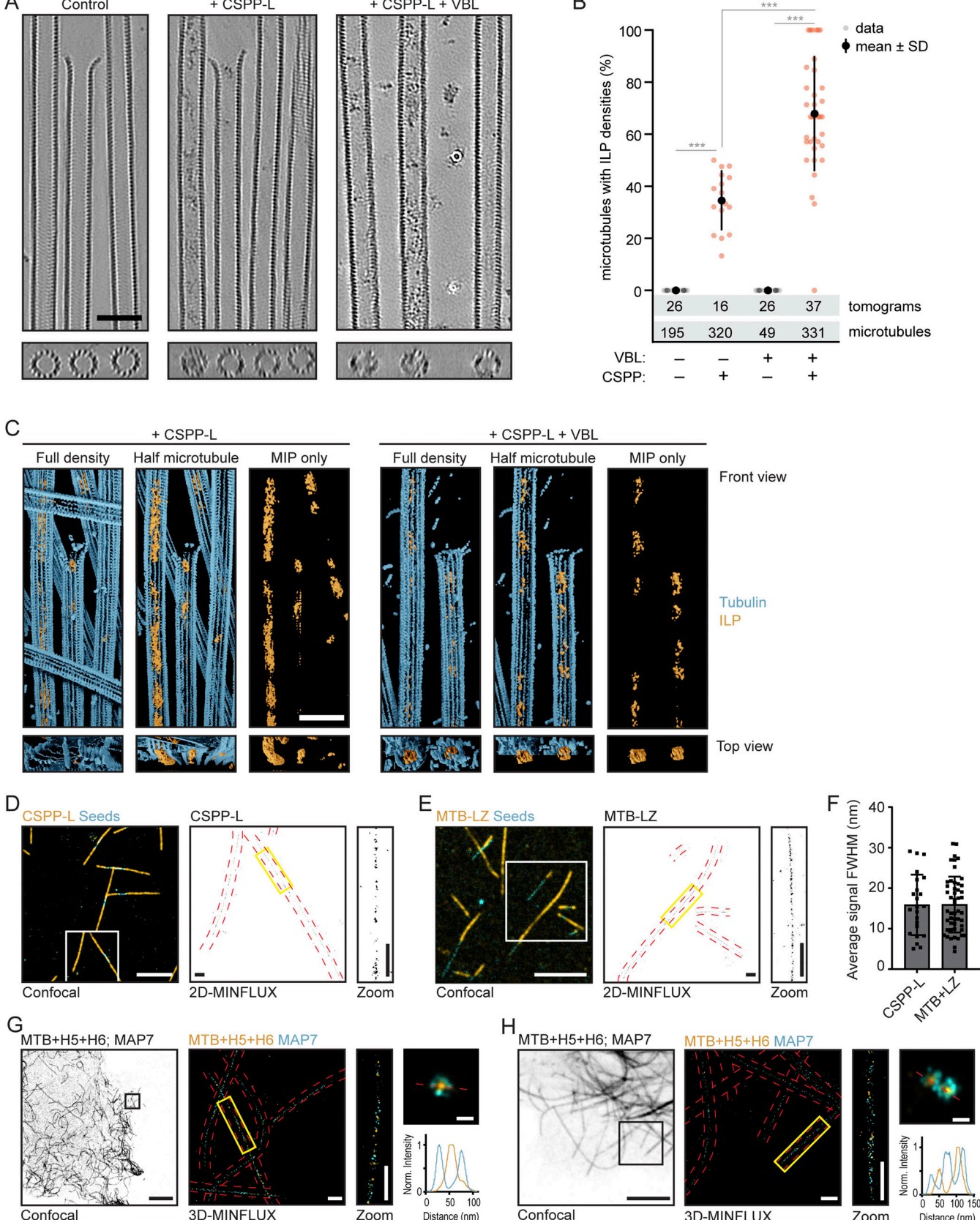

Figure 4. **CSPP1 binds to MT lumen. (A)** Denoised tomograms of dynamic MTs polymerized from GMPCPP-stabilized seeds in the presence or absence of 10 nM GFP-CSPP-L, with or without 250 nM vinblastine vitrified on EM grids. Scale bar, 50 nm. **(B)** Quantification of the percentage of MTs containing luminal densities from total MTs (from tomograms as shown in A). Orange and gray dots (single data points, tomograms), black circle (mean), SD (error bars). Number

of MTs and tomograms are displayed in the graph. ***, P < 0.001, Mann-Whitney test. Analysis from two independent experiments. **(C)** Reconstituted images from automated segmentation of denoised tomograms as in A. Scale bar, 50 nm. **(D and E)** Single color 2D-MINFLUX measurements of in vitro reconstituted MTs polymerized in the presence of SNAP-CSPP-L (D) or SNAP-MTB-LZ (E). Images were rendered with 1 nm voxel size for visualization. White boxes in confocal image indicate the region shown in the rendered 2D-MINFLUX image, yellow boxes in the 2D-MINFLUX image indicate the region of the zoom, red dashed lines represent the MT outline from the confocal image. Scale bars, 5 μm (confocal image); 500 nm (2D-MINFLUX image and zoom). **(F)** Quantification of the fitted, FWHM values per MT (from 2D-MINFLUX images as shown in D and E; see Fig. S3 C analysis details). Single data points are shown. Error bars represent SD. Number of measured MTs; CSPP-L, n = 23; MTB+LZ, n = 50. Analysis from three independent experiments. **(G and H)** Two examples of dual color 3D-MINFLUX measurements of COS-7 cells overexpressing GFP-MAP7 plus SNAP-MTB-H5-H6. Images were rendered with a 4-nm voxel size for visualization. Black boxes in the confocal image indicate the region shown in the rendered 3D-MINFLUX image, yellow boxes in the 3D-MINFLUX image indicate the region of the zoom, red dashed lines represent the MT outline from the confocal image. Top right image shows a maximum intensity projection of the cross-section of the MT over 800 nm. The red dashed line there indicates the line scan related to the bottom right graph. Scale bars, 10 μm (confocal image); 500 nm (3D-MINFLUX image and zoom), 50 nm (maximum intensity projection image). See also Fig. S3 and Videos 3 and 4.

and Video 3). They were occupying a variable length of the MT lumen, preventing further analysis of their structure. Some CSPP-L particles were bound close to the terminal flare of tubulin protofilaments, but we never observed them binding to tapered MT ends or other incomplete MT lattices.

Next, we aimed to confirm that the densities inside MTs we observed with cryo-ET indeed represent CSPP-L and determine the localization of shorter CSPP1 fragments. Since the latter would be difficult to achieve by cryo-ET due to the small protein size, we turned to MINFLUX microscopy, which allows localization of individual fluorophores with very high spatial resolution, as was demonstrated by the separation of, e.g., two fluorophores as close as 6 nm from each other (Balzarotti et al., 2017; Gwosch et al., 2020). The localization resolution of MINFLUX would allow us to determine whether the CSPP1 fragments localize inside or outside 25-nm-wide MT filaments.

For 2D MINFLUX measurements, we used fixed MTs that were grown in vitro in the presence of SNAP-tagged CSPP1 or its fragments. We first performed measurements for CSPP-L, the same protein we used for cryo-ET. From these measurements, we generated the intensity profile plot of each MT filament and extracted the full-width at half-maximum (FWHM) value as an estimation of the signal width. For CSPP-L, the width of the signal was 15.87 ± 7.47 nm (mean ± SD; Fig. 4, D and F; and Fig. S3, B and C). In theory, these values would be similar for proteins localizing to the MT seam, but together with our previous cryo-ET results, we believe these values align with a luminal localization. This is supported by the determined localization precision in x and y of the MINFLUX measurements, as these values were 3.7 and 3.2 nm, respectively (Fig. S3 D). The smallest CSPP1 fragment binding to MTs, H4+LZ, gave too much background signal to allow meaningful measurements. However, the smallest CSPP1 construct affecting MT dynamics in vitro, MTB+LZ (Fig. 3, A and E–G), gave a signal width of 16.35 ± 6.80 nm (mean ± SD), again aligning with a luminal localization (Fig. 4, E and F; and Fig. S3, B and C).

To validate in cells the results described above, we overexpressed SNAP-tagged CSPP1 fragments with the GFP-labeled N-terminal, MT-binding part of MAP7, a protein known to bind to MT exterior (Ferro et al., 2022). We overexpressed these constructs in COS-7 cells, fixed them, and stained them for SNAP and GFP. Unfortunately, we were not able to get high-quality data for full-length GFP-CSPP-L due to low expression levels of the protein, ultimately resulting in very sparse labeling

incompatible with MINFLUX measurements. However, the experiment was successful with the smaller SNAP-MTB+H5+H6 fragment, and the maximum intensity projection over the cross-section of the MT upon its overexpression together with GFP-MAP7 showed a ring of MAP7 signal surrounding the CSPP1 fragment (Fig. 4, G and H; Fig. S3 E; and Video 4). Acquisition of MINFLUX images for even shorter CSPP1 fragments was impeded by the high cytosolic background due to the presence of a significant pool of MT-unbound proteins. Taken together, the data obtained in vitro and in cells support the intraluminal localization of CSPP1 and indicate that the short MTB domain is sufficient for this localization.

### CSPP1 efficiently binds to sites where MT lattices are damaged

Since CSPP1 is an intraluminal protein, we next examined whether it can bind to sites of lattice damage, which would provide access to the MT lumen. First, we compared the binding of CSPP-L to rather stable GMPCPP-bound MTs and to Taxol-stabilized MTs, which are known to acquire extensive lattice defects when incubated in the absence of soluble tubulin and free Taxol (Aher et al., 2020; Arnal and Wade, 1995). In the absence of soluble tubulin, CSPP-L gradually accumulated at discrete sites on both types of MTs, but the binding to Taxol-stabilized MTs was faster, and CSPP-L signal intensity was higher (Fig. 5, A and B). Next, we induced local damage of GMPCPP-stabilized MTs using illumination with a pulsed 532-nm laser, as described previously (Aher et al., 2020). We chose MT regions where no prior CSPP-L signal was present and selected for analysis only those MTs that were not fully severed during laser illumination. We observed a strong accumulation of CSPP-L at the illuminated sites, whereas the CSPP-L signal was relatively stable within the same time period at the sites that were not damaged by the laser (Fig. 5, C–E). We then performed a similar experiment using dynamic MT lattices in the presence of soluble tubulin and observed that tubulin was recruited to the damage sites, suggesting that the photodamaged MT was repaired (Fig. 5, F–H). Moreover, when MTs were fully severed, the newly generated plus ends rarely switched to depolymerization, whereas depolymerization occurred frequently when CSPP1 was absent, as expected for severed MT plus ends that lack a GTP cap (Walker et al., 1989; Fig. 5, I and J). These results suggest that CSPP1 recognizes damaged MTs, stabilizes them by preventing depolymerization, and thus possibly facilitates MT repair.

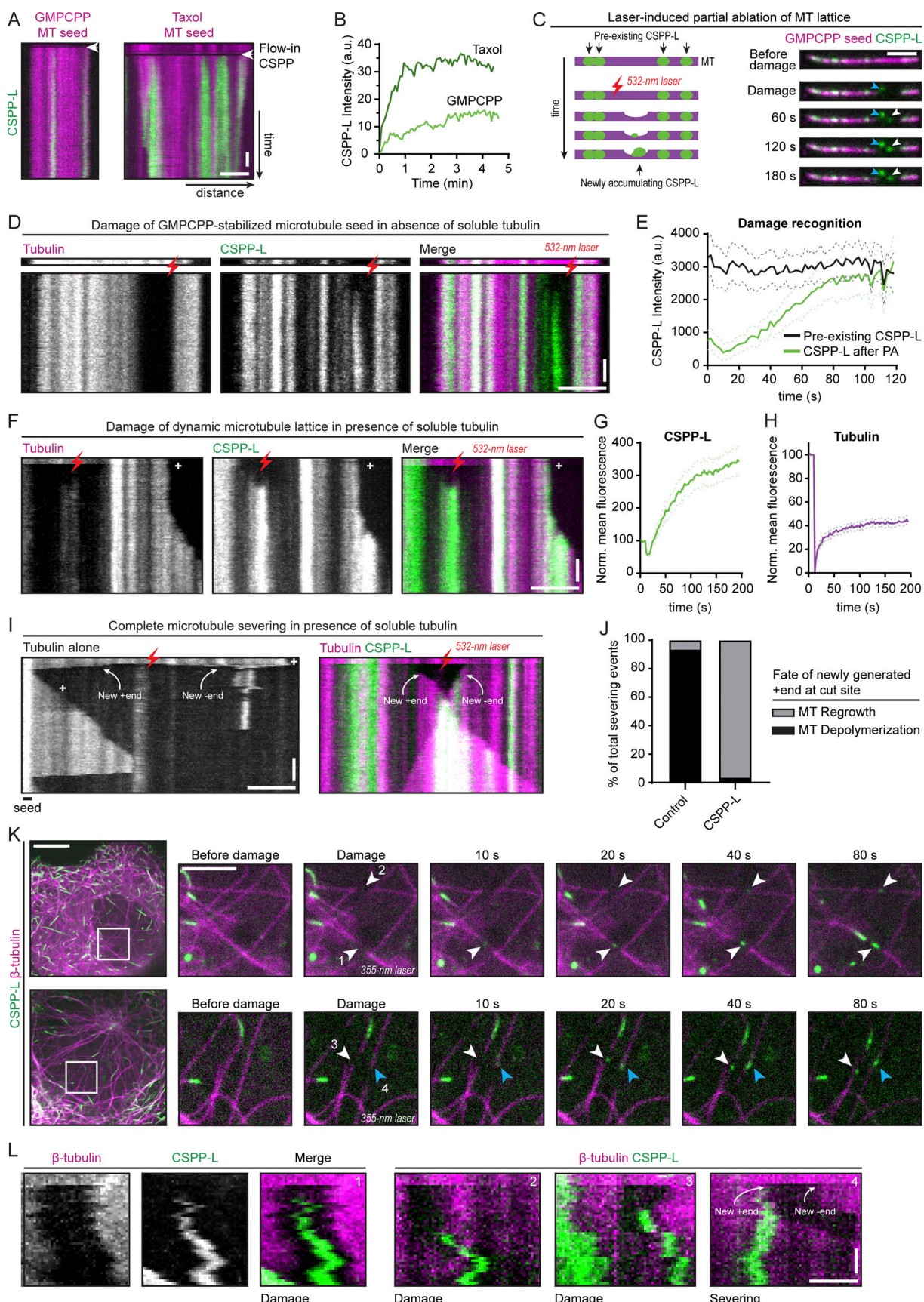

Figure 5. **CSPP1 binds to sites where MT lattices are damaged. (A)** Kymographs of GMPCPP- (left) and Taxol-stabilized (right) MT seeds. 5 nM GFP-CSPP-L was flushed in during acquisition in absence of free taxol or tubulin. Scale bars, 2 µm (horizontal) and 30 s (vertical). **(B)** GFP-CSPP-L intensity profile of

developing accumulation after flow-in of experiments done in A. **(C)** Schematic representation of laser damage of MT lattice at a region with no prior GFP-CSPP-L accumulation (left) and time-lapse images (right) of laser damage of a GMPCPP-stabilized MT seed in presence of 5 nM GFP-CSPP-L in absence of soluble tubulin. The MT region illuminated with the 532-nm pulsed laser is highlighted by a white arrowhead. The blue arrowhead indicates the damage inflicted on the coverslip. Scale bar, 2 µm. **(D)** Kymograph corresponding to time-lapse images shown in C. The laser-illuminated MT region is highlighted by a red lightning bolt. Scale bars, 2 µm (horizontal) and 30 s (vertical). **(E)** Averaged GFP-CSPP-L intensity profiles after photodamage (from kymographs as shown in D). Plots were aligned using half-maximum effective intensity values from nonlinear regression fits as reference points. Dashed lines represent SEM. Number of events analyzed, n = 15 from three independent experiments. **(F)** Kymograph illustrating laser damage of dynamic MT lattice in presence of 5 nM GFP-CSPP-L and 15 µM soluble tubulin. The laser-illuminated MT region is highlighted by a red lightning bolt. Scale bars, 2 µm (horizontal) and 30 s (vertical). **(G and H)** Mean intensity of GFP-CSPP-L (G) or tubulin (H) normalized to the intensity before damage (set at 100) over time at the photodamage site (from kymographs as shown in F). Dashed lines represent SEM. Number of events analyzed, n = 24 from four independent experiments. **(I)** Kymographs illustrating complete severing of dynamic MT lattice in presence of 15 µM soluble tubulin with or without 5 nM GFP-CSPP-L. The laser-illuminated MT region is highlighted by a red lightning bolt, the newly generated ends are indicated by white arrows. Scale bars, 2 µm (horizontal) and 30 s (vertical). **(J)** Plot showing the immediate fate of newly generated plus ends upon complete severing in presence of 15 µM soluble tubulin with or without 5 nM GFP-CSPP-L. Number of fragments analyzed; control, n = 46; CSPP-L, n = 56. Bars represent pooled data from four independent experiments. **(K)** Time lapse images of photodamage experiments in COS-7 cells overexpressing GFP-CSPP-L and β-tubulin-mCherry. Arrowheads indicate the events where MTs were damaged (white) or severed (blue). Imaging was performed using spinning disk microscopy and photodamage was induced with a 355-nm laser. Scale bars, 10 µm (left) and 4 µm (zoom). **(L)** Kymographs of the events shown in F. Scale bars, 1 µm (horizontal) and 20 s (vertical). See also Video 5.

Finally, we examined whether CSPP-L can recognize sites of MT damage in cells by performing laser microsurgery in COS-7 cells co-expressing GFP-CSPP-L and β-tubulin-mCherry. We damaged single MTs in the z-plane just below the nucleus by local illumination with a 355-nm laser and observed CSPP-L accumulations forming at the illuminated positions (Fig. 5, K and L; and Video 5). It was more difficult to introduce local MT damage by laser microsurgery in cells than in vitro because the intensity of the laser beam varied with MT positions in the z-plane, so the degree of the photodamage was difficult to predict. For the analysis, we only considered events where a new CSPP-L signal appeared at the position where the MT intensity was reduced after laser illumination. To distinguish partial damage from complete severing, we focused on the events where the illuminated MT was visible on both sides of the newly formed CSPP-L accumulation and where both MT parts moved synchronously with the photobleached region (Fig. 5, K and L). The average time between laser illumination and the appearance of the CSPP-L signal was 21 ± 13 s and the size of the CSPP-L accumulation was 564 ± 157 nm (mean ± SD, n = 83). Thus, CSPP1 can bind to damaged MT lattices in vitro and in cells.

**CSPP1 stabilizes damaged MTs and promotes lattice integrity**
Laser-severing experiments in vitro suggested that CSPP1 stabilizes GDP-bound MT plus ends. To further prove that CSPP1 can stabilize damaged MTs, we again used Taxol-stabilized MTs. As mentioned in the previous section, Taxol-stabilized MTs acquire extensive lattice defects when they are incubated in absence of soluble tubulin and free Taxol. In contrast, the presence of free Taxol can prevent MT disassembly and erosion by binding to and stabilizing the defects. The binding of CSPP-L to Taxol-stabilized MTs was suppressed by the presence of free Taxol (Fig. 6, A and B; and Fig. S4 A). In the absence of Taxol in solution, Taxol-stabilized MTs gradually depolymerized (Fig. 6 A). To quantify the effects of Taxol, CSPP-L, and free tubulin on MT stability, we determined the percentage of Taxol-stabilized seeds surviving after 5 min (Fig. 6 C). CSPP-L could slow down though not block MT depolymerization in a concentration-dependent manner (Fig. 6, A and C). The addition of free Taxol to these assays stabilized MTs completely, but when CSPP-

L was also present, stabilization was slightly reduced, suggesting a potential competition between Taxol and CSPP1 for MT binding. The addition of low concentrations of free tubulin (2–5 µM) in the absence of free Taxol had a very mild stabilizing effect in these assays, but in the presence of CSPP-L, complete MT stabilization was observed already at 2 µM tubulin (Fig. 6, A and C). At 5 µM tubulin, CSPP-L even facilitated new MT lattice outgrowth (Fig. 6 A), indicating that it might lower the tubulin concentration threshold for templated MT polymerization, as previously observed with some other MT regulators (Aher et al., 2018; Wieczorek et al., 2015). To confirm this conclusion, we repeated the assays with GMPCPP-stabilized MT seeds and found that CSPP-L strongly increased the frequency of MT outgrowth from seeds at 5 µM tubulin (Fig. 6 D and Fig. S4 B). Interestingly, CSPP-L intensity along the newly formed MT lattice was much higher when MTs were grown in 5 µM tubulin compared with 15 µM tubulin (Fig. 6, E and F). This suggests that CSPP-L binds to MTs more efficiently when they grow slowly. Thus, CSPP-L stabilizes MT polymerization intermediates at MT tips or damage sites, particularly when tubulin addition occurs slowly.

To better understand the mechanism underlying the activity of CSPP1 on damaged MTs, we again turned to cryo-ET. We stabilized MTs by the addition of Taxol and then resuspended them in a buffer containing CSPP-L in the absence of free tubulin with or without free Taxol (as shown in Fig. 6 A). The absence of free Taxol increased CSPP-L binding: on average 26 ± 23% of MTs contained intraluminal densities, compared with only 8 ± 5% in presence of Taxol (P < 0.01, Fig. 6, G and H). Despite the fact that samples with disassembling Taxol-stabilized MTs contained many incomplete lattices and tubulin sheets, we only observed intraluminal densities inside fully closed tubes (Fig. 6 G). Moreover, CSPP-L accumulation zones did not recruit CAMSAP3 and thus did not contain lattice apertures (Fig. S4 C), unlike previous observations with Fchitax-3 (Rai et al., 2020). The addition of vinblastine during MT growth led to the appearance of more numerous defects in the MT lattices (Fig. 6, I and J). However, the presence of both vinblastine and CSPP-L during MT growth led to a significant reduction in the number of lattice defects when compared with vinblastine alone

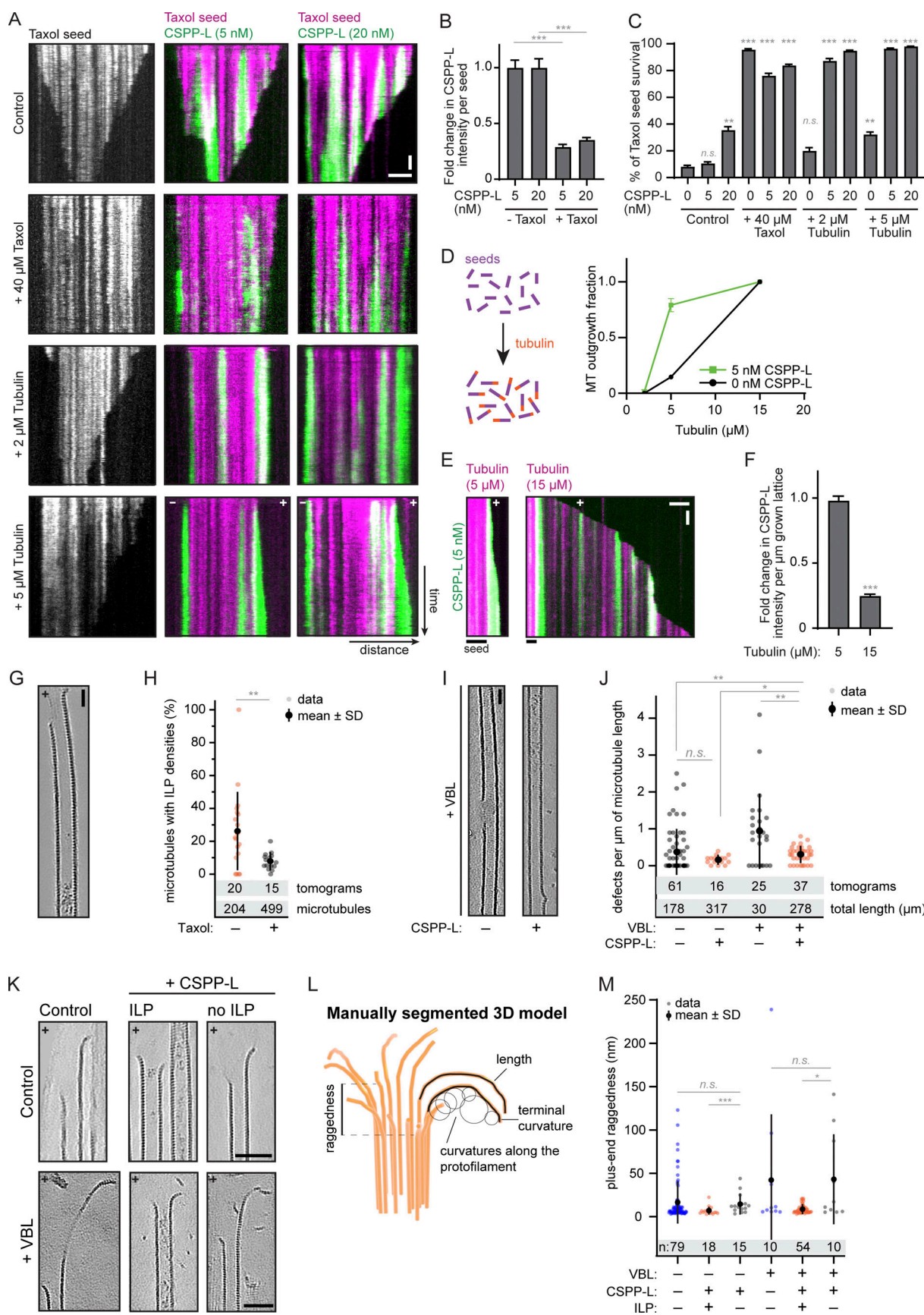

Figure 6. **CSPP1 stabilizes MTs by promoting lattice repair. (A)** Kymographs of Taxol-stabilized MT seeds in absence or presence of the indicated Taxol, tubulin, and GFP-CSPP-L concentrations. Scale bars, 2 µm (horizontal) and 60 s (vertical). **(B)** GFP-CSPP-L intensity quantification per MT seed (from

kymographs as shown in A). Mean GFP-CSPP-L intensity was measured along the entire length of the seed 2 min after flowing in the protein. The average mean intensity of GFP-CSPP-L in presence of 40 µM Taxol was normalized to the average mean intensity in absence of free Taxol. Total number of Taxol-stabilized MT seeds analyzed: 5 nM CSPP-L alone, $n = 102$; 20 nM CSPP-L alone, $n = 108$; 40 µM Taxol, $n = 99$; 40 µM Taxol together with 5 nM CSPP-L, $n = 84$; 40 µM Taxol together with 20 nM CSPP-L, $n = 114$. Bars represent pooled data from two independent experiments. Error bars represent SEM. ***, $P < 0.001$; Kruskal–Wallis test followed by Dunn's post-test. **(C)** Quantification of the percentage of MT seeds that survived 5 min after flow-in of the reaction mix (from kymographs as shown in A). Total number of Taxol-stabilized MT seeds analyzed: control: $n = 95$; 5 nM CSPP-L alone, $n = 120$; 20 nM CSPP-L alone, $n = 110$; 40 µM Taxol, $n = 99$; 40 µM Taxol together with 5 nM CSPP-L, $n = 84$; 40 µM Taxol together with 20 nM CSPP-L, $n = 120$; 2 µM tubulin alone, $n = 115$; 2 µM tubulin together with 5 nM CSPP-L, $n = 124$; 2 µM tubulin together with 20 nM CSPP-L, $n = 112$; 5 µM tubulin alone, $n = 122$; 5 µM tubulin together with 5 nM CSPP-L, $n = 106$; 5 µM tubulin together with 20 nM CSPP-L, $n = 128$. Bars represent pooled data from two independent experiments. Error bars represent SEM. ***, $P < 0.001$; **, $P < 0.01$; n.s., not significant; Kruskal–Wallis test followed by Dunn's post-test. In "control," conditions with 5 and 20 nM CSPP-L are compared to 0 nM CSPP-L, and for all other bars, comparisons were made to the same CSPP-L concentration in the control condition. **(D)** Quantification of the fraction of the total GMPCPP seeds that showed MT outgrowth within 10 min at indicated tubulin concentrations, with tubulin alone or together with 5 nM GFP-CSPP-L. Total number of GMPCPP seeds analyzed: 2 µM tubulin alone, $n = 74$; 5 µM tubulin alone, $n = 75$; 15 µM alone, $n = 69$; 2 µM tubulin together with 5 nM CSPP-L, $n = 70$; 5 µM tubulin together with 5 nM CSPP-L, $n = 66$; 15 µM tubulin together with 5 nM CSPP-L FL, $n = 71$. Symbols represent pooled data from two independent experiments. Error bars represent SEM. **(E)** Kymographs of GMPCPP-stabilized MT seeds in the presence of 5 nM GFP-CSPP-L and the indicated tubulin concentrations. Scale bars, 2 µm (horizontal) and 60 s (vertical). **(F)** GFP-CSPP-L intensity quantification per µm newly grown MT lattice (from kymographs as shown in E). GFP-CSPP-L integrated intensity was measured on newly grown lattice 5 min after flow-in of the reaction mix. The integrated intensity was normalized to newly grown MT lattice length, and the average mean intensity of GFP-CSPP-L in presence of 15 µM tubulin was normalized to the average mean intensity in presence of 5 µM tubulin. Total number of growth episodes analyzed: 5 µM tubulin, $n = 105$; 15 µM tubulin, $n = 104$. Bars represent pooled data from two independent experiments. Error bars represent SEM. ***, $P < 0.001$, Mann-Whitney test. **(G)** Denoised tomograms of dynamic MTs polymerized in the presence of 250 µM Taxol, resuspended in buffer containing only 20 nM GFP-CSPP-L with or without free 40 µM Taxol, vitrified on EM grids. Scale bar, 25 nm. **(H)** Quantification of the percentage of MTs containing luminal densities from total MTs (from tomograms as shown in G). Orange and gray dots (single data points, tomograms), black circle (mean), SD (error bars). **, $P < 0.01$, Mann–Whitney test. Analysis from two independent experiments. **(I)** Denoised tomograms of dynamic MTs polymerized in the presence or absence of 250 nM vinblastine with or without 20 nM GFP-CSPP-L, vitrified on EM grids. Scale bar, 25 nm. **(J)** Quantification of the number of defects per µm MT (from tomograms as shown in I. Orange and gray dots (single data points, tomograms), black circles (mean), SD (error bars). *, $P < 0.1$, **, $P < 0.01$, n.s., not significant, Mann–Whitney test. Analysis from two independent experiments. **(K)** Denoised tomograms of MT ends in the presence or absence of 250 nM vinblastine with or without 20 nM GFP-CSPP-L, vitrified on EM grids. Scale bars, 50 nm. **(L)** Parameters extracted from manual segmentations of terminal protofilaments. Raggedness is defined as the standard deviation in the coordinate along the MT axis of the first deflection point for each protofilament in a MT end (Gudimchuk et al., 2020). **(M)** Quantification of plus-end raggedness (from tomograms as shown in K). Blue, orange, and gray dots (single data points, tomograms), black circle (mean), SD (error bars). *, $P < 0.1$, ***, $P < 0.001$, n.s., not significant, Mann–Whitney test. Analysis from two independent experiments. See also Fig. S4.

---

$(0.3 \pm 0.2$ vs $1 \pm 1$ defects/µm, $P = 0.005)$. These observations, in combination with the increased number of ILP-containing MTs in the presence of vinblastine (Fig. 4, A and B), support our hypothesis that CSPP1 can enter MTs through lattice openings and then promote their repair.

To explain how CSPP1 stabilizes MTs, we analyzed the shapes of terminal tubulin flares in our cryo-ET samples. We observed flared protofilaments in all conditions tested, consistent with recently published cryo-ET analyses (Gudimchuk et al., 2020; McIntosh et al., 2018). We used manual segmentation to extract the parameters of the shape of the protofilament flares at the ends of MTs: protofilament curvature and length in 3D, as well as MT end raggedness, which we quantified as the standard deviation in the coordinate along the MT axis of the first deflection point for each protofilament in an MT end (Fig. 6, K and L). We used raggedness as a measure of MT taper length, as described in previous analyses (Gudimchuk et al., 2020). Comparing MT ends with particles within their lumen to ILP-free MT ends in the same sample, we did not observe any significant differences in protofilament length or curvature (Fig. S4 D). However, we did observe the absence of long tapers within ILP-containing MT ends comparing ILP-positive and ILP-negative MTs in presence of both CSPP-L and vinblastine (Fig. 6 M). Since long tapers are thought to form at faster-growing MTs or MTs in a precatastrophe state (Chretien et al., 1995; Coombes et al., 2013; Duellberg et al., 2016; Gudimchuk et al., 2020), reduced tapering in the presence of CSPP1 is consistent with the lack of catastrophes and slower MT growth (Fig. 1). Absence of long tapers might indicate that CSPP1 does not act at terminal

protofilament flares but stabilizes MTs by holding protofilaments together within the tube, thus preventing MT disassembly and allowing them to resume growth. In a similar way, CSPP1 could potentially bind to damaged lattices and hold protofilaments together to enable lattice repair by tubulin incorporation.

## Discussion

While a lot of information exists on the control of MT dynamics by proteins associated with the outer MT surface, the regulatory effects of factors binding to MT lumen are understood much less well. Here, we show that the ciliary tip regulator CSPP1 is an intraluminal protein and dissect its behavior and molecular function. We show that CSPP1 displays some striking parallels to MT-stabilizing compounds, such as taxanes and epothilones, which also bind to MT lumen (reviewed in Steinmetz and Prota, 2018). Similar to these compounds, CSPP1 binds to polymerizing MT ends in the precatastrophe state, when the GTP cap is diminished, prevents catastrophe, and induces MT pausing followed by growth (Fig. 7); at a low concentration, CSPP1 triggers the formation of sites of stabilized MT lattice that causes repeated rescues ("stable rescue sites" [Rai et al., 2020]). Preferential accumulation of CSPP1 at growing MT ends can be explained by the better accessibility of intraluminal binding sites, which become available when tubulin dimers are added to MT ends. Theory predicts that intraluminal diffusion of a protein with affinity for the inner MT surface would be very slow (Odde, 1998). Furthermore, unlike small molecules, CSPP1 would

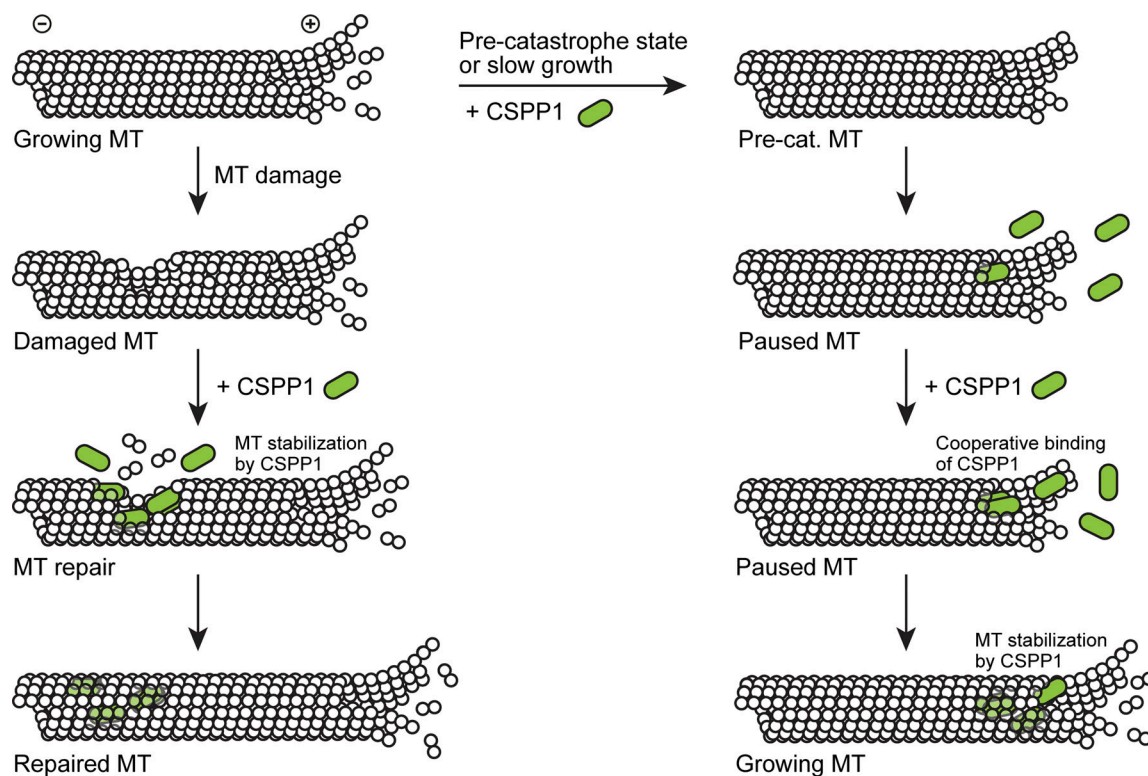

**Figure 7. Stabilization of perturbed or damaged MTs by CSPP1.** Schematic model showing MT stabilizing activity of CSPP1. Growing MTs can acquire damages, creating access for CSPP1 to MT lumen, and CSPP1 potentially promotes MT lattice repair (left). Alternatively, an MT can undergo a growth perturbation (e.g., switch to a pre-catastrophe state), resulting in slow growth and/or altered tip structure. This promotes the binding of CSPP1 to microtubule lumen (right). First, a single CSPP1 molecule will bind, followed by additional CSPP1 molecules, causing a pause and increasing MT stability. Subsequently, the MT can resume rapid growth, while CSPP1 remains localized in the MT lumen.

be too large to penetrate the MT lumen through the regular lattice fenestrations, although it does bind to sites where the lattice has been damaged (Fig. 7). Additionally, for CSPP1 to be able to accumulate inside the MT, this damage needs to be sufficiently large as accumulations are readily observed in Taxol-stabilized MTs with large defects but not in GMPCPP-stabilized MTs, which have smaller defects.

The selectivity of CSPP1 for precatastrophe MT ends could be explained by their specific conformation (such as the presence of tubulin sheets or tapers, or the loss of GTP-tubulin) or simply by their slow growth. The observations that CSPP1 binds to growing MT ends better when tubulin concentration is low and that it strongly accumulates inside MT lattices polymerized from the minus end, which grows much slower than the plus end, support this notion. Interestingly, CSPP1 initially binds to MTs as a monomer, but then a second molecule is recruited to the same site with some delay and likely stabilizes the binding (Fig. 7). Rapid tubulin addition to the MT tip and the ensuing fast closure of the tube might prevent the binding of the second CSPP1 molecule, which would be needed for the stable association of CSPP1 with the MT lumen. This would explain why CSPP1 accumulations form much more efficiently at slowly growing MT tips.

Our previous data demonstrated that Fchitax-3 binds to MT ends cooperatively (Rai et al., 2020). Since the binding profile of CSPP1 is very similar, it also seems to cooperatively bind to MT

tips, explaining how CSPP1 forms regions of high enrichment even when present at low concentrations. After binding, CSPP1 exerts an MT-stabilizing effect by preventing shrinkage; it could do so by supporting individual protofilaments and/or by promoting lateral interactions between protofilaments, and both mechanisms would be consistent with the action of MT-stabilizing agents (Elie-Caille et al., 2007; Prota et al., 2013; reviewed in Steinmetz and Prota, 2018). Spanning lateral protofilament contacts could potentially explain how CSPP1 reduces tip raggedness and why it is not found at protofilament flares.

Another interesting property of CSPP1 is its ability to induce pausing. While this property also resembles the effect of low concentrations of taxanes, in CSPP1, the lumen binding and growth-inhibiting functions depend on two separate protein domains. The presence of two activities, an activity that inhibits polymerization and an activity that prevents MT shrinkage, seems to be a common property of MT growth inhibitors, such as the kinesin-4 KIF21B (van Riel et al., 2017) or the centriolar protein CPAP (Sharma et al., 2016). In CSPP1, both regulatory domains are predicted to be helical and are quite short, with less than a hundred amino acids. The presence of α-helices seems to be a common property of ciliary MIPs, including many linearly arranged proteins that form the regularly spaced inner sheath within ciliary doublets (Gui et al., 2021; Ichikawa and Bui, 2018; Ma et al., 2019).

Identification of a minimal lumen-binding domain of CSPP1 (termed here the MTB) can be potentially useful for directing different protein activities to the MT lumen. It is possible that the binding site of the CSPP1 MTB domain overlaps with that of Taxol because we found some evidence of competition between Taxol and CSPP1 in MT stabilization assays. Importantly, there is also a notable difference between the effects of MT-stabilizing drugs and CSPP1: taxanes induce structural defects (holes) in MT lattices because they promote switching in protofilament number (Rai et al., 2021). In contrast, CSPP1 seems to promote lattice integrity. Although CSPP1 can specifically bind to the sites of lattice damage, CSPP1 densities are predominantly found within complete tubes; moreover, CSPP1 reduces the number of vinblastine-induced lattice defects and stabilizes eroding MT seeds. CSPP1 likely acts in part by stabilizing protofilament ends close to the damage sites and possibly by promoting tubulin incorporation to form complete tubes. Whether CSPP1 participates in the repair of MT defects in cells, either on cytoplasmic or axonemal MTs, remains to be determined. There are indications that cellular MTs can be damaged by interaction with other MTs, severing enzymes or motor proteins that use MTs as rails (Aumeier et al., 2016; Gazzola et al., 2022 *Preprint*; Triclin et al., 2021; Vemu et al., 2018). The ability of CSPP1 to specifically bind to incomplete MTs can be harnessed for studying MT damage and repair. Another protein, SSNA1, was also reported to bind to MT defects, although it appears much less potent than CSPP1 in stabilizing MTs because 0.5–5 µM SSNA1 was needed to affect MT growth in vitro (Lawrence et al., 2021), whereas CSPP1 displays strong effects already at 5–10 nM concentration. It would be interesting to examine whether SSNA1 is also an intraluminal protein, as it was reported to stabilize partial MT structures (Basnet et al., 2018).

CSPP1 also shows some similarities to another intraluminal protein that has been analyzed in vitro, MAP6 (Cuveillier et al., 2020). While MAP6 shows some strikingly distinct features, such as the induction of MT coiling and lattice apertures (Cuveillier et al., 2020), both MAP6 and CSPP1 are MT stabilizers, which reduce overall MT shrinkage and promote rescues. Furthermore, both proteins contain a short domain that can perturb processive growth. In the case of MAP6, this domain is also required for the formation of intraluminal particles, and without it, the protein seems to function on the outer MT surface. In contrast, our cryo-ET and MINFLUX data support the idea that CSPP1 binds only to the inner surface of the MT. It is however still possible that some parts of CSPP1 extend out of the tube. For example, the site of action of the growth-inhibiting part of CSPP1 is currently unclear, as the shape and curvature of the protofilament flares in the presence of CSPP1 looked very similar to that of control MTs and thus provided no clues on the nature of this activity. Furthermore, CSPP1 is part of a multiprotein module associated with ciliary tips (Latour et al., 2020), and two other members of the same module, TOGARAM1 and CEP104, are likely to bind to the outer MT surface because they contain canonical tubulin-binding TOG domains; moreover, CEP104 binds to EBs, which decorate MTs from the outside (Al-Jassar et al., 2017; Das et al., 2015; Jiang et al., 2012; Rezabkova et al., 2016).

CSPP1 participates in controlling the elongation and stability of ciliary axonemes, and when CSPP1 or its binding partners are absent, ciliogenesis is impaired and cilia are shorter (Frikstad et al., 2019; Latour et al., 2020; Patzke et al., 2010). Our findings help to explain the MT-stabilizing activity of CSPP1 and suggest that ciliary tips are kept in shape by protein complexes that span both the inner and the outer MT surface. This arrangement might be important for controlling different signaling pathways such as Hedgehog signaling, which strongly relies on the state of axoneme tip and is dysregulated by ciliopathies (Andreu-Cervera et al., 2021; Hildebrandt et al., 2011; Reiter and Leroux, 2017). Furthermore, the similarity between the activities of CSPP1 and MT-stabilizing agents raises an interesting possibility that the absence of CSPP1 or its binding partners might be compensated by such compounds, suggesting potential avenues for pharmacological intervention in ciliopathies.

## Materials and methods

### DNA constructs, cell lines, and cell culture

CSPP1 truncations expressed in mammalian cells were made from full-length constructs described previously (Patzke et al., 2005; Patzke et al., 2006) in modified pEGFP-C1 or pmCherry-C1 vectors with a StrepII tag. HEK293T cells and COS-7 cells (ATCC) were cultured in DMEM medium (Lonza) supplemented with 10% fetal calf serum (FCS; GE Healthcare Life Sciences) and 1% (v/v) penicillin/streptomycin. hTERT RPE-1 cells (ATCC) stably expressing low levels of mNG-CSPP1 were generated previously (Frikstad et al., 2019) and resorted to keeping low-expressing cells. An mNG-ARL13B expressing cell line was generated similarly. These cells were cultured in DMEM-F12 medium (Life Technologies) supplemented with 10% fetal calf serum (FCS; Life Technologies) and 1% (v/v) penicillin/streptomycin. For experiments with ciliated cells, RPE-1 cells were serum starved for 24 h by 2× washing with DMEM/F12 with 1% (v/v) penicillin/ streptomycin. All cells were routinely checked for mycoplasma contamination using the MycoAlertTM Mycoplasma Detection Kit (Lonza). For overexpression of CSPP1 constructs, COS-7 cells were transiently transfected with FuGENE6 (Promega) with different StrepII-GFP-CSPP1 constructs for 24 h. Single transfections were used for immunofluorescence experiments, and co-transfections with EB3-mCherry (Stepanova et al., 2003), βIVb-tubulin-mCherry (Bouchet et al., 2016) or StrepII-GFP-MAP7 FL (Hooikaas et al., 2019) were used for live-cell imaging or MINFLUX microscopy. Lipofectamine RNAiMAX (Invitrogen) was used to transfect RPE-1 cells with siRNA according to the manufacturer's instructions (see Table S1). Corresponding experiments were performed 48 h after siRNA transfection.

### Protein purification from HEK293T cells for in vitro reconstitution assays

For the purification of CSPP1 constructs, HEK293T cells were transiently transfected with polyethyleneimine (Polysciences) with different StrepII-GFP-CSPP1 constructs. The cells were harvested 28 h after transfection. Cells from a 15-cm dish were lysed in 500 µl lysis buffer (50 mM HEPES, 300 mM NaCl, 1 mM MgCl$_2$, 1 mM DTT, 0.5% Triton X-100, pH 7.4) supplemented

with protease inhibitors (Roche) on ice for 15 min. The lysate was cleared from debris by centrifugation and the supernatant was incubated with 20 µl StrepTactin beads (GE Healthcare) for 45 min. Beads were washed five times with a 300 mM salt wash buffer (50 mM HEPES, 300 mM NaCl, 1 mM MgCl₂, 1 mM EGTA, 1 mM DTT, and 0.05% Triton X-100; pH 7.4) and three times with a 150 mM salt wash buffer (similar to the 300 mM salt buffer but with 150 mM NaCl). The protein was eluted in elution buffer (similar to the 150 mM salt wash but supplemented with 2.5 mM d-Desthiobiotin [Sigma-Aldrich]) where the volume depended on the expression levels before harvesting. Purified proteins were snap-frozen and stored at –80°C.

## Mass spectrometry

To confirm we purified GFP-CSPP-L without any interactors that could affect its effect on MT dynamics, the purified protein sample was digested using S-TRAP microfilters (ProtiFi) according to the manufacturer's protocol. In short, 7 µg of protein sample was denatured in 5% SDS buffer and reduced and alkylated using DTT (20 mM, 10 min, 95°C) and iodoacetamide (IAA; 40 mM, 30 min). After acidification, the proteins were precipitated using a methanol triethylammonium bicarbonate buffer (TEAB) after which they were loaded on the S-TRAP column. The trapped proteins were washed four times with the methanol TEAB buffer and then digested using 1 µg Trypsin (Promega) overnight at 37°C. Digested peptides were eluted and dried in a vacuum centrifuge before liquid chromatography-mass spectrometry (LC-MS) analysis.

The sample was analyzed by reversed-phase nLC-MS/MS using an Ultimate 3000 UHPLC coupled to an Orbitrap Q Exactive HF-X mass spectrometer (Thermo Fisher Scientific). Digested peptides were separated using a 50-cm reversed-phase column packed in-house (Agilent Poroshell EC-C18, 2.7 µm, 50 cm × 75 µm). The peptides were eluted from the column at a flow rate of 300 nl/min using a linear gradient with buffer A (0.1% formic acid [FA]) and buffer B (80% acetonitrile [ACN], 0.1% FA) ranging from 13 to 44% B over 38 min. This procedure was followed by a column wash and re-equilibration step resulting in a total data acquisition time of 55 min. Mass spectrometry data were acquired using a data-dependent acquisition (DDA) method with the following MS1 scan parameters: maximum injection time of 20 ms, automatic gain control (AGC) target equal to 3E6, 60,000 resolution, and a scan range of 375–1,600 m/z, acquired in profile mode. The MS2 method was set at 15,000 resolution, an automatic maximum injection time, with an AGC target set to standard and an isolation window of 1.4 m/z. Scans were acquired using a fixed first mass of 120 m/z and a mass range of 200-2,000 and normalized collision energy (NCE) of 28. Precursor ions were selected for fragmentation using a 1-s scan cycle, a dynamic exclusion time set to 10 s, and a precursor charge selection filter for ions possessing +2 to +6 charges.

Raw files were processed using Proteome Discoverer (PD; version 2.4; Thermo Fisher Scientific). MSMS fragment spectra were searched using Sequest HT against a human database (UniProt, year 2020) that was modified to contain the exact protein sequence from SII-GFP-CSPP-L and a common contaminants database. The search parameters were set using a fragment mass tolerance of 0.06 Da and a precursor mass tolerance of 20 ppm. The maximum amount of missed cleavages for trypsin digestion was set to two. Methionine oxidation and protein N-term acetylation were set as variable modifications and carbamidomethylation was set as a fixed modification. Percolator was used to assign a 1% false discovery rate (FDR) for peptide spectral matches, and a 1% FDR was applied to protein and peptide assemblies. For peptide-spectrum match (PSM) inclusion, an additional filter was set to require a minimum Sequest score of 2.0. The Precursor Ion Quantifier node was used for MS1-based quantification and default settings were applied. Precursor ion feature matching was enabled using the Feature Mapper node. Proteins that matched the common contaminate database were filtered out from the results table.

## In vitro reconstitution assays
### MT seed preparation
Double-cycled GMPCPP-stabilized MT seeds or Taxol-stabilized MT seeds were used as templates for MT nucleation or to test protein binding in in vitro assays. GMPCPP-stabilized MT seeds were prepared as described before (Mohan et al., 2013). Briefly, a tubulin mix consisting of 70% unlabeled porcine brain tubulin, 18% biotin-labeled porcine tubulin, and 12% rhodamine-labeled porcine tubulin (all from Cytoskeleton) was incubated with 1 mM GMPCPP (Jena Biosciences) at 37°C for 30 min. Polymerized MTs were pelleted by centrifugation in an Airfuge for 5 min at 199,000 × g and then depolymerized on ice for 20 min. Next, MTs were let to polymerize again at 37°C with newly added 1 mM GMPCPP. Polymerized MT seeds were then pelleted as above and diluted tenfold in MRB80 buffer containing 10% glycerol. Last, MT seeds were frozen and stored at –80°C. Taxol-stabilized MT seeds were prepared as described before with some modifications (Aher et al., 2020). Briefly, a tubulin mix consisting of 28 µM porcine brain tubulin, 10% biotin-labeled porcine tubulin, and 4.5% rhodamine-labeled porcine tubulin was incubated with 2 mM GTP (Sigma-Aldrich) and 20 µm Taxol at 37°C for 35 min. Then, 20 µM Taxol was added to the tubulin mix, and polymerized MTs were pelleted by centrifugation for 15 min at 16,200 × g at room temperature. The MT pellet was resuspended in warm 20 µM Taxol solution in MRB80 buffer and stored at room temperature in the dark for a maximum of 1 d.

### In vitro reconstitution assays
In vitro assays with dynamic or stabilized MTs were performed as described before (Rai et al., 2020). In short, plasma-cleaned glass coverslips (square or rectangular) were attached on microscopic slides by two strips of double-sided tape. The coverslips were functionalized by sequential incubation with 0.2 mg/ml PLL-PEG-biotin (Susos AG) and 1 mg/ml neutravidin (Invitrogen) in MRB80 buffer (80 mM piperazine-N, N[prime]-bis (2-ethane sulfonic acid), pH 6.8, supplemented with 4 mM MgCl₂, and 1 mM EGTA). Then, GMPCPP- or Taxol-stabilized MT seeds were attached to the coverslips through biotin–neutravidin interactions. During the subsequent blocking step with 1 mg/ml κ-casein, the reaction mix containing the different

concentrations of purified proteins and drugs was spun down in an Airfuge for 5 min at 119,000 × g. For dynamic MTs, the reaction mix consisted of MRB80 buffer supplemented with 15 µM porcine brain tubulin (100% dark porcine brain tubulin when 20 nM GFP-EB3 or mCherry-EB3 was added, or 97% dark porcine brain tubulin with 3% rhodamine- or HiLyte488-labeled porcine brain tubulin), 50 mM KCl, 1 mM GTP, 0.2 mg/ml κ-casein, 0.1% methylcellulose, and oxygen scavenger mix (50 mM glucose, 400 µg/ml glucose oxidase, 200 µg/ml catalase and 4 mM DTT). For stabilized MTs, porcine tubulin, GTP, and EB3 were omitted from the reaction mix. After spinning, the reaction mix was added to the flow chamber and the flow chamber was most often sealed with vacuum grease or left open (for flow-in assays during acquisition or for MINFLUX sample preparation). MTs were imaged immediately at 30°C using a TIRF microscope. All tubulin products were from Cytoskeleton Inc.

To estimate the number of GFP-CSPP-L molecules per 8 nm MT, two parallel flow chambers were made on the same coverslip. In one chamber, regular MT dynamic assay in the presence of GMPCPP-stabilized MT seeds with tubulin, EB3-mCherry, and 5 nM GFP-CSPP-L was performed. The other chamber was incubated with strongly diluted GFP protein so that single molecules were detectable. MTs were left to polymerize for 5–10 min. Then, for both the chamber with single GFP molecules and the chamber with dynamic MT and GFP-CSPP-L, 20 images of unexposed coverslip areas were acquired at 100-ms exposure time using high laser intensity. To measure the intensity of single binding events of GFP-CSPP1 fragments upon binding, similar assays were performed with minor modifications. The GFP-CSPP1 fragment concentrations were lower and MT growth was imaged overtime at 2 s per frame.

### In vitro assays for Cryo-ET sample preparation
Sample preparation for imaging in vitro MTs with cryo-ET is a slightly modified version of the method described above. All steps occur in a tube instead of a flow chamber. After centrifugation of the reaction mix for dynamic MTs, GMPCPP-stabilized seeds and 5-nm gold particles were added, and MTs were left to polymerize for 20–30 min at 37°C. Then, 3.5 µl was transferred to a recently glow-discharged, lacey carbon grid suspended in the chamber of Leica EM GP2 plunge freezer, equilibrated at 37°C and 98% relative humidity. The grid was immediately blotted for 4 s and plunge-frozen in liquid ethane.

### In vitro assays for MINFLUX sample preparation
Sample preparation for imaging in vitro MTs with MINFLUX microscopy is a slightly modified version of the method described above. For flow chambers, round plasma-cleaned coverslips were attached to big, rectangular coverslips via two strips of glue (Twinsil). The reaction mix contained the same components as for dynamic MTs, supplemented with a CF680-GFP-Nanobody and SNAP-Abberior FLUX-640. After the addition of the reaction mix, the chamber was left open and was incubated in a 30°C incubator for 15 min. To remove the background signal, the flow chamber was washed with a second reaction mix containing 25 µM tubulin before fixing with 1% glutaraldehyde (Electron Microscopy Sciences) for 5 min at room temperature.

After washing with MRB80, the round glass coverslip was demounted and stored in MRB80 at 4°C or incubated with gold nanoparticles (Nanopartz) for 5 min. Then, the coverslips were mounted in GLOX buffer (50 mM Tris/HCl, pH 8, 10 mM NaCl, 10% (w/v) d-glucose, 500 µg/ml glucose oxidase, and 40 µg/ml glucose catalase) supplemented with 56 mM 2-Mercaptoethylamin (MEA) and sealed with glue (Picodent Twinsil).

### Immunofluorescence staining of fixed cells
#### Sample preparation for widefield and airyscan confocal imaging
For immunofluorescence staining experiments, COS-7 cells were seeded on coverslips 1 d before transfection. Cells were fixed after 24 h with either –20°C MeOH for 10 min (staining for acetylated tubulin, α-tubulin, PCM1, and CSPP1) or –20°C MeOH for 10 min followed by 4% paraformaldehyde for 15 min at room temperature (staining for α-tubulin and EB1). This was followed by permeabilization with 0.15% Triton X-100 for 2 min. Next, samples were blocked with 1% bovine serum albumin (BSA) diluted in phosphate-buffered saline (PBS) supplemented with 0.05% Tween-20 for 45 min at room temperature and sequentially incubated with primary antibodies for 1 h at room temperature and fluorescently labeled with secondary antibodies for 45 min at room temperature. Finally, samples were washed, dried, and mounted in Vectashield (Vector laboratories).

For immunofluorescence staining experiments of RPE-1 cells stably expressing low levels of mNG-CSPP1, cells were fixed and stained as described before (Frikstad et al., 2022). Briefly, cells were seeded on coverslips 1 d before fixation. Cells were fixed with 1.6% paraformaldehyde in PBS for 10 min at room temperature and then postfixed in –20°C MeOH for 20 min. Next, samples were blocked and permeabilized with 1% bovine serum albumin (BSA) diluted in phosphate-buffered saline (PBS) supplemented with 0.5% Triton X-100 for 15 min and sequentially incubated with primary antibodies for 2 h at room temperature and fluorescently labeled with secondary antibodies for 45 min at room temperature. Finally, samples were washed, dried, and mounted in ProLong Diamond antifade mounting medium (Thermo Fisher Scientific).

#### Sample preparation for MINFLUX microscopy imaging
For immunofluorescence staining experiments, COS-7 cells were seeded on coverslips 1 d before transfection. 24 h after transfection, the cells were incubated with warm extraction buffer (0.2% glutaraldehyde, 0.35% Triton X-100 in MRB80) for 2 min before incubation with fixation buffer (0.1% glutaraldehyde, 4% paraformaldehyde and 4% sucrose [w/v]) for 10 min at room temperature (staining for α-tubulin and EB1) followed by permeabilization with 0.5% Triton X-100 for 10 min. Next, samples were quenched with 100 mM NaBH$_4$ before blocking with Image-iT Signal Enhancer (Thermo Fisher Scientific) for 30 min at room temperature and sequentially incubated with 1 µM Alexa647-SNAP-dye (NEB) and 1 mM DTT in PBS for 1 h at room temperature. Next, samples were blocked with 1% bovine serum albumin (BSA) diluted in phosphate-buffered saline (PBS) for 50 min at room temperature and sequentially incubated with CF680-GFP-Nanobody for 1 h at room temperature or overnight at 4°C. Coverslips were incubated with gold nanoparticles

(Nanopartz) for 5 min. Then, the coverslips were mounted in GLOX buffer supplemented with 56 mM MEA and sealed with glue (Picodent Twinsil).

## Western blot analysis

For Western blot analysis, RPE-1 cells were washed twice with PBS before they were lysed in cold 2× SDS-PAGE sample buffer (supplemented with 5 µl/ml Benzonase) on ice for 10 min. Samples were boiled for 5 min before loading on SDS-PAGE followed by Western blotting for analysis.

## Microscopy

### Widefield microscopy

Fixed RPE-1 cells were imaged using appropriate optical filters on a multifluorescent bead calibrated AxioImager Z1 ApoTome microscope system (Carl Zeiss) equipped with a 100× or a 63× lens (both PlanApo N.A.1.4) and an AxioCam MRm camera. This setup gives confocal sections on an epifluorescence microscope. Images are presented as maximal projections of z-stacks using Axiovision 4.8.2 software (Carl Zeiss).

Fixed and stained COS-7 cells were imaged using widefield fluorescence illumination on a Nikon Eclipse Ni upright microscope equipped with a Nikon DS-Qi2 camera (Nikon), an Intensilight C-HGFI precentered fiber illuminator (Nikon), ET-DAPI, ET-EGFP, and ET-mCherry filters (Chroma), controlled by Nikon NIS Br software and using a Plan Apo Lambda 60× NA 1.4 oil objective (Nikon). For presentation, images were adjusted for brightness using ImageJ 1.50b.

### Airyscan confocal microscopy

Fixed and stained COS-7 cells were imaged using a Carl Zeiss LSM880 Fast AiryScan microscope fitted with the following laser lines: 405 nm, Argon Multiline, 561 and 633 nm. For imaging, an Alpha Plan-APO 100×/1.46 Oil DIC VIS objective was used combined with AiryScan and PMT detectors. ZEN 2.3 software was used to control the microscope and process the raw images. For presentation, images were adjusted for brightness using ImageJ 1.50b.

### 3D-structured illumination microscopy (3D-SIM)

For FRAP experiments, live RPE-1 cells were imaged at 30°C in TIRF mode on the DeltaVision OMX V4 Blaze 3D-SIM microscope equipped with a six-color solid-state illumination and six lasers (405, 445, 488, 514, 568, and 642 nm), UltimateFocus Hardware Autofocus System, three high speed water-cooled PCO.edge sCMOS cameras, and a 60× NA 1.42 oil PLAPON6 PSF objective. The setup was controlled by softWoRx software. The raw pixel size is 0.08 µm and the reconstructed pixel size is 0.04 µm. The 488-nm laser was used for photobleaching at 30% laser power for 15–30 ms illumination with the following imaging sequence: 10 s prebleach imaging, bleach, 50 s recovery imaging. Analysis was performed using the FRAP profiler script in ImageJ.

### TIRF microscopy

In vitro reconstitution assays and live COS-7 cells overexpressing GFP-CSPP-L and mCherry-EB3 were imaged on previously described (iLas2) TIRF microscope setups (Aher et al., 2020).

In brief, we used an inverted research microscope Nikon Eclipse Ti-E (Nikon) with the perfect focus system (Nikon), equipped with Nikon CFI Apo TIRF 100 × 1.49 N.A. oil objective (Nikon) and controlled with MetaMorph 7.10.2.240 software (Molecular Devices). The microscope was equipped with TIRF-E motorized TIRF illuminator modified by Gataca Systems (France). To keep the in vitro samples at 30°C, a stage-top incubator model INUBG2E-ZILCS (Tokai Hit) was used. For excitation, 490 nm 150 mW Vortran Stradus 488 laser (Vortran) and 561 nm 100 mW Cobolt Jive (Cobolt) lasers were used. We used ET-GFP 49002 filter set (Chroma) for imaging of proteins tagged with GFP or tubulin labeled with Hylite488 or ET-mCherry 49008 filter set (Chroma) for imaging of proteins tagged with mCherry or tubulin labeled with rhodamine. Fluorescence was detected using a Prime BSI camera (Teledyne Photometrics) with the intermediate lens 2.5X (Nikon C mount adaptor 2.5X) or an EMCCD Evolve 512 camera (Roper Scientific) without an additional lens. The final resolution using Prime BSI camera was 0.068 µm/pixel, and using EMCCD camera, it was 0.063 µm/pixel.

The iLas3 system (Gataca Systems) is a dual laser illuminator for azimuthal spinning TIRF (or Hilo) illumination and a targeted photomanipulation option. This system was installed on Nikon Ti microscope (with the perfect focus system, Nikon), equipped with 489 nm 150 mW Vortran Stradus 488 laser (Vortran) and 100 mW 561 nm OBIS laser (Coherent), 49002 and 49008 Chroma filter sets, EMCCD Evolve DELTA 512 camera (Teledyne Photometrics) with the intermediate lens 2.5X (Nikon C mount adaptor 2.5X), CCD camera CoolSNAP MYO (Teledyne Photometrics), and controlled with MetaMorph 7.10.2.240 software (Molecular Device). To keep the in vitro samples at 30°C or the live cells at 37°C, a stage-top incubator model INUBG2E-ZILCS (Tokai Hit) was used. The final resolution using the EMCCD camera was 0.064 µm/pixel and using the CCD camera it was 0.045 µm/pixel. This microscope was also used for photoablation. The 532 nm Q-switched pulsed laser (Teem Photonics) as part of iLas3 system was used for photoablation by targeting the laser on the TIRF microscope very close but not directly at the MT lattice to induce damage or directly at the MT lattice for complete severing. For photodamage, a circle with a diameter of 7 pixels was used for 50 ms illumination at 20–25% laser power of the 532-nm pulsed laser.

### Spinning disk microscopy

Photodamage assays in cells were performed using spinning disk microscopy. COS-7 cells overexpressing GFP-CSPP-L and β-tubulin-mCherry were imaged using confocal spinning disc fluorescence microscopy on an inverted research microscope Nikon Eclipse Ti-E (Nikon), equipped with the perfect focus system (Nikon), Nikon Plan Apo VC 100× N.A. 1.40 oil objective (Nikon) and a spinning disk-based confocal scanner unit (CSU-X1-A1, Yokogawa). The system was also equipped with ASI motorized stage with the piezo plate MS-2000-XYZ (ASI), Photometrics PRIME BSI sCMOS camera (Teledyne Photometrics) and controlled by the MetaMorph 7.10.2.240 software (Molecular

Devices). For imaging, we used 487 nm 150 mW Vortran Stradus 488 (Vortran) and 100 mW 561 nm OBIS (Coherent) lasers, the ET-EGFP/mCherry filter (Chroma) for spinning-disc-based confocal imaging. The final resolution using PRIME BSI camera was 0.063 μm/pixel. To keep the live cells at 37°C, a stage-top incubator model INUBG2E-ZILCS (Tokai Hit) was used. The 355 nm laser (Teem Photonics) of the iLAS pulse system was used to induce photodamage by targeting the laser on the spinning disk microscope in a 1-pixel thick line across MTs in the z-plane under the nucleus at 9–11% laser power to induce damage.

### MINFLUX microscopy

MINFLUX imaging was performed on an Abberior MINFLUX microscope (Abberior) equipped with a 1.4 NA 100× oil objective lens as previously described (Schmidt et al., 2021). Two color images were recorded using ratiometric detection on two avalanche photodiodes with Abberior software. The fluorescence signal of two far-red fluorophores was split at 685 nm into two detection channels (Ch1: 650–685 nm and Ch2: 685–750 nm), the ratio between both detector channels allowed to assign individual single molecule events to the respective fluorophores. Images were acquired in 2D or 3D MINFLUX imaging mode using a 642 nm excitation laser (17.4 μW/cm²). Laser powers were measured at the position of the objective back focal plane using a Thorlabs PM100D power meter equipped with a S120C sensor head.

### Cryo-ET

Images were recorded on a JEM3200FSC microscope (JEOL) equipped with an in-column energy filter operated in zero-loss imaging mode with a 30 eV slit. Movies consisting of 8–10 frames were recorded using a K2 Summit direct electron detector (Gatan), with a target total electron dose of 80 e–/Å². Images were recorded at 300 kV with a nominal magnification of 10,000, resulting in a pixel size of 3.668 Å at the specimen level. Imaging was performed using SerialEM software (Mastronarde, 2005), recording bidirectional tilt series starting from 0° ± 60°; tilt increment 2°; and target defocus –4 μm.

### Image analysis
### Analysis of MT plus end dynamics in vitro

Movies of dynamic MTs, acquired as described above, were corrected for drift, and kymographs were generated using the ImageJ plugin KymoResliceWide v.0.4 (https://github.com/ekatrukha/KymoResliceWide). The MT tips were traced with lines, and measured lengths and angles were used to calculate the MT dynamics parameters such as growth rate, pause duration, event duration, and all transition events. All events with growth rates faster than 0.24 μm/min were categorized as growth events and all events with shrinkage rates faster than 0.24 μm/min were categorized as shrinkage events. The events with slower growth rates or faster shrinkage rates than the before mentioned rates were categorized as pause events. Only growth events longer than 0.40 μm and pause events longer than 20 s were included in the analysis. Transition frequency was calculated by dividing the sum of the transition events per experiment by the total time this event could have occurred.

### Quantification of EB1 comets

Images of COS-7 cells overexpressing GFP-CSPP1 constructs and stained for α-tubulin and EB1 were acquired on a widefield microscope as described above. The background was subtracted using the Rollin Ball Background Substraction plugin in ImageJ. This plugin uses the rolling-ball algorithm where we set the radius to 10 pixels. EB1 comets were detected by "MaxEntropy" thresholding and subsequent particle analysis with a minimal size cut-off of 0.10 μm², and the total number of EB1 comets per cell was normalized to 100 μm².

### 3D volume reconstruction and analysis

Reconstruction, denoising, and analysis of tomographic volumes were performed as described previously (Ogunmolu et al., 2021 Preprint). In brief, direct electron detector movie frames were aligned using MotionCor2 (Zheng et al., 2017) and then split into even and odd stacks used further for denoising. Tilt series alignment and tomographic reconstructions were performed with IMOD 4.11 (Kremer et al., 1996). Final tomographic volumes were binned by two, corrected for the contrast transfer function, and the densities of gold beads were erased in IMOD. CryoCARE denoising was performed on tomograms reconstructed from the same tilt-series using even and odd movie frames (Buchholz et al., 2019). Tubulin lattice defects were identified upon visual inspection of denoised tomograms in 3dmod as interruptions of regular MT lattice that could not be attributed to missing wedge artifacts. MTs were sometimes damaged at MT-carbon or MT-MT contacts followed by blotting; these instances were not included in the quantification of defects.

Automated segmentation of denoised tomograms into tubulin and intraluminal densities was performed using the tomoseg module of EMAN2.2 (Chen et al., 2017). To do this, we trained three separate neural networks: "MTs," "tubulin," and "ILP." The resulting segmentations were used to mask the denoised tomographic densities using UCSF Chimera (Pettersen et al., 2004). This resulted in volume maximum projections of "MTs"- and "tubulin"-masked densities in cyan, and "ILP"-masked densities in yellow. Final visualization and rendering were performed in Blender using 3D scenes imported from UCSF Chimera.

Manual segmentation to obtain protofilament shapes at MT ends was performed as described previously (McIntosh et al., 2018; Ogunmolu et al., 2021 Preprint) using 3dmod (Kremer et al., 1996). Protofilament coordinates were further analyzed using Matlab scripts available at https://github.com/ngudimchuk/Process-PFs.

### MINFLUX data analysis

Images of MTs were rendered from MINFLUX data as a density map as described in Schmidt et al. (2021). For the two-channel data, the channels were separated by applying cut-off on the "dcr" (detector channel ratio) attribute of the MINFLUX data. The cut-off values were decided by the fitting of a linear mixture of two Gaussian distributions over the "dcr" values. Data point with "dcr" value in the range between 0 and $\mu + 0.5\sigma$ of the first Gaussian component was considered as belonging to the first channel, and data point with "dcr" value in the range between μ

- 0.5σ of the second Gaussian component and 1.0 was considered as belonging to the second channel. The rendered MINFLUX data were exported as TIFF images and used for subsequent analysis in Fiji.

To determine whether the fluorescence signal originated from protein binding on the outside or the luminal side of the MTs, we measured the lateral width of the MT filaments in the rendered MINFLUX images. To do that, we applied a custom analysis workflow implemented in Fiji. Briefly, for a given MT filament, we first extracted the central line along its longitudinal axis. The signal intensity of the nearby regions were then plotted against its distance to the central line, and summed along the length of the filament. Thus we generated the "profile plot," similar to ImageJ's intensity profile plot, for each MT filament. We then extracted the FWHM of the intensity profile plot, as the estimation of the signal width of the given filament. A first Fiji script was created to automatically generate the central line segments from the rendered MINFLUX images. In short, it applies a line and curvilinear filter to the images to generate first an MT segmentation and then extracts the skeleton of each segmented MT filament as the central line segments. Some manual correction can be applied here to remove the bad segmentation or line segment results, but in most cases, it was not necessary. A second Fiji script was then applied to measure the FWHM of the intensity profile of each of the filaments. It reports the FWHM, as well as the length of each filaments and summarizes the results of all filaments to facilitate further statistical analysis.

### Statistical analysis

Statistical significance was analyzed using the nonparametric Mann–Whitney $U$ test (when comparing only two conditions) or the nonparametric Kruskal–Wallis test followed by Dunn's post test (when comparing multiple conditions in the same plot), as indicated in the figure legends. Additional details such as explanation, number of measurement, and precision measures can be found in the figure legends. The statistical analyses were performed using GraphPad Prism 9.

### Online supplemental material

Fig. S1 shows the characterization of CSPP-L in vitro and in cells. Fig. S2 illustrates the effects of shorter CSPP1 constructs on MT dynamics. Fig. S3 illustrates the localization of CSPP1 to MT lumen. Fig. S4 shows CSPP1 binding to MTs and its effects on MT growth and demonstrates that CSPP1 does not change protofilament length or curvature at MT plus ends. Video 1 shows the dynamics of MTs growing in vitro in the presence CSPP-L. Video 2 shows the dynamics of MTs labeled with CSPP-L and EB3 in COS-7 cells. Video 3 shows a 3D view of MTs in the presence of CSPP-L. Video 4 shows a 3D view of a MT with MAP7 as an outside ring and CSPP1 construct MTB+H5+H6 inside. Video 5 shows a photodamage of MTs in COS-7 cells overexpressing GFP-CSPP-L and β-tubulin-mCherry. Table S1 lists the key recources such as plasmids, chemicals, and analysis scripts used in this study. Data S1, S2, S3, S4, S5, S6, and S7 contain numerical data of the plots in the main and supplementary figures.

### Data availability

All data that support the conclusions are available from the authors on request, and/or also available in the supplemental material.

All cryo-ET data shown in the paper were deposited in EMDB. Accession numbers are: Fig. 4 A: control - EMD-15236; + CSPP-L - EMD-15237; + CSPP-L + VBL - EMD-15238. Fig. 4 C: + CSPP-L—EMD-15237; + CSPP-L + VBL—EMD-15239. Fig. 6 G: taxol MTs + CSPP-L, no taxol added—EMD-15245. Fig. 6.I: + VBL—EMD-15246; + VBL + CSPP-L—EMD-15247. Fig. 6 K: control—EMD-15249; + CSPP-L + ILP—EMD-15237; + CSPP-L—EMD-15237; + VBL—EMD-15248; + VBL + CSPP-L + ILP—EMD-15250; + VBL + CSPP-L—EMD-15251.

All MATLAB and Fiji Groovy scripts for MINFLUX Analysis, together with sample data and sample figures, are available online at https://github.com/EMBL-ICLM/microtubule_width_measurement_MINFLUX.

Python scripts for cryoCARE denoising are available online at https://github.com/NemoAndrea/cryoCARE-hpc04, and MATLAB scripts for analysis of protofilament shapes are available online at https://github.com/ngudimchuk/Process-PFs.

## Acknowledgments

We thank A. Jakobi (Delft University of Technology, Delft, The Netherlands) for providing access to a GPU computational cluster and K. Schink for assistance with FRAP experiments.

This work was supported by the European Research Council Synergy grant 609822 to M. Dogterom and A. Akhmanova, the ZonMW TOP 91216006 program to A. Akhmanova, the Norwegian Cancer Society Career Development grant 6839316 to S. Patzke, and QMUL Startup grant (SBC8VOL2) to V.A. Volkov. We acknowledge the access and services provided by the Imaging Centre at the European Molecular Biology Laboratory, supported by the Boehringer Ingelheim Foundation. The MINFLUX data acquisition was facilitated by the Christian Boulin Fellowship awarded to C.M. van den Berg.

Author contributions: C.M. van den Berg designed and performed protein purifications, in vitro reconstitution and cellular experiments, analyzed data, and wrote the paper; V.A. Volkov designed and performed cryo-ET experiments, analyzed data, and wrote the paper; S. Schnorrenberg, Z. Huang, and T. Zimmermann facilitated and performed MINFLUX imaging and analysis; K.E. Stecker performed and analyzed mass spectrometry experiments; I. Grigoriev performed and analyzed live cell imaging experiments using TIRF microscopy; S. Gilani and K.A.M. Frikstad generated cell lines, performed and analyzed FRAP experiments; S. Patzke performed RPE-1 IF experiments, provided reagents, and contributed to writing; M. Dogterom coordinated the project; and A. Akhmanova coordinated the project and wrote the paper.

Disclosures: The authors declare no competing interests exist.

Submitted: 12 August 2022

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

**Supplemental material**

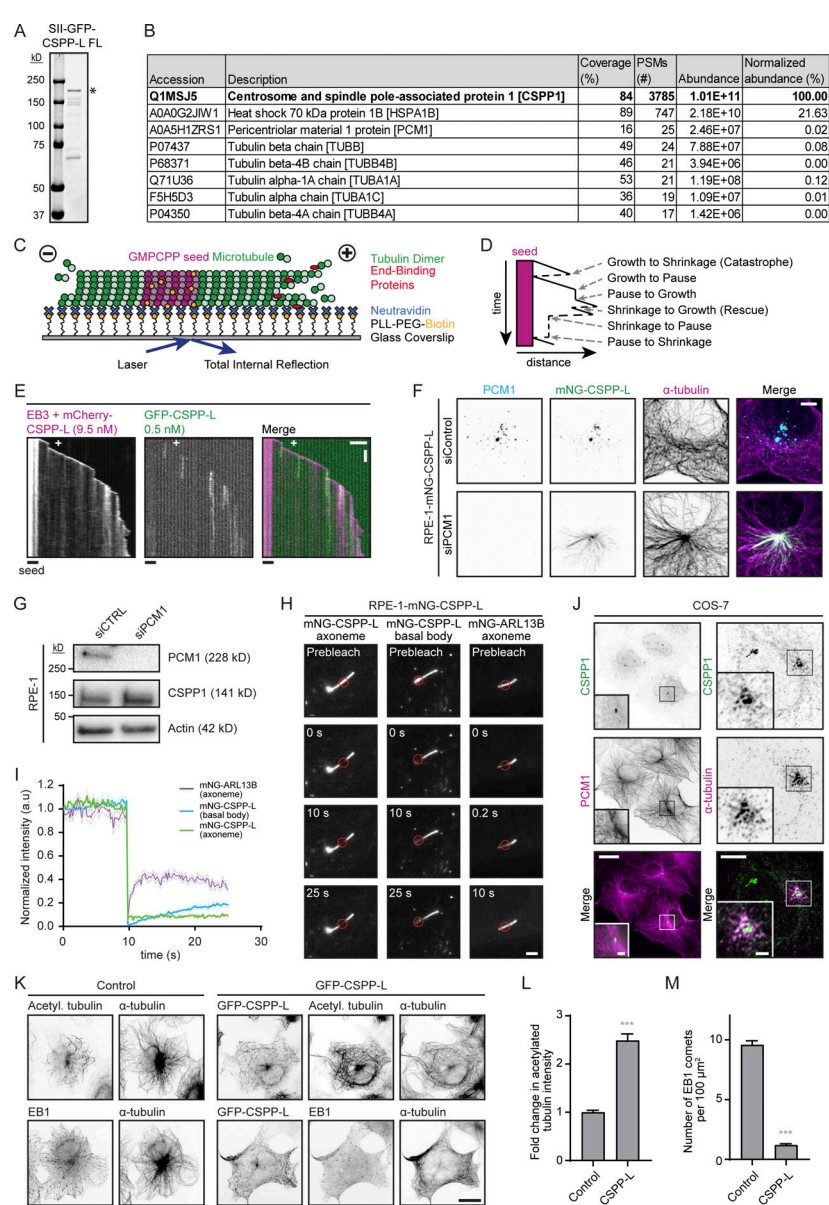

Figure S1. **Related to Fig. 1.** Characterization of CSPP-L in vitro and in cells. **(A)** Analysis of purified GFP-CSPP-L by SDS-PAGE. Asterisk indicates the full-length protein band. Protein concentrations were determined from BSA standard. **(B)** Mass spectrometry analysis of purified GFP-CSPP-L. **(C)** Schematic representation of the in vitro reconstitution assays with dynamic MTs for imaging with TIRF microscopy. GMPCPP-stabilized MT seeds containing fluorescent tubulin, such as rhodamine tubulin (for visualization) and biotinylated tubulin (for surface attachment via NeutrAvidin), are immobilized on a plasma-cleaned coverslip coated with biotinylated poly(L-lysine)-[g]-poly(ethylene glycol) (PLL-PEG-biotin), which is coupled to NeutrAvidin. MT growth from GMPCPP-stabilized seeds is initiated and visualized by the addition of tubulin supplemented with fluorescently labeled tubulin or by the addition of unlabeled tubulin combined with fluorescently-tagged EB3. MT plus- and minus-ends are indicated. **(D)** Schematic representation of a kymograph visualizing the various transition events observed and quantified in this paper. **(E)** Kymographs illustrating MT growth with 20 nM GFP-EB3 together with 9.5 nM mCherry-CSPP-L and 0.5 nM GFP-CSPP-L. Scale bars, 2 μm (horizontal) and 60 s (vertical). **(F)** Maximum projections of z-stacks of interphase RPE-1 cells expressing low levels of mNG-CSPP-L transfected with control siRNA or siRNA against PCM1 and stained for PCM1 (blue) and α-tubulin (magenta). Scale bar, 5 μm. **(G)** Western blot analysis of RPE-1 cells treated with the indicated siRNAs. **(H)** FRAP analysis of ciliated RPE-1 cells stably expressing low levels of mNG-CSPP-L or mNG-ARL13B imaged in TIRF mode on DeltaVision OMX V4 Blaze 3D-SIM. FRAP areas are indicated by the red circles. Scale bar, 2 μm. **(I)** Average normalized fluorescence intensity recovery after photobleaching of axonemes and basal bodies as shown in H. Values were normalized to the fluorescence signal in the FRAP area of the first acquired frame. Number of analyzed FRAP areas, mNG-CSPP1 axoneme, n = 15; mNG-CSPP1 basal body, n = 10; mNG-ARL13B axoneme, n = 15. Thick lines represent pooled data from three independent experiments. Light thin lines represent SEM. **(J)** Images of COS-7 cells stained for CSPP1 and α-tubulin (widefield, left) or PCM1 (confocal, right). Left scale bars, 25 and 2 μm (zoom); right scale bars, 5 and 2 μm (zoom). **(K)** Widefield fluorescence images of COS-7 cells overexpressing GFP-CSPP-L and stained for α-tubulin and acetylated tubulin or EB1. Scale bar, 20 μm. **(L)** Quantification of the mean acetylated tubulin intensity in COS-7 cells (from images as in H). The average mean intensity of cells overexpressing GFP-CSPP-L was normalized to the average mean intensity in control cells. Total number of cells analyzed: control cells, n = 137; cells overexpressing GFP-CSPP-L, n = 77. Bars represent pooled data from two independent experiments. Error bars represent SEM. ***, P < 0.001; Mann-Whitney test. **(M)** Quantification of the number of EB1 comets per 100 μm² in COS-7 cells (from images as in H). Total number of cells analyzed: control cells, n = 111; cells overexpressing GFP-CSPP-L, n = 75. Bars represent pooled data from two independent experiments. Error bars represent SEM. ***, P < 0.001; Mann–Whitney test. Source data are available for this figure: SourceData FS1.

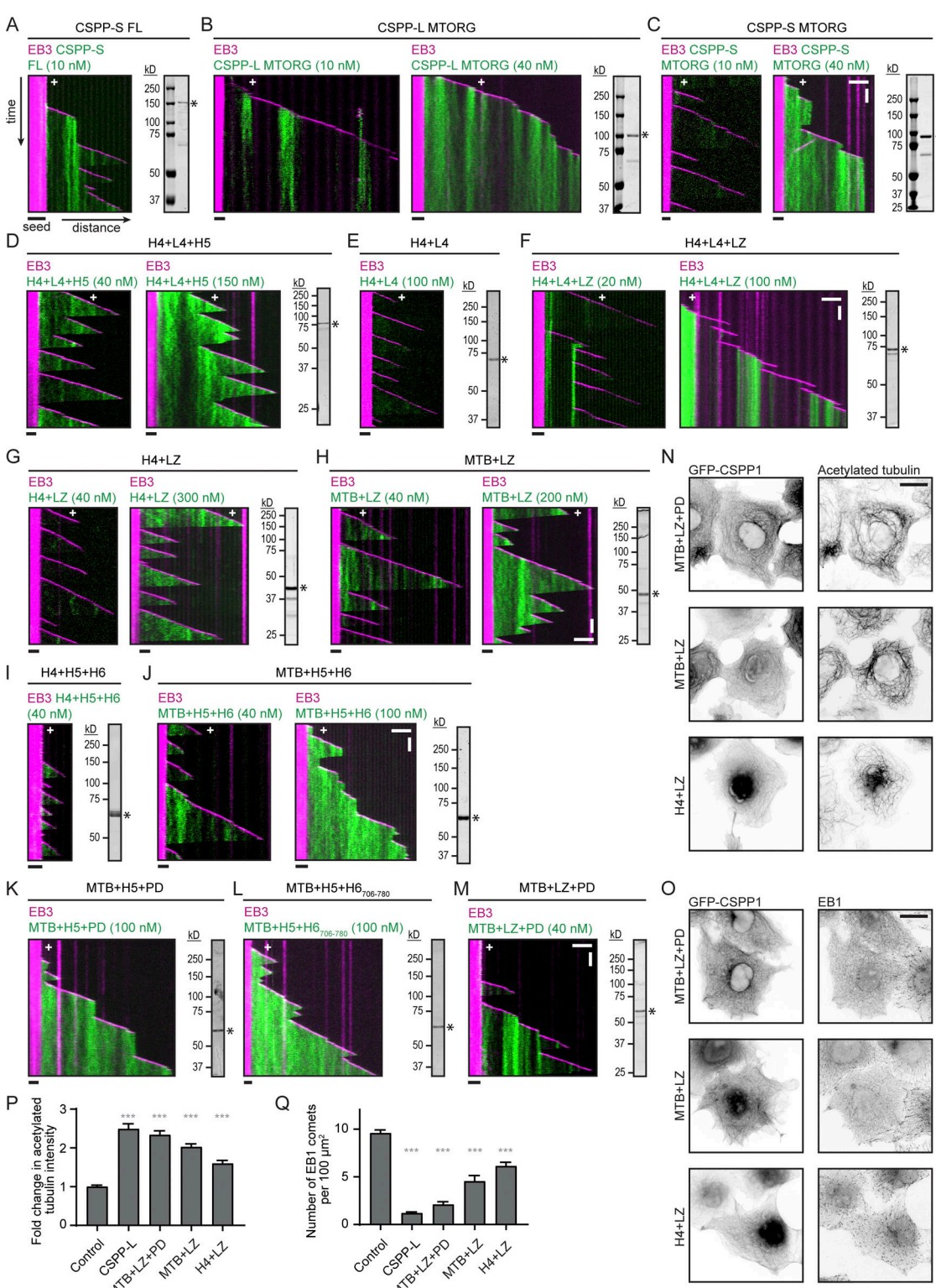

Figure S2. **Related to** Fig. 3. Shorter CSPP1 constructs are less potent in stabilizing MTs in cells. **(A–M)** Kymographs of MT growth with 20 nM mCherry-EB3 together with the indicated GFP-CSPP1 constructs at the indicated concentrations. Scale bars, 2 μm (horizontal) and 60 s (vertical). Images of SDS-PAGE gels with purified proteins are included for each construct. Asterisk indicates full-length protein band. **(N and O)** Widefield fluorescence images of COS-7 cells overexpressing GFP-CSPP-L and stained for α-tubulin and acetylated tubulin (N) or EB1 (O). Scale bar, 20 μm. **(P and Q)** Quantification of mean acetylated tubulin intensity (P) or quantification of number of EB1 comets per 100 μm² (Q) per COS-7 cell (from images as in N and O). Quantification and statistics as in S1I. Total number of cells analyzed acetylated tubulin, EB1: control cells, $n = 137$, $n = 111$; cells overexpressing GFP-CSPP-L, $n = 77$, $n = 75$; cells overexpressing GFP-MTB+LZ+PD, $n = 83$, $n = 72$; cells overexpressing GFP-MTB+LZ, $n = 70$, $n = 61$; cells overexpressing GFP-H4+LZ, $n = 50$, $n = 75$. Bars represent pooled data from two independent experiments. Data for control and GFP-CSPP-L is the same as in Fig. S1 I. Error bars represent SEM. ***, $P < 0.001$; Kruskal–Wallis test followed by Dunn's post-test. Source data are available for this figure: SourceData FS2.

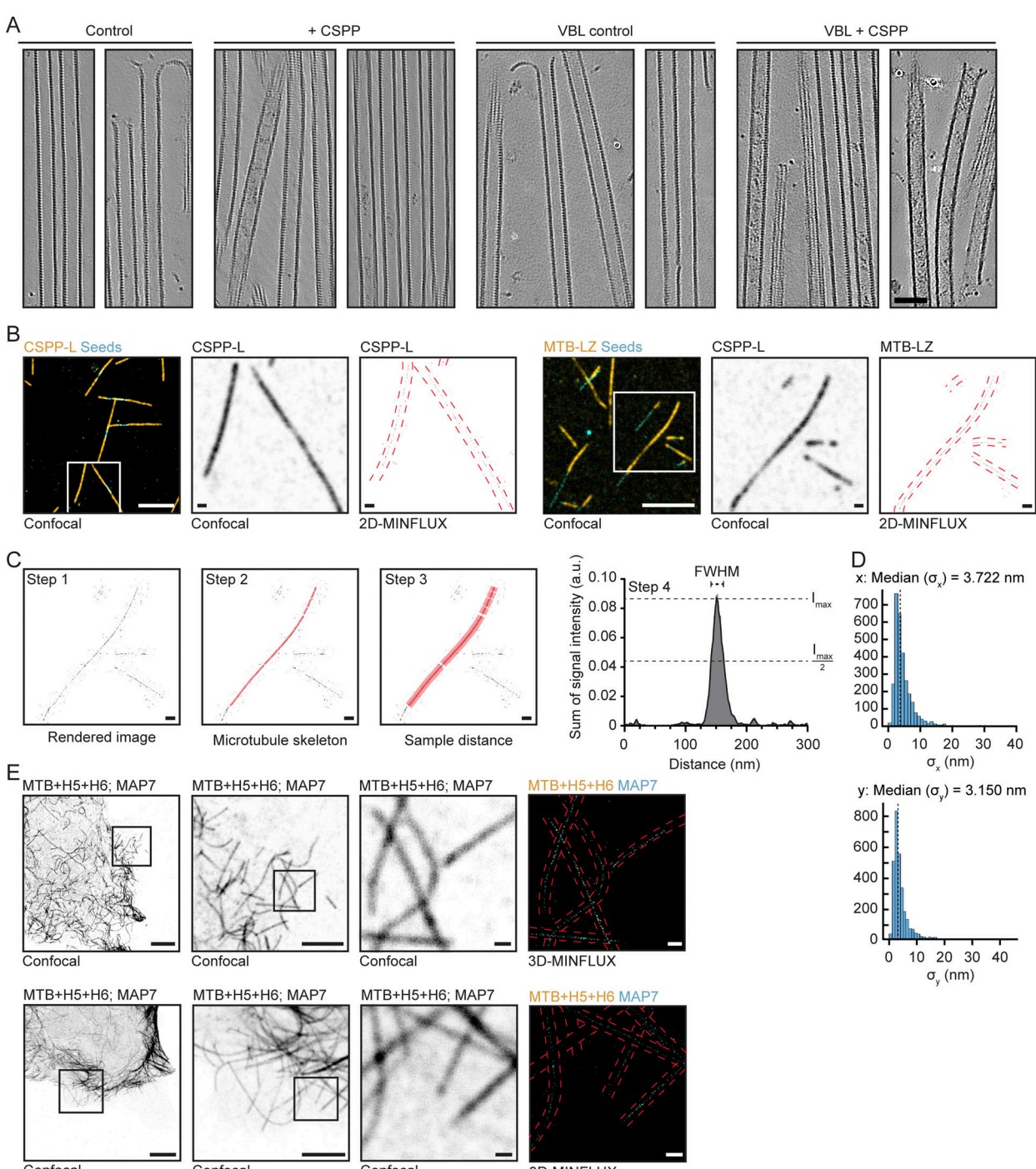

Figure S3. **Related to** Fig. 4. Characterization of CSPP1 inside the MT lumen. **(A)** Additional tomograms of dynamic MTs polymerized from GMPCPP-stabilized seeds in the presence or absence of 10 nM GFP-CSPP-L, with or without 250 nM vinblastine vitrified on EM grids. Scale bar, 50 nm. See also Fig. 4 A. **(B)** Additional zooms of confocal images of in vitro MINFLUX regions shown in Fig. 4, D–E. Large confocal field of view (FOV) and 2D-MINFLUX images are identical to the images in Fig. 4, D–E. Panel representation as in Fig. 4, D–E. Scale bars, 5 µm (Large FOV confocal image); 500 nm (zoom FOV confocal image and 2D-MINFLUX image). **(C)** Analysis workflow of FWHM value determination in the plot shown in Fig. 4 F. An MT in from in the image in Fig. 4 E was chosen as an example. First, images of MTs were rendered from MINFLUX data as density map (step 1). Then, for a given MT filament, we extracted the central line along its longitudinal axis (step 2). The signal intensities of the nearby regions were then plotted against its distance to the central line and summed along the length of the filament (step 3). We then extracted the FWHM of the intensity profile plot, as the estimation of the width of the given filament (step 4). $I_{max}$ is maximum intensity and $I_{max}/2$ is half-maximum intensity. Scale bar, 500 nm. **(D)** Standard deviation histograms (x- and y-axis) of groups of greater than four successive localizations from the same fluorophore depicting the localization precision of one example of a single color 2D-MINFLUX measurement of in vitro MTs polymerized in presence of SNAP-CSPP-L. The localization precision of all single-color 2D-MINFLUX measurements used for the FWHM analysis was in the range of 3.0 and 4.3 nm. **(E)** Dual color 3D-MINFLUX measurements of COS-7 cells overexpressing GFP-MAP7 together with SNAP-MTB-H5-H6. Panel representation as in Fig. 4, G–H. Scale bars, 10 µm (large FOV confocal image); 5 µm (medium FOV confocal image); 500 nm (small FOV confocal image, 3D-MINFLUX image and zoom); 50 nm (maximum intensity projection image).

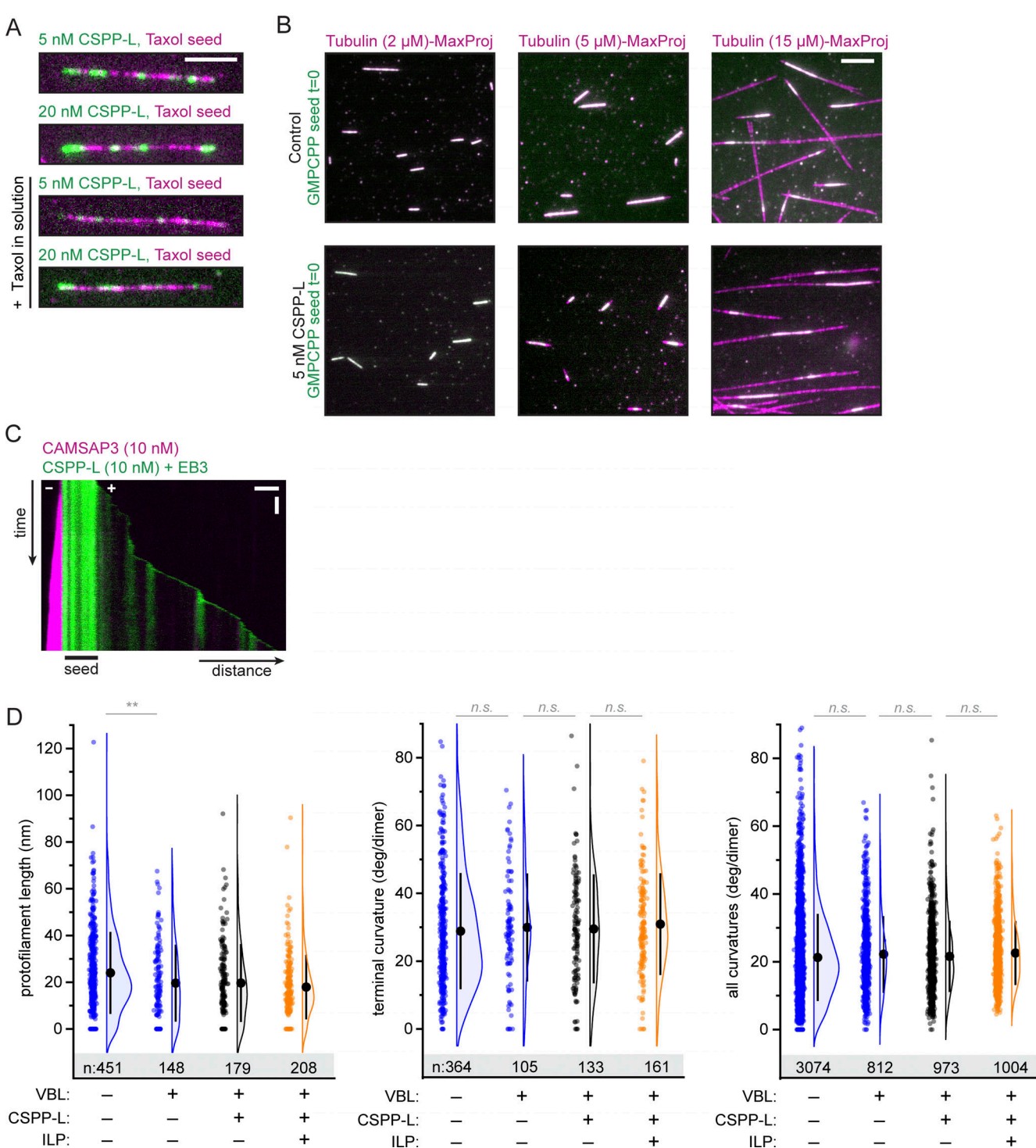

Figure S4.  **Related to** Fig. 6. CSPP1 stabilized MTs, but there is no change in protofilament length or curvature. **(A)** Still images from assays with Taxol-stabilized MT seeds in the absence or presence 40 µM Taxol and the indicated GFP-CSPP-L concentrations. Scale bar, 5 µm. See also Fig. 6 A. **(B)** MT outgrowth from GMPCPP stabilized MT seeds 5 min after flowing in 5 nM GFP-CSPP-L at the indicated tubulin concentrations. The first frame of the acquisition (green) is overlayed with the maximum projections of 5 min acquisition (magenta) illustrating the newly grown MT lattice. Scale bar, 5 µm. **(C)** Kymograph illustrating MT growth in the presence of 20 nM mCherry-EB3 together with 10 nM mCherry-CSPP-L and 10 nM GFP-CAMSAP3. Scale bars, 2 µm (horizontal) and 60 s (vertical). **(D)** Quantification of protofilament length, terminal curvature, and total curvature (from tomograms as shown in 1K). Blue, orange, and gray dots (single data points, tomograms), black circle (mean), SD (error bars). **, P < 0.01, n.s., not significant, Mann–Whitney test. Analysis from two independent experiments.

Video 1. **Dynamics of MTs growing in vitro in the presence CSPP-L.** TIRF microscopy imaging of in vitro reconstituted MTs growing from GMPCPP-stabilized MT seeds, in the presence of 15 µM tubulin (supplemented with 3% rhodamine-tubulin; magenta) and 10 nM GFP-CSPP-L (green). Sequential dual-color acquisition at 2 s per frame over the course of 10 min, displayed at 20 frames per second. Video corresponds to Fig. 1 B.

Video 2. **Dynamics of MTs labeled with CSPP-L and EB3 in COS-7 cells.** TIRF microscopy imaging of MTs in COS-7 cells overexpressing GFP-CSPP-L (green) and EB3-mCherry (magenta). Simultaneous dual-color acquisition at 100 ms per frame over the course of 50 s, displayed at 20 frames per second. Arrowheads point to the events of interest. Video corresponds to Fig. 1 G.

Video 3. **3D view of MTs in the presence of CSPP-L.** Rendering of a tomogram acquired with cryo-ET of MTs grown in vitro in the presence of GFP-CSPP-L. The denoised densities were segmented into tubulin and MTs (blue) and all other densities (orange) as described in Materials and methods. Video corresponds to Fig. 4 C.

Video 4. **3D view of a MT showing MAP7 as an outside ring with CSPP1 construct MTB+H5+H6 inside.** Rendering of a dual-color 3D-MINFLUX acquisition in COS-7 cells overexpressing SNAP-MTB+H5+H6 (orange) and GFP-MAP7 (blue). The rotational view (left) and the flythrough view (right) of the MT filament are shown. Video corresponds to Fig. 4 G.

Video 5. **Photodamage of MTs in COS-7 cells overexpressing GFP-CSPP-L and β-tubulin-mCherry.** Spinning disk confocal imaging of a photodamage experiment in COS-7 cells overexpressing GFP-CSPP-L (green) and EB3-mCherry (magenta). Simultaneous dual-color acquisition at 2 s per frame over the course of 100 s, displayed at 10 frames per second. Arrowheads point to the events of interest. Video corresponds to Fig. 5 K.

**Provided online are Table S1, Data S1, Data S2, Data S3, Data S4, Data S5, Data S6, and Data S7. Table S1 lists key resources. Data S1 shows numerical data of plots in Fig. 1 in an Excel sheet with the numerical data from the quantification of MT plus end dynamics in vitro and line scans of EB3 and CSPP-L in COS-7 cells. Data S2 shows numerical data of plots in Fig. 2 in an Excel sheet with the numerical data from line scans of Fchitax-3 and CSPP-L, the quantification of the number of CSPP-L molecules per 8 nm, intensity profiles of EB3 and CSPP-L, CSPP-L intensity quantification, and the quantification of MT plus end dynamics in vitro in presence or absence of vinblastine. Data S3 shows numerical data of plots in Fig. 3 in an Excel sheet with the numerical data from the quantification of CSPP-L intensity over time after MT binding and the quantification of MT plus end dynamics in vitro in presence of CSPP1 constructs. Data S4 shows numerical data of plots in Fig. 4 in an Excel sheet with the numerical data from the quantification the percentage of MTs containing luminal densities. Data S5 shows numerical data of plots in Fig. 5 in an Excel sheet with the numerical data from the quantifications of CSPP-L and tubulin intensity on stabilized MT seeds in presence or absence of MT damage and the quantification of the fate of newly generated MT plus ends at the cut site. Data S6 shows numerical data of plots in Fig. 6 in an Excel sheet with the numerical data from the quantifications of CSPP-L intensity on Taxol-stabilized MT seeds, seed survival, MT outgrowth, and CSPP-L intensity in presence of various tubulin concentrations, MTs with ILP densities, number of lattice defects, and plus-end raggedness. Data S7 shows numerical data of plots in Figs. S1, S2, S3, and S4 in an Excel sheet with the numerical data from mass spectrometry analysis, FRAP recovery curves, quantifications of acetylated tubulin, and number of EB3 comets in COS-7 cells and quantification of protofilament length, terminal curvature, and all curvatures.**

