## [Peer Review File · The Journal of Cell Biology]

CSPP1 stabilizes growing microtubule ends and damaged lattices from the luminal side

Cyntha van den Berg, Vladimir Volkov, Sebastian Schnorrenberg, Ziqiang Huang, Kelly Stecker, Ilya Grigoriev, Sania Gilani, Kari-Anne Myrum Frikstad, Sebastian Patzke, Timo Zimmermann, Marileen Dogterom, and Anna Akhmanova

Corresponding Author(s): Anna Akhmanova, Utrecht University

Review Timeline:

Submission Date:	2022-08-12
Editorial Decision:	2022-09-16
Revision Received:	2022-12-14
Editorial Decision:	2023-01-07
Revision Received:	2023-01-11

Monitoring Editor: Tarun Kapoor

Scientific Editor: Lucia Morgado-Palacin

Transaction Report:

DOI: <https://doi.org/10.1083/jcb.202208062>

September 16, 2022

Re: JCB manuscript #202208062

Dr. Anna Akhmanova
Utrecht University
Biology
Cell Biology, Neurobiology and Biophysics, Department of Biology, Faculty of Science Utrecht University Padualaan 8
Utrecht 3584 CH
Netherlands

Dear Dr. Akhmanova,

Thank you for submitting your manuscript entitled "CSPP1 stabilizes growing microtubule ends and damaged lattices from the luminal side". The manuscript was assessed by expert reviewers, whose comments are appended to this letter. We invite you to submit a revision if you can address the reviewers' key concerns, as outlined here.

As you will see, the reviewers appraise the high quality of the data and feel the conclusions are well supported. However, they request additional clarifications to help readers better understand the experimental design and the interpretation of data. Reviewer #2 asks for the relevance of the findings thus would like that you experimentally assess the contribution of CSPP1 to primary cilia assembly and/or activity (point 1). This reviewer would also like more evidence supporting that CSPP1 is a microtubule inner protein (point 2). In addition, reviewers #1 and #2 acknowledge that the cryo-ET data clearly shows luminal density in the presence of CSPP1. However, both have trouble interpreting the MINFLUX data and think such data do not conclusively support the luminal localization of CSPP1, thus they ask you to either show a different analysis (or tone down the results) or removing the data in question.

We hope you can address the other reviewers' points, which seem reasonable and should be feasible.

GENERAL GUIDELINES:

Text limits: Character count for an Article is < 40,000, not including spaces. Count includes title page, abstract, introduction, results, discussion, and acknowledgments. Count does not include materials and methods, figure legends, references, tables, or supplemental legends.

Figures: Articles may have up to 10 main text figures. Figures must be prepared according to the policies outlined in our Instructions to Authors, under Data Presentation, <https://jcb.rupress.org/site/misc/ifora.xhtml>. All figures in accepted manuscripts will be screened prior to publication.

*****IMPORTANT:** It is JCB policy that if requested, original data images must be made available. Failure to provide original images upon request will result in unavoidable delays in publication. Please ensure that you have access to all original microscopy and blot data images before submitting your revision. ***

Supplemental information: There are strict limits on the allowable amount of supplemental data. Articles may have up to 5 supplemental figures. Up to 10 supplemental videos or flash animations are allowed. A summary of all supplemental material should appear at the end of the Materials and methods section.

Please note that JCB now requires authors to submit Source Data used to generate figures containing gels and Western blots with all revised manuscripts. This Source Data consists of fully uncropped and unprocessed images for each gel/blot displayed in the main and supplemental figures. Since your paper includes cropped gel and/or blot images, please be sure to provide one Source Data file for each figure that contains gels and/or blots along with your revised manuscript files. File names for Source Data figures should be alphanumeric without any spaces or special characters (i.e., SourceDataF#, where F# refers to the associated main figure number or SourceDataFS# for those associated with Supplementary figures). The lanes of the gels/blots should be labeled as they are in the associated figure, the place where cropping was applied should be marked (with a box), and molecular weight/size standards should be labeled wherever possible.

Source Data Figures should be provided as individual PDF files (one file per figure). Authors should endeavor to retain a

minimum resolution of 300 dpi or pixels per inch. Please review our instructions for export from Photoshop, Illustrator, and PowerPoint here: <https://rupress.org/jcb/pages/submission-guidelines#revised>

The typical timeframe for revisions is three to four months. While most universities and institutes have reopened labs and allowed researchers to begin working at nearly pre-pandemic levels, we at JCB realize that the lingering effects of the COVID-19 pandemic may still be impacting some aspects of your work, including the acquisition of equipment and reagents. Therefore, if you anticipate any difficulties in meeting this aforementioned revision time limit, please contact us and we can work with you to find an appropriate time frame for resubmission. Please note that papers are generally considered through only one revision cycle, so any revised manuscript will likely be either accepted or rejected.

Thank you for this interesting contribution to Journal of Cell Biology. You can contact us at the journal office with any questions, cellbio@rockefeller.edu.

Sincerely,

Tarun Kapoor
Monitoring Editor
Journal of Cell Biology

Lucia Morgado-Palacin, PhD
Scientific Editor
Journal of Cell Biology

Reviewer #1 (Comments to the Authors (Required)):

CSPP1 is a protein that localizes to centrosomes and cilia. Mutations in the gene that codes for CSPP1 results in Joubert syndrome, a debilitating human ciliopathy. Recent works have identified CSPP1 as a key component of a multi-protein complex that regulates cilia structure and length, but intrinsic activities of CSPP1 on microtubule structures is poorly understood. In the submitted manuscript, van den Berg et al. investigate the effects of CSPP1 (full length isoforms, and structure-function constructs) on microtubule dynamics. The authors reconstitute microtubule dynamics in the presence of CSPP1 constructs and determine that CSPP1 best binds to slowly polymerizing microtubule ends, with a preference for binding pre-catastrophe states (preceded in time by a loss of EB protein at the microtubule plus end, which is anti-correlated with a gain in CSPP1 binding to the site). The authors find that CSPP1 induces microtubule pause, then regrowth, and map separate domains for microtubule rescue and stabilization, growth inhibition, and oligomerization. A dual approach, using cryo-ET and high resolution, fluorescence microscopy (MINFLUX), maps CSPP1 to the microtubule lumen, thus designating it a microtubule inner protein (MIP). Damaging the microtubule lattice with a laser, or using taxol to promote lattice defects during polymerization, leads to increased CSPP1 binding, suggesting that CSPP1 can enter the microtubule lumen through damage sites, bind, and promote stabilization. The work is well executed and provides key mechanistic insight into the centrosomal / ciliary protein CSPP1. The work lays a solid foundation, upon which higher-order investigations involving CSPP1 and its interactors can be conducted. The following details points that should be addressed to best make the manuscript acceptable for publication.

1. The mass spec data (Fig S1B) shows that PCM1 co-purified with CSPP1. Figure 1G shows that the proteins co-localize in cells. A) While the authors state that no microtubule regulators co-purified with CSPP1, they should state that small amounts of PCM1 did. If a CSPP1-PCM1 interaction has not been identified to date, it would benefit the field to have the authors point out the co-purification and co-localization. B) If CSPP1 localizes to lumen of centriolar microtubules, what can the authors speculate about its localization to the pericentriolar region? Would this be in microtubule minus ends?
2. Figure S1I: It would be beneficial to present the acetylated tubulin intensity per area, rather than per cell (as the authors did for EB1 comet density in S1J).
3. Figure 1H, panel 3, seems to indicate that GFP-CSPP-L is present on the kymograph trace before the microtubule grows to the that position? Can the trace be improved to remove what may be a nearby punctae of GFP-CSPP-L fluorescence?
4. The authors find that removing H5 to yield the H4+L4 construct results in loss of microtubule binding and rescue activity, but this is rescue when the GCN4 leucine zipper is appended: H4+L4+LZ, suggesting that H5 mediates oligomerization. Thus, the

authors call H5 a dimerization domain, or DD. As oligomerization, and the resulting effects of avidity are critical to the mechanism, the authors should determine the oligomeric state of a H5-containing construct. While a dimeric version rescues, the H5 region may mediate a different oligomeric state, such as trimerization or tetramerization.

5. In figure 3C, the authors present the dynamics state of microtubules in the presence of CSPP1 constructs, and show that MTB+LZ+PD can restore the general dynamics of CSPP-L, however, it does this at 10-fold higher concentration, with no pause activity noted at the comparative 10 nm concentration. The authors should make this more evident to the reader, and discuss potential causes: e.g. regions not included in the MTB+LZ+PD construct or a specific geometry of these domains is critical for full length CSPP1 activity.

6. While the cryo-ET data clearly shows luminal density in the presence of CSPP1, I found the MINFLUX data harder to interpret. A) Technically, if CSPP1 bound the exterior of the microtubule along its seam, it would give a similar result. Thus, throughout this section, the authors should refrain from stating that the MINFLUX data indicates that CSPP1 localizes to the lumen. Stating that the MINFLUX data is aligned with a luminal localization would be best. B) A max projection analysis along a length of a microtubule shown in Fig S3B for CSPP1 constructs and MAP7 in vitro would be beneficial. C) The caption for Fig 4I and J states "(2D-MINFLUX image and Zoom)" - but I believe this should be "(3D-MINFLUX...)image and Zoom)".

7. Figure 6E and F show decreased CSPP-L binding when tubulin levels are increased. The authors state that this may be due to a preference for slow growing microtubules, but it may also reflect an ability to bind free tubulin and perhaps limit the incorporation of this bound pool of tubulin. The authors should test if CSPP1 can bind free tubulin.

Reviewer #2 (Comments to the Authors (Required)):

This manuscript characterizes the effects of the CSPP1 protein on MT dynamics. CSPP1 has been predominantly studied in the context of its association with ciliopathies. CSPP1 localizes to primary cilia tips and to centrosomes. Important to this paper, cytoplasmic MT localization and regulation has not been identified. Here the authors show that, like taxol, CSPP1 localizes to the lumen of MTs and binds slow growing MTs. Moreover, CSPP1 appears to promote slow MT growth and shortening using two separate protein domains, suppressing MT cat. Finally, the authors show that CSPP1 binds to damaged MTs lattices and perhaps stabilizes them. Overall, the experiments are nicely performed, and the conclusions are well supported and this work studies a protein that likely controls MT dynamics from the MT lumen, like taxanes. A missing piece to this puzzle is whether these activities are relevant to primary cilia assembly and/or activity(s) or if it is active at cytoplasmic microtubules that experience lattice defects.

MAJOR

1. Is the activity of CSPP1 relevant to MTs in vivo? The authors argue that endogenous antibody stained CSPP1 localizes to centrosomes and centriole satellites (see below), but there is no localization to cytoplasmic MTs of this endogenous protein. The in vivo data that is presented is overexpressed protein. Moreover, no experiments are performed in the context of the primary cilium making it difficult to judge whether the findings here are relevant to the in vivo biology of CSPP1. First, I would suggest that the authors produce global MT damage and/or depolymerization in Cos7 cells to determine whether endogenous CSPP1 localizes to foci that represent new assembly. If no, localization is observed, then I'm concerned that the studies here may not represent the normal biological function of CSPP1. Second, for in vitro experiments, where the MT lattice is disrupted by laser ablation, it seems like the authors can show that a) tubulin is recruited with CSPP1 to these sites of ablation? and b) that this increases the stability of a MT that would otherwise depolymerize without CSPP1 to suppress cat? These experiments will strengthen the argument that the findings in this story are relevant to MT control in the cell. While they do not address whether this phenomenon is associated with the primary cilia, I believe they would be sufficient for strengthening the functional importance of this protein in the context of this study.

2. Is it really a MIP? The current literature of microtubule inner proteins (MIPs) is related to axoneme proteins within the lumen of the A- or B-tubule that are density mapped to a precise location and structure within the MT lattice, with a consistent repeat structure representing the MT structure. The data presented for CSPP1 clearly shows that this protein is within the lumen and does not appear to diffuse along the lattice (is somewhat stable). However, it appears that CSPP1 protein is differentially localized within the MT lumen in a manner that is not consistent with a MIP. Moreover, it appears that there is some level of aggregation that is non-uniform. I would suggest rather than defining this as a MIP, I would simply state this is a microtubule lumen localizing protein, or something to this effect. I also wonder if low CSPP1 levels could be detected as densities if uniform labeling is created in vinblastine treated samples.

3. Manuscript is written for the MT aficionado. Because JCB is a general cell biology journal, the manuscript needs to be more accessible to the general readership. I found the text and figures difficult to follow and would suggest that the authors include diagrams and explanatory models throughout the manuscript. In many places the transitions in text were not directed at testing an explicit question but rather the text required that the reader understand the potential biological implications of the performed experiments. Figure 1G-I was not clear. Kymographs need diagrams to explain them. I believe that the structure function figure could be trimmed to the relevant finding with models to explain. This text was very long with limited outcomes. The Figure 3 should have a model of the protein. Moreover, it is not clear why the splicing differences are important. Finally, in some places in the manuscript taxol addition is used to produce lattice damage and in others it is a stabilizing activity. It became confusing as to when each of these were used. For example, comparison of Figure 5A and Figure 6 use taxol for opposite functions. Finally, a

summary model of what CSPP1 is doing may help in understanding the unique differences in this protein between plus end (and minus end) and lattice regulation. I realize that this mechanism is not completely worked out, but it would be helpful to discuss a model.

4. It should be noted throughout the manuscript that the CSPP1 that is expressed is an N-terminal tagged protein.
5. Fig 1G-I is important to show that when EB3 drops off (GTP hydrolysis?) then CSPP begins loading, in vivo overexpression. However, it was difficult to follow this in the presented images? Diagrams, clearer images, and quantification of timing? Is it possible that this is not necessarily a pre-cat event but rather a loss in EB3 allows CSPP1 binding?
6. Figure 2F, it is not clear that there is a several second delay between loss in EB and gain in CSPP1. Rather they appear to be coincident events.
7. Figure 4A-C nicely shows CSPP1 is in the lumen and that this is increased with vinblastine treatment. I cannot say the same for the MINIFLUX results. The presented data in G and H were not convincing that MAP7 was outer while CSPP and MTB+LZ were inner. I would suggest removing these data or only showing I and J. Although I wasn't convincing either.
8. What is the functional significance of the terminal flare raggedness? Terminal flare activity seems weak and not supported by fact that you don't see cspp1 at these sites. It is also difficult to see how the means were generated from the presented data. Overall, all conditions seem very similar. I also wonder if the authors were to measure flare parameters when the CSPP1 is close to the end versus further from the end? Would this increase the correlation or reveal a change. Regardless, I don't find the current data convincing because most of the datapoints are all very close to ~5?

MINOR COMMENTS

1. Knowing that cilia are conserved structures, how conserved is the CSPP1 protein?
2. PCM1 and Hsp70 both have connections to MTs and their dynamics. While I agree that the activities measured in this study are likely due to CSPP1 activity, it is probably important to address and discuss potential roles that these proteins could play in MT dynamics. Perhaps the truncation mutants do not bind these components?
3. Figure 2B colors are flipped.
4. Figure 2G Vinblastin needs an "e".
5. Co-localization of CSPP1 at "centrioles" and "centriolar satellites" in Cos7 cells is not convincing. Hard to say but likely centrosomal unless imaging showing centriolar localization is presented. Moreover, is CSPP1 in the lumen of triplet MTs of these? Regardless, insets that co-localize PCM1 and CSPP1 could be used to convincingly show that it is at centriolar satellites.
6. Is it known if CSPP1 forms a dimer? I'm not sure this is relevant to the story but it is a big part of the structure function analysis. It was not clear how the authors concluded the dimerization domain?
7. Could the increased acetylation result from a more direct activity of CSPP1? Rather than simply stabilizing all MTs.
8. Because of mammalian target of rapamycin (mTOR)'s well established transcription factor roles, I would not recommend using the acronym MTOR here.
9. Page 11 "...binding to Taxol-stabilized microtubules was stronger and faster (Fig. 5A, B)." How is it known that the binding is "stronger"?

Reviewer #3 (Comments to the Authors (Required)):

The authors present a detailed and comprehensive characterization of CSPP1, a protein associated with cilia and centrosomes, and show that CSPP1 alters microtubule dynamics in a very interesting way and does so by binding to the inside of the microtubule lumen, reminiscent of taxanes. The thorough study includes in vitro and cellular microtubule dynamics characterization, along with impressive cryoET and MINIFLUX imaging that clearly show the protein inside the microtubule. The presentation is a bit overwhelming in places due to the abundance of data, but in general the presentation is clear and convincing, and the manuscript is well written other than a few grammar slips.

We had no major comments, but did have a number of points on to clarify the presentation and address a few controls.

Comments:

1. Introduction - The intro is short with a long third paragraph previewing the results. It would be nice to instead have a paragraph three on MIPs and taxanes binding to inside microtubules, which is a major point of the manuscript, and then conclude with a shorter paragraph 4 previewing results.
2. Somewhere near the start of results, the authors should include a sentence justifying use of singlet microtubules for these experiments, since it is generally known and centriolar protein.
3. The authors should describe the previous work that establishes the Microtubule organizing domain from the Centrosome targeting domain in the overall protein structure of Fig 1A.
4. Because CSPP1 alters microtubule growth rates, how do the authors ensure they are calling the proper microtubule polarity in their in vitro experiments?

5. The point that when the microtubule enters a pre-catastrophe state the affinity of CSPP goes up as the affinity for EB3 goes down is a very interesting one. However, the figures are not particularly convincing on this point, and it isn't clear why in Fig 1 it jumps from in vitro to cellular images. The authors should consider using S1F instead of Fig 1G-I to show this point. Alternatively, the in vitro data is really in fig 2, so reworking the order in the text to build the arguments could address this.
6. Page 7, 10 lines from bottom, the "mentioned previously" and "previously performed analysis" is confusing - not clear what previous you are talking about.
7. The result of 0.5 nM CSPP without versus with vinblastine in Fig 2G is a particularly impressive result.
8. The MCAK experiment in Fig 2K is complicated because of all of the competing processes going on (catastrophe, pausing, etc). The text gives few details, and the addition of the MCAK point as written confuses the arguments rather than clarifying. The authors should either expand on the MCAK point to develop it, or they should remove it to streamline the arguments and put it in the supplemental.
9. Relevant to Fig. 3A, it isn't clear how it is determined that full-length is a dimer. Presumably this is from the alpha-fold structure. Related to this is that the authors should show their alpha-fold structure, at least in the supplement. And further description is needed to clarify how they determine that the DD domain is in fact a dimerization domain.
10. Fig. 4 is beautiful. One suggestion - it would be nice to see the distributions from D and E that are used to calculate the FWHM in F. Additionally, it would be nice to see in I and J the distribution of distances from the centroid rather than a slice - this would be more comprehensive analysis of the data.
11. The authors should more carefully define what they mean by "taper", "flare" and "raggedness".
12. Page 11: "...Taxol-stabilized microtubules, which are known to acquire extensive lattice defects when incubated without soluble tubulin...". This is a strong statement, and it is referenced, but the authors should expand on this point to clarify their precise meaning and not cast aspersion on all taxol-stabilized. (It is not actually "well known".)
13. Fig 6 E&F and bottom page 12 "This suggests that CSPP-L binds to microtubules more efficiently when they grow slowly, either due to a slow on-rate...". This is a faulty kinetic argument because if it has a slow on-rate, then the difference is simply the duration the microtubules are present under the two conditions, and so the efficiency of binding is not different, only the duration allowed for binding is different.
14. There is a controversy in the microtubule dynamics field about whether plus-ends grow as sheets (the textbook explanation) or as isolated, curved protofilaments that straighten and laterally associate (the McIntosh model). The authors are remarkably understated in addressing this controversy, while presenting extremely strong data showing curved protofilaments that avoid using the somewhat controversial McIntosh protofilament tracing approach. It is understandable if the authors want to stay above the fray, but they should more deliberately point more their clear evidence of curved protofilaments at the growing plus-end. It would even be justified to incorporate the very interesting Fig. S4D results into Fig. 6.
15. The reference list uses inconsistent formatting and needs to be cleaned up.

Point-by-point response to the reviewer comments on the manuscript

“CSPP1 stabilizes growing microtubule ends and damaged lattices from the luminal side” by van den Berg et al.

We thank the reviewers for the constructive and helpful feedback. We revised the manuscript in light of their comments, as described below.

Reviewer #1 (Comments to the Authors (Required)):

CSPP1 is a protein that localizes to centrosomes and cilia. Mutations in the gene that codes for CSPP1 results in Joubert syndrome, a debilitating human ciliopathy. Recent works have identified CSPP1 as a key component of a multi-protein complex that regulates cilia structure and length, but intrinsic activities of CSPP1 on microtubule structures is poorly understood. In the submitted manuscript, van den Berg et al. investigate the effects of CSPP1 (full length isoforms, and structure-function constructs) on microtubule dynamics. The authors reconstitute microtubule dynamics in the presence of CSPP1 constructs and determine that CSPP1 best binds to slowly polymerizing microtubule ends, with a preference for binding pre-catastrophe states (preceded in time by a loss of EB protein at the microtubule plus end, which is anti-correlated with a gain in CSPP1 binding to the site). The authors find that CSPP1 induces microtubule pause, then regrowth, and map separate domains for microtubule rescue and stabilization, growth inhibition, and oligomerization. A dual approach, using cryo-ET and high resolution, fluorescence microscopy (MINFLUX), maps CSPP1 to the microtubule lumen, thus designating it a microtubule inner protein (MIP). Damaging the microtubule lattice with a laser, or using taxol to promote lattice defects during polymerization, leads to increased CSPP1 binding, suggesting that CSPP1 can enter the microtubule lumen through damage sites, bind, and promote stabilization. The work is well executed and provides key mechanistic insight into the centrosomal / ciliary protein CSPP1. The work lays a solid foundation, upon which higher-order investigations involving CSPP1 and its interactors can be conducted. The following details points that should be addressed to best make the manuscript acceptable for publication.

1. The mass spec data (Fig S1B) shows that PCM1 co-purified with CSPP1. Figure 1G shows that the proteins co-localize in cells. A) While the authors state that no microtubule regulators co-purified with CSPP1, they should state that small amounts of PCM1 did. If a CSPP1-PCM1 interaction has not been identified to date, it would benefit the field to have the authors point out the co-purification and co-localization. B) If CSPP1 localizes to lumen of centriolar microtubules, what can the authors speculate about its localization to the pericentriolar region? Would this be in microtubule minus ends?

Reply: A) The interaction between CSPP1 and PCM1 has been shown previously (Frikstad et al., 2019; PMID: 31412255), and we modified the text on page 5 to point this out. CSPP1 localization to centriolar satellites and its interactions with centriolar satellite proteins detected by copurification or proximity proteomics were additionally reported in several independent centrosome/cilia-related and large-scale proteomic studies; examples include but are not limited to: Gupta et al., 2015 (PMID:26638075); Shearer et al., 2018 (PMID: 29742019); Gheiratmand et al.; 2019 (PMID:31304627); Go et al., 2021 (PMID:34079125). In the revised manuscript, we have included new images (Fig. S1F, J), illustrating the colocalization of CSPP1 with centriolar satellites.

B) In RPE1 cells expressing near-endogenous levels of mNeonGreen(mNG)-CSPP1, the protein localized to the centrioles and centriolar satellites and did not localize to cytoplasmic microtubules (Frikstad et al., 2019; PMID: 31412255), and we show similar data in the new Fig. S1F. We do not know whether CSPP1 localizes to the lumen of centriolar microtubules, because so far, we and others did not obtain sufficient resolution of CSPP1 labeling on centrioles to address this question. There is also no evidence that CSPP1 would localize to microtubule minus ends in the pericentriolar region: CSPP1 localizes to

centriolar satellites, which are membraneless granules that play a role in different cellular processes but do not nucleate microtubules and do not colocalize with microtubule minus ends. However, depletion of PCM1, which inhibits satellite formation, led to relocalization of CSPP1 to cytoplasmic microtubules (new Fig. S1F). These results suggest that centriolar satellites can sequester CSPP1 from microtubules, and the abundance of centriolar satellites might control the localization of CSPP1 to microtubules. CSPP1 localization to centriolar satellites might be a way of concentrating and storing the protein near centrioles for cilia formation.

2. Figure S1I: It would be beneficial to present the acetylated tubulin intensity per area, rather than per cell (as the authors did for EB1 comet density in S1J).

Reply: There was a mistake in the figure legend, which we have corrected. In this panel (now Fig. S1L), we show the mean acetylated tubulin intensity, where the total intensity per cell is divided by the cell area.

3. Figure 1H, panel 3, seems to indicate that GFP-CSPP-L is present on the kymograph trace before the microtubule grows to the that position? Can the trace be improved to remove what may be a nearby punctae of GFP-CSPP-L fluorescence?

Reply: We apologize for the confusion. The green signal represented a long pause event of a microtubule end, and the magenta EB3 comet was background signal. We have changed this kymograph and added their schematic representations with additional annotations to avoid confusion (modified Fig. 1G,H).

4. The authors find that removing H5 to yield the H4+L4 construct results in loss of microtubule binding and rescue activity, but this is rescue when the GCN4 leucine zipper is appended: H4+L4+LZ, suggesting that H5 mediates oligomerization. Thus, the authors call H5 a dimerization domain, or DD. As oligomerization, and the resulting effects of avidity are critical to the mechanism, the authors should determine the oligomeric state of a H5-containing construct. While a dimeric version rescues, the H5 region may mediate a different oligomeric state, such as trimerization or tetramerization.

Reply: In the original version of the paper, we had not investigated the oligomerization state of CSPP-L and its fragments to support our dimerization claim. We have now compared the intensity of CSPP-L molecules upon initial binding to microtubules to the intensity of single GFP molecules on the same coverslip in another chamber and found that CSPP-L molecules initially bound to microtubules as monomers (new Fig. 3B). Interestingly, CSPP-L signal increased to an average of two molecules within 25 seconds after initial binding, indicating that CSPP-L binds cooperatively and potentially dimerizes (new Fig. 3C). If H5 would facilitate cooperative binding, similar results would be expected for H4+L4+H5 while no increase in signal would be expected for H4+L4. Unfortunately, H4+L4 strongly diffused along the microtubule lattice precluding precise quantification of the signal over time (Fig. 3D). Diffusion was less pronounced in the presence of H4+L4+H5, indicating that H5 could decrease lateral mobility (Fig. 3D), similar full-length CSPP-L, which shows very little lateral mobility. Thus, H5 reduces lateral mobility, potentially by facilitating cooperative binding of CSPP-L molecules on microtubules. Because we could not conclusively prove that H5 indeed causes dimerization, we removed the term “DD” and renamed the constructs accordingly.

5. In figure 3C, the authors present the dynamics state of microtubules in the presence of CSPP1 constructs and show that MTB+LZ+PD can restore the general dynamics of CSPP-L, however, it does this at 10-fold higher concentration, with no pause activity noted at the comparative 10 nm concentration. The authors should make this more evident to the reader, and discuss potential causes:

e.g. regions not included in the MTB+LZ+PD construct or a specific geometry of these domains is critical for full length CSPP1 activity.

Reply: We fully agree that additional regions with weak microtubule affinity and/or the specific geometry of these domains likely explain the difference between the full-length protein and the shorter construct. We have added the potential reasons for this difference more clearly in the text, page 10.

6. While the cryo-ET data clearly shows luminal density in the presence of CSPP1, I found the MINFLUX data harder to interpret. A) Technically, if CSPP1 bound the exterior of the microtubule along its seam, it would give a similar result. Thus, throughout this section, the authors should refrain from stating that the MINFLUX data indicates that CSPP1 localizes to the lumen. Stating that the MINFLUX data is aligned with a luminal localization would be best. B) A max projection analysis along a length of a microtubule shown in Fig S3B for CSPP1 constructs and MAP7 in vitro would be beneficial. C) The caption for Fig 4I and J states "(2D-MINFLUX image and Zoom)" - but I believe this should be "(3D-MINFLUX...)image and Zoom)".

Reply: A) We agree with this statement and modified the text on pages 11-12 accordingly. B) The dataset was acquired as a 2D-dataset, and therefore, a max projection analysis would not be meaningful. We have removed the panels with in vitro reconstitutions of CSPP-L + MAP7, as was suggested by another reviewer. C) This was indeed a mistake, which we have now corrected.

7. Figure 6E and F show decreased CSPP-L binding when tubulin levels are increased. The authors state that this may be due to a preference for slow growing microtubules, but it may also reflect an ability to bind free tubulin and perhaps limit the incorporation of this bound pool of tubulin. The authors should test if CSPP1 can bind free tubulin.

Reply: It seems very unlikely that decreased CSPP-L binding to microtubules caused by a three-fold increase in free tubulin levels reflects the ability of the protein to bind to free tubulin dimers. In Fig. 1D, we show that 0.5 nM CSPP-L is able to form strong accumulations on the microtubule lattice in the presence of 15 μ M tubulin. In this assay, free tubulin is present in a 30,000-fold molar excess compared to CSPP-L, indicating that it is unlikely that CSPP1 can be sequestered by free tubulin from binding to microtubules.

Reviewer #2 (Comments to the Authors (Required)):

This manuscript characterizes the effects of the CSPP1 protein on MT dynamics. CSPP1 has been predominantly studied in the context of its association with ciliopathies. CSPP1 localizes to primary cilia tips and to centrosomes. Important to this paper, cytoplasmic MT localization and regulation has not been identified. Here the authors show that, like taxol, CSPP1 localizes to the lumen of MTs and binds slow growing MTs. Moreover, CSPP1 appears to promote slow MT growth and shortening using two separate protein domains, suppressing MT cat. Finally, the authors show that CSPP1 binds to damaged MTs lattices and perhaps stabilizes them. Overall, the experiments are nicely performed, and the conclusions are well supported and this work studies a protein that likely controls MT dynamics from the MT lumen, like taxanes. A missing piece to this puzzle is whether these activities are relevant to primary cilia assembly and/or activity(s) or if it is active at cytoplasmic microtubules that experience lattice defects.

MAJOR

1. Is the activity of CSPP1 relevant to MTs in vivo? The authors argue that endogenous antibody stained CSPP1 localizes to centrosomes and centriole satellites (see below), but there is no localization to cytoplasmic MTs of this endogenous protein. The in vivo data that is presented is overexpressed protein. Moreover, no experiments are performed in the context of the primary cilium making it difficult to judge whether the findings here are relevant to the in vivo biology of CSPP1. First, I would suggest that the authors produce global MT damage and/or depolymerization in Cos7 cells to determine whether endogenous CSPP1 localizes to foci that represent new assembly. If no, localization is observed, then I'm concerned that the studies here may not represent the normal biological function of CSPP1. Second, for in vitro experiments, where the MT lattice is disrupted by laser ablation, it seems like the authors can show that a) tubulin is recruited with CSPP1 to these sites of ablation? and b) that this increases the stability of a MT that would otherwise depolymerize without CSPP1 to suppress cat? These experiments will strengthen the argument that the findings in this story are relevant to MT control in the cell. While they do not address whether this phenomenon is associated with the primary cilia, I believe they would be sufficient for strengthening the functional importance of this protein in the context of this study.

Reply: Under native conditions in vivo, CSPP1 binds to microtubules in cilia but not to cytoplasmic microtubules. It is extremely difficult to obtain mechanistic insight into microtubule-protein interactions in cilia in vivo. This is mainly due to the resolution limit in fluorescence microscopy, slow ciliary growth and the complicated nature of protein complexes controlling cilia formation. Previous studies have generated data about CSPP1 localization in cilia and demonstrated that CSPP1 participates in controlling cilia outgrowth (Frikstad et al., 2019; PMID: 31412255). These studies implied a role of CSPP1 in cilia/axonemal microtubule stabilization, but no mechanistic insights were obtained. To get a better understanding of the interaction between CSPP1 and microtubules, in this paper, we used an in vitro reconstitution approach. This allowed us to study the interaction in a very controlled way. We found that CSPP1 stabilizes in vitro reconstituted microtubules by inducing pauses and preventing depolymerization. This aligns nicely with the role of CSPP1 in cilia stability deduced from the CSPP1 depletion data. The detailed analysis of CSPP1 function at the ciliary tip, where it acts, will be challenging, not least because it works in a complex with several other microtubule-associated proteins, such as TOGARAM1 and CEP104 (Latour et al., 2020; PMID: 32453716). Such analysis would require dissection of individual and combined activities of these proteins, which goes far beyond the scope of the current study.

Still, we aimed to examine whether the behavior of CSPP1 that we observed in vitro could also be observed in a cellular environment. We now included data showing that very low (close to endogenous) levels of mNG-CSPP-L (Frikstad et al., 2019; PMID: 31412255) can localize to cytoplasmic microtubules, if its sequestration in centriolar satellites is blocked by depleting its binding partner

PCM1 (new Fig. S1F, G). We then used mildly overexpressed GFP-CSPP-L to show that also in cells, CSPP-L is a microtubule stabilizer that localizes to well-confined accumulations. We have now also added FRAP data of cilia in RPE1 cells with expressing mNG-CSPP1 (Fig. S1H, I), which show that mNG-CSPP1 does not recover at all after photobleaching while recovery of ciliary marker ARL13B is very fast. Absence of recovery after photobleaching indicates low protein turnover, indicating that it is stably bound. This aligns well with our in vitro data where we see well-confined CSPP1 localization.

We addressed the second point by performing additional experiments where we induced damage of dynamic microtubules in vitro by laser illumination. We showed that CSPP1 is recruited to these damage sites, and that tubulin can be recruited to these sites as well (new Fig. 5F-H). Additionally, when microtubules were fully severed in the presence of CSPP1, the newly generated plus end rarely switched to depolymerization, while depolymerization occurred frequently when CSPP1 was absent, as is expected from literature (Walker et al., 1989, PMID: 2921286) (new Fig. 5I, J). These results suggest that CSPP1 recognizes damaged microtubules, stabilizes them by preventing depolymerization and possibly facilitates microtubule repair through tubulin incorporation.

2. Is it really a MIP? The current literature of microtubule inner proteins (MIPs) is related to axoneme proteins within the lumen of the A- or B-tubule that are density mapped to a precise location and structure within the MT lattice, with a consistent repeat structure representing the MT structure. The data presented for CSPP1 clearly shows that this protein is within the lumen and does not appear to diffuse along the lattice (is somewhat stable). However, it appears that CSPP1 protein is differentially localized within the MT lumen in a manner that is not consistent with a MIP. Moreover, it appears that there is some level of aggregation that is non-uniform. I would suggest rather than defining this as a MIP, I would simply state this is a microtubule lumen localizing protein, or something to this effect. I also wonder if low CSPP1 levels could be detected as densities if uniform labeling is created in vinblastine treated samples.

Reply: We agree that at present we do not have information on whether CSPP1 localizes repetitively in the microtubule structure like the structural MIPs that have been identified in the axoneme so far. We did not perform structural studies with sufficient resolution to show whether CSPP1 localizes in a regular pattern. Therefore, in the revised text we refer to CSPP1 as an intraluminal microtubule-binding protein.

3. Manuscript is written for the MT aficionado. Because JCB is a general cell biology journal, the manuscript needs to be more accessible to the general readership. I found the text and figures difficult to follow and would suggest that the authors include diagrams and explanatory models throughout the manuscript. In many places the transitions in text were not directed at testing an explicit question but rather the text required that the reader understand the potential biological implications of the performed experiments. Figure 1G-I was not clear. Kymographs need diagrams to explain them. I believe that the structure function figure could be trimmed to the relevant finding with models to explain. This text was very long with limited outcomes. The Figure 3 should have a model of the protein. Moreover, it is not clear why the splicing differences are important. Finally, in some places in the manuscript taxol addition is used to produce lattice damage and in others it is a stabilizing activity. It became confusing as to when each of these were used. For example, comparison of Figure 5A and Figure 6 use taxol for opposite functions. Finally, a summary model of what CSPP1 is doing may help in understanding the unique differences in this protein between plus end (and minus end) and lattice regulation. I realize that this mechanism is not completely worked out, but it would be helpful to discuss a model.

Reply: A diagram explaining kymographs is included in Fig. S1D; we indicated the time and distance axes in Fig. 1C and provided additional annotations in other kymographs of in vitro experiments. We

have done our best to improve the text transitions to better explain the logic, improved Fig. 1G-I and added schemes and better annotations to the kymographs shown in Fig. 1H. We did not add the AlphaFold model of the protein, because due to many disordered regions, it is not informative beyond delineating the regions that are likely to be helical, which are indicated in Figure 3A. Further, we prefer to keep the data and analysis in Fig. 3, because it is one of the first examples where an intraluminal microtubule-binding protein is dissected, and the identification of targeting domains will be useful, for example, for researchers who would like to target different markers or activities (e.g. modifying enzymes) into the microtubule lumen. As to splicing, there was indeed a strong tendency in the cell biology field to ignore the splice isoforms, but with the advent of deep RNAseq-based analysis, this trend is changing. In case of CSPP1, we see some effects of alternatively spliced protein parts on microtubule binding and wish to report them, because this information could be useful for future functional studies. Furthermore, we have explained the use of Taxol-stabilized microtubules for different purposes more clearly in the text (e.g., pages 12, 13 and 14). Finally, a summary model has been included in a new Fig. 7.

4. It should be noted throughout the manuscript that the CSPP1 that is expressed is an N-terminal tagged protein.

Reply: In all figure legends, we have indicated clearly which tag was used and where it was located (GFP-CSPP-L for N-terminal tagged CSPP-L). When we discuss overexpression of DNA constructs in cells, we indicated the tag and its localization in the main text. However, we chose not to do this when discussing in vitro experiments because in our opinion this would decrease the readability and we occasionally switch from GFP to mCherry. To stress that we use N-terminally tagged constructs, we have now indicated this clearly at the beginning of the results and before we go into detail about the truncated CSPP1 variants (pages 5 and 8).

5. Fig 1G-I is important to show that when EB3 drops off (GTP hydrolysis?) then CSPP begins loading, in vivo overexpression. However, it was difficult to follow this in the presented images. Diagrams, clearer images, and quantification of timing? Is it possible that this is not necessarily a pre-cat event but rather a loss in EB3 allows CSPP1 binding?

Reply: In this figure, our intention was to show that GFP-CSPP1 in cells behaves in a way similar to that observed in our in vitro reconstitution assays. We have added schemes to Figure 1H to help the reader understand the kymographs. EB3 binds to growing microtubule ends because it recognizes the GTP cap at microtubule plus ends. The pre-catastrophe phase is characterized by the loss of the GTP cap, which leads to the loss of binding sites for EB3. If EB3 would simply prevent CSPP1 from binding to the microtubule tip, we would expect that CSPP1 would bind homogeneously to the microtubules in the absence of EB3. We did not observe this in our in vitro reconstitutions (Fig. 1C), as also with tubulin alone, we detect distinct accumulations, similar to what we observe in presence of EB3.

6. Figure 2F, it is not clear that there is a several second delay between loss in EB and gain in CSPP1. Rather they appear to be coincident events.

Reply: We agree that the delay is not always detectable, and the two events can appear concomitantly, and modified the text accordingly (page 7).

7. Figure 4A-C nicely shows CSPP1 is in the lumen and that this is increased with vinblastine treatment. I cannot say the same for the MINFLUX results. The presented data in G and H were not convincing that MAP7 was outer while CSPP and MTB+LZ were inner. I would suggest removing these data or only showing I and J. Although I wasn't convincing either.

Reply: We agree with this comment, and we removed panels G, H and I, because they were not sufficiently convincing due to low protein density. However, we did get high quality data for the smaller CSPP1 construct, MTB+H5+H6, and are convinced that this fragment localizes in the microtubule lumen (Fig. 4G, H in the revised manuscript). We note that the localization of this deletion mutant was the most important goal of the MINFLUX experiments, because smaller fragments are more difficult to detect by cryo-ET.

8. *What is the functional significance of the terminal flare raggedness? Terminal flare activity seems weak and not supported by fact that you don't see cspp1 at these sites. It is also difficult to see how the means were generated from the presented data. Overall, all conditions seem very similar. I also wonder if the authors were to measure flare parameters when the CSPP1 is close to the end versus further from the end? Would this increase the correlation or reveal a change. Regardless, I don't find the current data convincing because most of the datapoints are all very close to ~5?*

Degree of raggedness, or tapering, was previously correlated to dynamicity of microtubules, with more tapered ends being characteristic of faster growing microtubules, or pre-catastrophe states (Chretien et al., 1995, PMID: 7775577; Coombes et al., 2013, PMID: 23831290; Duellberg et al., 2016, PMID: 27489342; Gudimchuk et al., 2020, PMID: 32724196). Our analysis presented in Fig. 6K-M indicates that the presence of luminal CSPP-L particles next to the taper correlates with the absence of long tapers, consistent with the TIRF data on slow growth and absence of catastrophes in the presence of CSPP-L. If indeed long tapers are characteristic of a pre-catastrophe state, these events in a population are rare, which is consistent with a very skewed distribution of taper lengths that we observe in tomograms (Fig. 6M). Based on these data, we conclude that the presence of CSPP-L density close to microtubule end has an effect on the presence of long tapers, but not on any parameters of protofilament shapes (compare black and orange data in Fig. 6M and Fig. S4D – black for CSPP1 present, but terminal ILP absent, orange for CSPP1 and terminal ILP density present). We added this information to the text to better explain our interpretation of the data (page 15).

MINOR COMMENTS

1. *Knowing that cilia are conserved structures, how conserved is the CSPP1 protein?*

Reply: CSPP1 is evolutionary highly conserved in eukaryotes which have cilia/flagella. CSPP1 orthologues are reported in 113 species (EggNOG5.0, EggNOG Database | Orthology predictions and functional annotation (embl.de)). Orthologues of furthest evolutionary distance include the monoflagellates *Micromonas pusicella* (Chlorophyta), and *Monosiga brevicollis* and *Salpingoeca rosetta* (Choanoflagellata) – progenitors of metazoan life. Notably, no obvious orthologues are yet detected in the “classical” flagella model organism of *Chlorophyta*, *Chlamydomonas*. Following the evolution of multicellular life, CSPP1 orthologues are found in the non-bilaterian animals, Cnidaria (*Nematostella vectensis*). Interestingly, within the bilaterian branch, no orthologues are reported in the genomes of arthropods or nematodes, while within deuterostomes examples are found in urochordata (*Ciona intestinalis*) and across all vertebrates. This phylogenetic pattern is not unique to CSPP1. In fact, a thorough phylogenetic profiling analysis of ciliary genes not including CSPP1, identified a “lost-in-ecdysozoa” cluster, which is particularly enriched in genes associated with GO terms “microtubule cytoskeleton” and “centrosome” (Nevers et al., 2017, PMID: 28460059). This cluster includes IFT25 and IFT27 of the IFT-B complex, and LZTFL1 of the BBSome complex, which are implied in the function of the Sonic Hedgehog (SHh) pathway, which is primarily cilia-dependent in vertebrates and either lost or mostly cilia-independent in *C.elegans* or *D.melanogaster*. Previous studies implicated CSPP1 in centrosome, cilia and Hedgehog signaling function, and the study presented here strongly suggests that this function particularly involves the stabilization and maintenance of microtubules. This is consistent with a role of CSPP1 in stabilization and maintenance

of ciliary microtubules and organization of the ciliary tip for reception and transduction of the SHH-signalling pathway. Detailed evolutionary analysis of CSPP1 goes beyond the scope of this paper, but we now mention some evolutionary aspects in the Introduction (page 3).

2. PCM1 and Hsp70 both have connections to MTs and their dynamics. While I agree that the activities measured in this study are likely due to CSPP1 activity, it is probably important to address and discuss potential roles that these proteins could play in MT dynamics. Perhaps the truncation mutants do not bind these components?

Reply: In the literature, there is indeed evidence for disorganization of the microtubule network upon PCM1 depletion (Dammermann and Merdes, 2002, PMID: 12403812). PCM1 is a core component of centriolar satellites, whereby it concentrates many proteins in the pericentriolar region, among which proteins that are directly interacting with microtubules. However, PCM1 has not been localized on microtubules and it does not have a microtubule-binding domain. The existing evidence suggests that PCM1 does not directly affect microtubule dynamics, but it concentrates proteins which do. Therefore, we believe that PCM1 does not play a role in our in vitro reconstitutions, especially as the major effects of CSPP1 on microtubules can be reproduced with short fragments of the protein that do not localize to the centrosome and centriolar satellites because they lack the C-terminal domain responsible for this PCM1-dependent localization.

Heat shock proteins are among the most common contaminants in protein purification experiments. Over the years, we have purified many different proteins that differently affect microtubule dynamics, by either stabilizing or destabilizing them at nanomolar concentrations (for example, CSPP1 and MCAK shown in this paper), and we always observed some level of Hsp70 contamination. If Hsp70 contamination would play a significant role in microtubule dynamics, the effect of various purified proteins would not be so different. Therefore, we don't expect that Hsp70 plays a role in our in vitro assays. Additionally, we have not found any literature describing the direct effect of Hsp70 on microtubule dynamics in in vitro reconstitution assays. This is now briefly mentioned on page 5.

3. Figure 2B colors are flipped.

Reply: We corrected the figure.

4. Figure 2G Vinblastin needs an "e".

Reply: We apologize, this mistake has been corrected.

5. Co-localization of CSPP1 at "centrioles" and "centriolar satellites" in Cos7 cells is not convincing. Hard to say but likely centrosomal unless imaging showing centriolar localization is presented. Moreover, is CSPP1 in the lumen of triplet MTs of these? Regardless, insets that co-localize PCM1 and CSPP1 could be used to convincingly show that it is at centriolar satellites.

Reply: We agree that the image was not very convincing. We acquired new images on another microscope and replaced the previous image with a more convincing one (Fig. S1J). Here, clear localization to the centrioles and centriolar satellites is visible. Additionally, we show that CSPP1 colocalizes with centriolar satellites in RPE-1 cells with low expression of mNG-CSPP1 (new Fig. S1F). We have no data that would allow us to prove that CSPP1 is within the lumen of triplet microtubules. CSPP1 might localize to the outside of the triplet MTs through binding to one of its partners. The C-terminal tail of CSPP1 was previously determined to be required for centrosomal localization and the isolated domain also localized to the centrosome (Patzke et al., 2006, PMID: 16826565), where it might bind to the outer surface of the triplets.

6. Is it known if CSPP1 forms a dimer? I'm not sure this is relevant to the story but it is a big part of the structure function analysis. It was not clear how the authors concluded the dimerization domain?

Reply: This point was also raised by the other reviewers. We have addressed this question under point 4 of reviewer 1. We found that CSPP1 binds microtubules as a monomer but then appears to dimerize with some delay (new Fig. 3B, C). This potentially cooperative mechanism is interesting and deserves further study, as it might help to understand how CSPP1 accumulations on microtubules are formed. We removed the term “dimerization domain” to avoid confusion.

7. Could the increased acetylation result from a more direct activity of CSPP1? Rather than simply stabilizing all MTs.

Reply: Until now, CSPP1 has not been shown to have an acetylase activity and it contains no conserved domains that would hint at such an activity, but a potential crosstalk between CSPP1 and microtubule acetylases in the microtubule lumen will require further investigation.

8. Because of mammalian target of rapamycin (mTOR)'s well established transcription factor roles, I would not recommend using the acronym MTOR here.

Reply: To indeed avoid confusion with the mammalian target of rapamycin (mTOR), but still acknowledge the domain previously named MTOR by Patzke et al., 2006, we have changed the acronym to MTORG (MicroTubule ORganizing region).

9. Page 11 "...binding to Taxol-stabilized microtubules was stronger and faster (Fig. 5A, B)." How is it known that the binding is "stronger"?

Reply: We apologize for imprecise wording. We did not want to imply that the binding is stronger but wanted to point out that the binding to Taxol-stabilized microtubules was faster and the CSPP-L signal intensity was higher. We have modified the text on page 12 to avoid confusion.

Reviewer #3 (Comments to the Authors (Required)):

The authors present a detailed and comprehensive characterization of CSPP1, a protein associated with cilia and centrosomes, and show that CSPP1 alters microtubule dynamics in a very interesting way and does so by binding to the inside of the microtubule lumen, reminiscent of taxanes. The thorough study includes in vitro and cellular microtubule dynamics characterization, along with impressive cryoET and MINIFLUX imaging that clearly show the protein inside the microtubule. The presentation is a bit overwhelming in places due to the abundance of data, but in general the presentation is clear and convincing, and the manuscript is well written other than a few grammar slips.

We had no major comments, but did have a number of points on to clarify the presentation and address a few controls.

Comments:

1. Introduction - The intro is short with a long third paragraph previewing the results. It would be nice to instead have a paragraph three on MIPs and taxanes binding to inside microtubules, which is a major point of the manuscript, and then conclude with a shorter paragraph 4 previewing results.

Reply: Thanks for this suggestion. We have now included on the first page of the Introduction some text about microtubule intraluminal proteins and taxanes and shortened the last paragraph of the Introduction.

2. Somewhere near the start of results, the authors should include a sentence justifying use of singlet microtubules for these experiments, since it is generally known and centriolar protein.

Reply: We have added the justification of using singlet microtubules at the start of the Results. CSPP1 is a ciliary tip protein, and recent work from the Pigino lab showed that plus ends of primary cilia are microtubule singlets (Kiesel et al, 2020; PMID: 32989303).

3. The authors should describe the previous work that establishes the Microtubule organizing domain from the Centrosome targeting domain in the overall protein structure of Fig 1A.

Reply: We have modified the text to describe this work (page 5).

4. Because CSPP1 alters microtubule growth rates, how do the authors ensure they are calling the proper microtubule polarity in their in vitro experiments?

Reply: We have measured the growth rates of the fastest growing microtubule end. We have found that the growth rates of these ends only mildly decreased but were not as low as the measured minus-end growth rate of control microtubules, which we have previously determined (Doodhi et al., 2016; PMID: 27321995). The faster growing ends in presence of CSPP1 and EB3 grew at a similar rate as the plus ends in presence of EB3 alone. Additionally, we have combined minus-end binding protein CAMSAP3 with CSPP1 (Fig. S4C), where CAMSAP3 labels the minus end; this experiment is fully consistent with our assignment of microtubule polarity in other experiments.

5. The point that when the microtubule enters a pre-catastrophe state the affinity of CSPP goes up as the affinity for EB3 goes down is a very interesting one. However, the figures are not particularly convincing on this point, and it isn't clear why in Fig 1 it jumps from in vitro to cellular images. The authors should consider using S1F instead of Fig 1G-I to show this point. Alternatively, the in vitro data is really in fig 2, so reworking the order in the text to build the arguments could address this.

Reply: Our aim was to reproduce the main observations from our in vitro data in cells, in order to show that CSPP1 can display the same behavior as observed in a simple in vitro system also in a cellular environment, where other microtubule regulators are present. We prefer the current organization of the figure panels because we first confirm that the phenomena we observe in vitro (CSPP1 binding to growing ends, pausing) also occur in cells, and then, in Fig. 2, we compare the behavior of CSPP1 with the taxane derivative Fchitax-3 and come to the conclusion that growth perturbations promote CSPP1 binding, similar to what we have previously shown for Fchitax-3 (Rai et al., 2020, PMID: 31819210).

6. Page 7, 10 lines from bottom, the "mentioned previously" and "previously performed analysis" is confusing - not clear what previous you are talking about.

Reply: The 'mentioned previously' we have modified to 'mentioned above' to clearly indicate that we mentioned this earlier in our paper. The 'previously performed analyses' was changed to 'previously published analyses', to make the statement more clear (page 8 of the revised manuscript).

7. The result of 0.5 nM CSPP without versus with vinblastine in Fig 2G is a particularly impressive result.

Reply: We completely agree.

8. The MCAK experiment in Fig 2K is complicated because of all of the competing processes going on (catastrophe, pausing, etc). The text gives few details, and the addition of the MCAK point as written confuses the arguments rather than clarifying. The authors should either expand on the MCAK point to develop it, or they should remove it to streamline the arguments and put it in the supplemental.

Reply: We have expanded our text on pages 7-8 to explain more clearly the reasoning behind the MCAK data.

9. Relevant to Fig. 3A, it isn't clear how it is determined that full-length is a dimer. Presumably this is from the alpha-fold structure. Related to this is that the authors should show their alpha-fold structure, at least in the supplement. And further description is needed to clarify how they determine that the DD domain is in fact a dimerization domain.

Reply: We reasoned that full length CSPP-L would be a dimer because in experiments with deletion mutants (Fig. 3), the predicted helical region H5, which could potentially form a coiled coil, could be functionally substituted with a leucine zipper. We have performed additional experiments to investigate the oligomerization state of CSPP-L (see also our answer to comment 4 of reviewer 1). In short, it turned out that GFP-CSPP-L initially binds to microtubules as a monomer, but then a second molecule of GFP-CSPP-L is recruited to the site where the first one was bound (new Fig. 3B,C). We have no precise information on whether additional oligomerization takes place and on which part of CSPP1 is responsible for the oligomerization, although the helical region H5 seems to be a likely candidate. We find these data interesting, because they might help to explain why CSPP1 binds better to slowly growing microtubule tips. We propose that stable binding requires more than one CSPP1 molecule, but the oligomerization process is slow and limited by tube closure, and therefore CSPP1 accumulates much more prominently at slowly growing microtubule plus ends (now discussed on page 16). Using AlphaFold to investigate homo-oligomerization of CSPP1 did not provide conclusive answers, so we prefer not to include this analysis in the paper.

10. Fig. 4 is beautiful. One suggestion - it would be nice to see the distributions from D and E that are used to calculate the FWHM in F. Additionally, it would be nice to see in I and J the distribution of distances from the centroid rather than a slice - this would be more comprehensive analysis of the

data.

Reply: We do agree that it is nice to show how we arrived at the determined FWHM values. Therefore, we have now included an example with the explanation of the analysis pipeline in Fig. S3C. Furthermore, we have performed extensive additional analyses suggested by the reviewer, but they were unfortunately not sufficiently conclusive to be included in the paper. We have included in the Appendix below a detailed account of the attempted data analysis approaches and the issues we have encountered.

11. The authors should more carefully define what they mean by "taper", "flare" and "raggedness".

Reply: We added a sentence to clarify this. Exact definitions of quantified parameters are given in Fig. 6L.

12. Page 11: "...Taxol-stabilized microtubules, which are known to acquire extensive lattice defects when incubated without soluble tubulin...". This is a strong statement, and it is referenced, but the authors should expand on this point to clarify their precise meaning and not cast aspersion on all taxol-stabilized (It is not actually "well known".)

Reply: We agree that we should have indicated more clearly that Taxol-stabilized microtubules fall apart when incubated without free tubulin and free Taxol (because Taxol binding to microtubules is reversible). We now indicated more clearly how Taxol-stabilized microtubules were used in different experiments (e.g., pages 12,13 and 14).

13. Fig 6 E&F and bottom page 12 "This suggests that CSPP-L binds to microtubules more efficiently when they grow slowly, either due to a slow on-rate...". This is a faulty kinetic argument because if it has a slow on-rate, then the difference is simply the duration the microtubules are present under the two conditions, and so the efficiency of binding is not different, only the duration allowed for binding is different.

Reply: We have now removed this sentence and instead made an addition to the Discussion (page 16), where we describe our thinking more clearly. As explained above, we now think (based on the new data in Fig. 3B, C), that CSPP1 initially binds as a monomer, but then at least one additional CSPP1 molecule binds, and this likely stabilizes the binding. This oligomerization step is likely limited by the tube closure, and therefore stable CSPP1 accumulations preferentially form on slowly growing microtubule ends.

14. There is a controversy in the microtubule dynamics field about whether plus-ends grow as sheets (the textbook explanation) or as isolated, curved protofilaments that straighten and laterally associate (the McIntosh model). The authors are remarkably understated in addressing this controversy, while presenting extremely strong data showing curved protofilaments that avoid using the somewhat controversial McIntosh protofilament tracing approach. It is understandable if the authors want to stay above the fray, but they should more deliberately point more their clear evidence of curved protofilaments at the growing plus-end. It would even be justified to incorporate the very interesting Fig. S4D results into Fig. 6.

Reply: We agree with the reviewer that our data are consistent with "the McIntosh model" of microtubules growing with bent protofilaments and now mention this in the text on page 15. We indeed do not observe flat sheets of protofilaments. However, the focus of this study was to quantify the effects of CSPP1. Using the protofilament tracing approach, we confirm that protofilament shapes are not affected by presence of luminal densities of CSPP1. This is not surprising given that we don't

observe CSPP1 densities at bent protofilaments, therefore data in Fig. S4D are rather a control experiment.

15. The reference list uses inconsistent formatting and needs to be cleaned up.

We have improved the formatting of the reference list.

Appendix

Response to Reviewer 3, Comment 10: *Additionally, it would be nice to see in I and J the distribution of distances from the centroid rather than a slice - this would be more comprehensive analysis of the data.*

The relevant figure from the submitted manuscript, corresponding to the comment, is shown above. The reviewer suggested that it would be nice to perform a distribution of distance analysis on the 3D MINFLUX dataset. It turns out to be difficult if not impossible. We will list here the technical challenges we have encountered.

Essentially, the goal is to perform a specific type of analysis on the 2-channel 3D MINFLUX-based microtubule imaging. If we understood correctly, the reviewer suggested to extract the signal of each channel, plot the signal against the distance to the centroid of the microtubule (center-line in 3D) and compare the two channels. The technical challenges are:

1. Reconstruct the microtubule in 3D from MINFLUX localization data, which could be sparse.
2. Extract faithfully the 3D skeleton (i.e.: a 3D spline, which consist of the centroids, or the central line of the microtubule) from the reconstructed microtubule.
3. Determine the curvature, or the angle along the 3D skeleton, in order to extract the geometrical plane, that perpendicularly dissects the microtubule, at each point of the 3D skeleton.
4. Visualize and quantitatively analyze the results.

First, in 2D cases (manuscript Fig.4 G and H) we have already seen that the MINFLUX signal is sparse and discrete. We devised a method that works to reconstruct microtubules in 2D. This method doesn't scale up easily to 3D so far. This is partly because the data points to interpolate in 3D are even more sparse and hard to be linked together, even by eye. We have nevertheless tried a deep learning based approach (ANNA-PALM: <https://imod.pages.pasteur.fr/anna-palm-web/#/>; <https://github.com/imodpasteur/Anet-ImageJ>) to reconstruct the imaged structure. With the pre-trained models, one can reconstruct a microtubule from the localization data. Unfortunately, this does not work with 3D data. Besides, this method currently only allows rather small images (e.g.: 512x512 pixels), and is not qualitatively better than the approach we used. To summarize, a good 3D reconstruction of the microtubule structure from MINFLUX data is difficult. However, we can, to some extent, bypass this problem by pre-selecting relatively straight and clean microtubule candidates from the dataset (e.g.: a rather straight microtubule filament without other structures in its vicinity). One such example shown in Fig.1 below, and we will use it for further explanations.

Figure 1: 3D rendering of 2-channel 3D MINFLUX data Figure 2: view from rotated angle from Fig.1

Figure 3: skeleton of Fig.1 in 3D view Figure 4: distance transform of 3D skeleton, highlighting An equidistant line around 20 pixels from the skeleton.

Given a good input dataset (Fig. 1), with some filtering and enhancement steps similar to our presented method for the 2D cases, we can extract the 3D skeleton of the microtubule as shown in Fig. 3. It should be noted that we manually tweaked the parameters and trimmed a bit the result, so this procedure is not yet automatable or readily applicable to other datasets.

With the 3D skeleton, for each given 3D point on the skeleton, its equidistant points form a spherical shell, which is apparently not what we would need. The equidistant points we want would be a circle in a plane that is perpendicular to the interpolated spline segments from the skeleton. Therefore, to

compute the equidistant points along the central line of the microtubule, we would also need to compute the 3D angle of the short linear segments between two adjacent points in 3D. With that, we then would need to compute the plane that is orthogonal to the linear segments and passes through the center of these segments. We would then dissect the original 3D dataset with this orthogonal plane and measure the distance of each point on that plane to the center point, which should be on our skeleton of the microtubule. In the case of a completely straight line, we would simply look from the side. This is basically shown in the original manuscript Figure 4 I and J, the upper right panel. Fig. 2 shows the case when we look from a similar angle using the example in Fig. 1.

The above geometrical computation, although challenging, is technically possible. However, there is a prerequisite that needs to be fulfilled: the angle θ , or the reference axis of each orthogonal plane. In other words, for each computed orthogonal plane, one needs to determine where to draw the horizontal and vertical axis. However, this is difficult to define, as the 3D spline can twist and rotate along its length in 3D.

A similar question was posted on the image.sc forum (<https://forum.image.sc/t/straighten-along-a-segmented-line-in-3d/41476/2>). There, the final solution is also to simplify the 3D dataset to a 2D dataset, by projecting the 3D dataset to a 2D plane with user manual input, and analyze the 2D dataset. In the comment list of the post, several image analysis expert also pointed out the technical challenges similar to what we described. To our knowledge, there is still not an easy way or readily usable solution for the proposed problem.

As an alternative, a way to bypass this difficulty is to compute the Euclidean Distance Transform (EDT) from the 3D skeleton. The EDT is basically the Euclidean distance of each 3D point in the dataset, that indicates the distance between the single 3D point and its nearest skeleton point. An example of the EDT from the skeleton is shown in Fig.4. The pixel value in EDT is basically the distance between each point and the nearest skeleton point. When we highlight a certain value in range, for example 20 pixels, we see the red line that delineates around the skeleton in that 2D slice. In fact, if we sum all points in the data present in this 20 pixel value range, we basically sum up all the equidistant points in the 3D space that are 20 pixels away from our skeleton. Therefore, in this way we can to some extent bypass the problem of 3D geometrical transformation and computation. We therefore convert the 3D problem to 2D problem, as we can compute the equidistant points in each 2D slice one after another and sum them up in the end. The drawback of this method is that it would contain the points located at the two ends of the skeleton, which for our purpose were not supposed to be included.

Figure 5: sum of signal vs. distance plot of the two channels of 3D MINFLUX dataset shown in Fig.1

Fig. 5 shows the plots resulting from the described EDT approach. With the example shown in Fig. 1 and the generated 3D skeleton, we computed the EDT and plotted the sum of the signals against their distance to the central line of the microtubule. Note that the X axis shows the distance as a radius (not diameter as we did in 2D case), since it's the distance from a given point to the center of the microtubule. We can see that the signal in the 1st channel (red) reaches a peak around 20 nm radius,

and 2nd channel (green) reaches the first peak around 5 nm. Alternatively, this indicates a diameter around 40 nm for the 1st channel, and 10 nm for the 2nd channel.

However, this is far from a precise measurement, especially for the 2nd channel. We can see that the total sum of signal values even at the peak is around 25, and the whole curve looks quite noisy. We would need to systematically collect much more data to start trying quantitative analysis based on this approach. And we should also bear in mind that to get this result, we hand-selected the best quality 3D data that contains a single and straight microtubule.

To summarize, we conclude that the proposed 3D analysis is technically very challenging. It would be demanding regarding the size, quality and certain properties of the data (straight single microtubules). It is hampered by several unsolved technical challenges. Since the outcome of such analysis is not critical to justify the major scientific findings in this paper, we have not continued with it and have not included it in the paper.

January 7, 2023

RE: JCB Manuscript #202208062R

Dr. Anna Akhmanova
Utrecht University
Biology
Cell Biology, Neurobiology and Biophysics, Department of Biology, Faculty of Science Utrecht University Padualaan 8
Utrecht 3584 CH
Netherlands

Dear Dr. Akhmanova:

Thank you for submitting your revised manuscript entitled "CSPP1 stabilizes growing microtubule ends and damaged lattices from the luminal side". Two of the original reviewers have now assessed your revised manuscript and, as you can see, they are satisfied with revisions. Thus, we would be happy to publish your paper in JCB pending final revisions necessary to meet our formatting guidelines (see details below). In your final revision, please be sure to address the request of reviewer #2 of better defining the term "raggedness" via appropriate text edits.

To avoid unnecessary delays in the acceptance and publication of your paper, please read the following information carefully. Please go through all the formatting points paying special attention to those marked with asterisks.

A. MANUSCRIPT ORGANIZATION AND FORMATTING:

Full guidelines are available on our Instructions for Authors page, <https://jcb.rupress.org/submission-guidelines#revised>.
Submission of a paper that does not conform to JCB guidelines will delay the acceptance of your manuscript.

1) Text limits: Character count for Articles and Tools is < 40,000, not including spaces. Count includes title page, abstract, introduction, results, discussion, and acknowledgments. Count does not include materials and methods, figure legends, references, tables, or supplemental legends.

2) Figures limits: Articles and Tools may have up to 10 main text figures.

*** Please note that main text figures should be provided as individual, editable files.

3) Figure formatting:

*** Molecular weight or nucleic acid size markers must be included on all gel electrophoresis. Please add MW markers to Fig S1G.

*** Scale bars must be present on all microscopy images, including inset magnifications. Please add scale bars to Figs. 4C and S1J (inset magnifications).

Also, please avoid pairing red and green for images and graphs to ensure legibility for color-blind readers. If red and green are paired for images, please ensure that the particular red and green hues used in micrographs are distinctive with any of the colorblind types. If not, please modify colors accordingly or provide separate images of the individual channels.

4) Statistical analysis:

Error bars on graphic representations of numerical data must be clearly described in the figure legend.

The number of independent data points (n) represented in a graph must be indicated in the legend. Please, indicate whether 'n' refers to technical or biological replicates (i.e. number of analyzed cells, samples or animals, number of independent experiments).

If independent experiments with multiple biological replicates have been performed, we recommend using distribution-reproducibility SuperPlots (please, see Lord et al., JCB 2020) to better display the distribution of the entire dataset, and report statistics (such as means, error bars, and P values) that address the reproducibility of the findings.

*** Statistical methods should be explained in full in the materials and methods in a separate section. Please, provide more detail on the statistical methods used in the study.

For figures presenting pooled data the statistical measure should be defined in the figure legends.

Please also be sure to indicate the statistical tests used in each of your experiments (both in the figure legend itself and in a separate methods section) as well as the parameters of the test (for example, if you ran a t-test, please indicate if it was one- or two-sided, etc.).

If you used parametric tests in your study (i.e. t-tests), you should have first determined whether the data was normally distributed before selecting that test. In the stats section of the methods, please indicate how you tested for normality. If you did not test for normality, you must state something to the effect that "Data distribution was assumed to be normal but this was not formally tested."

5) Abstract and title:

The abstract should be no longer than 160 words and should communicate the significance of the paper for a general audience.

The title should be less than 100 characters including spaces. Make the title concise but accessible to a general readership.

6) Materials and methods:

Should be comprehensive and not simply reference a previous publication for details on how an experiment was performed. The text should not refer to methods "...as previously described."

Also, the materials and methods should be included with the main manuscript text and not in the supplementary materials.

7) For all cell lines, vectors, constructs/cDNAs, etc. - all genetic material: please include database / vendor ID (e.g., Addgene, ATCC, etc.) or if unavailable, please briefly describe their basic genetic features, even if described in other published work or gifted to you by other investigators (and provide references where appropriate).

Please be sure to provide the sequences for all of your oligos: primers, si/shRNA, RNAi, gRNAs, etc. in the materials and methods.

You must also indicate in the methods the source, species, and catalog numbers/vendor identifiers (where appropriate) for all of your antibodies, including secondary. If antibodies are not commercial, please add a reference citation if possible.

8) Microscope image acquisition:

The following information must be provided about the acquisition and processing of images:

- a. Make and model of microscope
- b. Type, magnification, and numerical aperture of the objective lenses
- c. Temperature
- d. imaging medium
- e. Fluorochromes
- f. Camera make and model
- g. Acquisition software
- h. Any software used for image processing subsequent to data acquisition. Please include details and types of operations involved (e.g., type of deconvolution, 3D reconstitutions, surface or volume rendering, gamma adjustments, etc.).

10) Supplemental materials:

There are strict limits on the allowable amount of supplemental data. Articles/Tools may have up to 5 supplemental figures. There is no limit for supplemental tables.

*** Please note that supplemental figures and tables should be provided as individual, editable files.

A summary of all supplemental material should appear at the end of the Materials and Methods section (please see any recent

JCB paper for an example of this summary).

11) Video legends: Should describe what is being shown, the cell type or tissue being viewed (including relevant cell treatments, concentration and duration, or transfection), the imaging method (e.g., time-lapse epifluorescence microscopy), what each color represents, how often frames were collected, the frames/second display rate, and the number of any figure that has related video stills or images.

12) eTOC summary:

A ~40-50 word summary that describes the context and significance of the findings for a general readership should be included on the title page.

*** The statement should be written in the present tense and refer to the work in the third person. It should begin with "First author name(s) et al..." to match our preferred style.

13) Conflict of interest statement:

JCB requires inclusion of a statement in the acknowledgements regarding competing financial interests. If no competing financial interests exist, please include the following statement: "The authors declare no competing financial interests."

14) A separate author contribution section is required following the Acknowledgments in all research manuscripts.

*** All authors should be mentioned and designated by their first and middle initials and full surnames and the CRediT nomenclature is encouraged (<https://casrai.org/credit/>).

15) ORCID IDs: ORCID IDs are unique identifiers allowing researchers to create a record of their various scholarly contributions in a single place. At resubmission of your final files, please consider providing an ORCID ID for as many contributing authors as possible.

16) Materials and data sharing:

All animal and human studies must be conducted in compliance with relevant local guidelines, such as the US Department of Health and Human Services Guide for the Care and Use of Laboratory Animals or MRC guidelines, and must be approved by the authors' Institutional Review Board(s). A statement to this effect with the name of the approving IRB(s) must be included in the Materials and Methods section.

*** As a condition of publication, authors must make protocols and unique materials (including, but not limited to, cloned DNAs; antibodies; bacterial, animal, or plant cells; and viruses) described in our published articles freely available upon request by researchers, who may use them in their own laboratory only. All materials must be made available on request and without undue delay. We strongly encourage to deposit all the cell lines/strains and reagents generated in this study in public repositories.

All datasets included in the manuscript must be available from the date of online publication, and the source code for all custom computational methods, apart from commercial software programs, must be made available either in a publicly available database or as supplemental materials hosted on the journal website. Numerous resources exist for data storage and sharing (see Data Deposition: <https://rupress.org/jcb/pages/data-deposition>), and you should choose the most appropriate venue based on your data type and/or community standard. If no appropriate specific database exists, please deposit your data to an appropriate publicly available database.

17) Please note that JCB now requires authors to submit Source Data used to generate figures containing gels and Western blots with all revised manuscripts. This Source Data consists of fully uncropped and unprocessed images for each gel/blot displayed in the main and supplemental figures. The Source Data files will be directly linked to specific figures in the published article.

Since your paper includes cropped gel and/or blot images, please be sure to provide one Source Data file for each figure that contains gels and/or blots along with your revised manuscript files. File names for Source Data figures should be alphanumeric without any spaces or special characters (i.e., SourceDataF#, where F# refers to the associated main figure number or SourceDataFS# for those associated with Supplementary figures). The lanes of the gels/blots should be labeled as they are in the associated figure, the place where cropping was applied should be marked (with a box), and molecular weight/size standards should be labeled wherever possible.

B. FINAL FILES:

Thank you for your attention to these final processing requirements. Please revise and format the manuscript and upload materials within 7 business days. If complications arising from measures taken to prevent the spread of COVID-19 will prevent you from meeting this deadline (e.g. if you cannot retrieve necessary files from your laboratory, etc.), please let us know and we can work with you to determine a suitable revision period.

Thank you for this interesting contribution, we look forward to publishing your paper in Journal of Cell Biology.

Sincerely,

Tarun Kapoor
Monitoring Editor
Journal of Cell Biology

Lucia Morgado-Palacin, PhD
Scientific Editor
Journal of Cell Biology

Reviewer #1 (Comments to the Authors (Required)):

The authors have adequately addressed initial comments raised by me, and I have not in the process raised additional concerns or comments from me. The structure-function reconstititional and cellular work will further the field's mechanistic understanding of regulators of ciliary microtubules. I view the work as acceptable for publication.

Reviewer #2 (Comments to the Authors (Required)):

Overall, the authors satisfactorily addressed my initial review.

I appreciate the difficulties in showing the in vivo relevance to the beautiful in vitro work that is presented.

Regarding Fig 6K-M: I understand the existing literature describing MT end tapering (variance in PF length) relative to growth. However, this manuscript uses the term raggedness. Thus, a definition of raggedness and tapering relative to the existing literature should be defined. Moreover, the methods for how "raggedness" is calculated in this manuscript should be defined. This term was not used in the McIntosh et al 2018 manuscript that is referenced in the Methods. The text on Page 15 highlights long tapers (indicating increased variance) but then 6M refers to raggedness. I think these terms should be consistently used with clearly defined terms. Is this one protofilament relative to the mean of the others or how is this long taper / raggedness really defined? I also wonder if the variance in "raggedness" here is important to the conclusions described. While I do not find these data in their current form, particularly strong, I am fine with inclusion of this figure in the manuscript if the text and Methods more clearly defines raggedness.